# Thioredoxin is a metabolic rheostat controlling regulatory B cells

Hannah F. Bradford [1] ✉, Thomas C. R. McDonnell[2], Alexander Stewart[3], Andrew Skelton[4], Joseph Ng [5], Zara Baig[1], Franca Fraternali[5], Deborah Dunn-Walters [3], David A. Isenberg [6], Adnan R. Khan [4], Claudio Mauro [7] & Claudia Mauri [1]✉

Metabolic programming is important for B cell fate, but the bioenergetic requirement for regulatory B (B_{reg}) cell differentiation and function is unknown. Here we show that B_{reg} cell differentiation, unlike non-B_{reg} cells, relies on mitochondrial electron transport and homeostatic levels of reactive oxygen species (ROS). Single-cell RNA sequencing analysis revealed that *TXN*, encoding the metabolic redox protein thioredoxin (Trx), is highly expressed by B_{reg} cells, unlike Trx inhibitor *TXNIP* which was downregulated. Pharmacological inhibition or gene silencing of *TXN* resulted in mitochondrial membrane depolarization and increased ROS levels, selectively suppressing B_{reg} cell differentiation and function while favoring pro-inflammatory B cell differentiation. Patients with systemic lupus erythematosus (SLE), characterized by B_{reg} cell deficiencies, present with B cell mitochondrial membrane depolarization, elevated ROS and fewer Trx+ B cells. Exogenous Trx stimulation restored B_{reg} cells and mitochondrial membrane polarization in SLE B cells to healthy B cell levels, indicating Trx insufficiency underlies B_{reg} cell impairment in patients with SLE.

B_{reg} cells limit host-mediated immunopathology in response to unrestrained inflammatory challenge via the production of interleukin (IL)-10, IL-35 and transforming growth factor β (TGFβ)[1]. B_{reg} cells maintain peripheral tolerance by several direct and indirect suppressive mechanisms, targeting a wide variety of immune effector cell types including CD4+ T cells, invariant natural killer T cells and monocytes[2]. B_{reg} cells are reduced in number and function in autoimmunity, including in SLE, and it has been shown that restoration of B_{reg} numbers and function in autoimmune patients is associated with better clinical outcomes[3,4]. Conversely, in several studies of murine cancer models and human tumors, B_{reg} cells support a pro-tumorigenic response, are enriched in tumor tertiary lymphoid structures and are associated with a poor prognosis[5,6]. In humans, B_{reg} cells account for 1–2% of the total B cell population in peripheral blood, but upon activation, for example, via TLR9 engagement, they compose up to 15–20% of total B cells. Although in response to activation the majority of B_{reg} cells reside within the immature and plasmablast populations of B cells, to a certain extent, B cells at most stages of maturity can differentiate into B_{reg} cells[1].

The metabolic demands of B cells adjust according to their functional needs. Naive resting B cells are largely inactive and metabolically quiescent. Upon BCR or co-receptor engagement, they become metabolically active, utilizing glycolysis, fatty acid oxidation (FAO) and oxidative phosphorylation (OXPHOS) to meet their energetic demands[7–9]. In mice, activation of naive B cells induces an upregulation of glycolysis, with a progressive reliance on OXPHOS during plasma cell differentiation, and mature plasma cells additionally rely heavily

[1]Institute of Immunity and Transplantation, Pears Building, UCL Division of Infection and Immunity, University College London, London, UK. [2]Department of Biochemical Engineering, University College London, London, UK. [3]School of Biosciences and Medicine, University of Surrey, Guildford, UK. [4]UCB Pharma, Slough, Berkshire, UK. [5]Institute of Structural and Molecular Biology, University College London, London, UK. [6]Centre for Rheumatology, Division of Medicine, University College London, London, UK. [7]Institute of Inflammation and Ageing, College of Medical and Dental Sciences, University of Birmingham, Birmingham, UK. ✉e-mail: hannah.bradford.12@ucl.ac.uk; c.mauri@ucl.ac.uk

on glucose uptake and mitochondrial pyruvate import to facilitate antibody glycosylation[10,11]. While the understanding of B cell metabolism continues to expand, the metabolic requirements of $B_{reg}$ cells remain unknown.

The thioredoxin system, including the oxidoreductase thioredoxin (Trx), thioredoxin reductase (TrxR) and NADPH, regulates multiple cellular processes, including gene expression, the antioxidant response that maintains the redox state of cells, apoptosis and proliferation, with thioredoxin-interacting protein (Txnip) acting as a negative regulator of Trx function[12,13]. There are two human isoforms of Trx: Trx, located in the mitochondrial intermembrane space, nucleus and cytoplasm, and the mitochondria-exclusive Trx2 isoform[14].

Trx and Trx2 function as an antioxidant system that maintains thiol-related redox status by neutralizing the oxidizing effects of excess ROS[15]. Dysregulation of the Trx system affects cellular functions and cell fate, including survival and cell death, leading to human diseases including cancer and autoimmunity[16,17]. Here we aim to identify the metabolic requirements and the accompanying molecular pathways regulating the differentiation of B cells into $B_{reg}$ cells.

## Results

### IL-10$^+$ $B_{reg}$ cells use OXPHOS to meet their metabolic needs

Several surface markers have been associated to human IL-10$^+$ $B_{reg}$ cells; however, the presence of *IL10* messenger RNA and IL-10 expression remains the hallmark of human $B_{reg}$ cell identification. Stimulation with CpGC, a TLR ligand, expands $B_{reg}$ cells to 10–20% of total B cells (Supplementary Fig. 1a,b). To assess the metabolic pathways preferentially used by $B_{reg}$ cells, B cells were negatively purified from healthy peripheral blood mononucleated cells (PBMCs) and stimulated with CpGC, previously shown to promote the expansion of IL-10$^+$ $B_{reg}$ cells within CD24$^{hi}$CD38$^{hi}$ (immature), CD24$^{int}$CD38$^{int}$ (mature-naive), CD24$^+$CD38$^{lo}$ (memory) and CD24$^{lo/-}$CD38$^{hi}$ (plasmablasts) B cell subsets (Supplementary Fig. 1c,d), together with inhibitors of different metabolic pathways or under nutrient starvation and/or repleting conditions (schematic in Fig. 1a). Inhibition of glycolysis by 2-deoxyglucose (2-DG) (a glucose analog) or FAO by etomoxir (carnitine palmitoyltransferase I (CPT1) inhibitor) or malonyl Co-A (allosteric inhibitor limiting uptake of fatty acids by mitochondria) did not alter $B_{reg}$ cell intracellular expression or secretion of IL-10 (Fig. 1b and Extended Data Fig. 1a). As 2-DG has been shown to have off-target effects and to further exclude the role of aerobic glycolysis in $B_{reg}$ cell differentiation, B cells were cultured in glucose-free media, or in glucose-free media replenished with high-glucose concentrations. B cells grown under these conditions did not show alteration in the IL-10$^+$ $B_{reg}$ cell frequencies compared with the control group (Extended Data Fig. 1b). Replacement of glucose with galactose, a respiratory substrate known to slow glycolysis and to promote a compensatory increase in OXPHOS[18], boosted IL-10$^+$ B cell differentiation compared with the control group (Extended Data Fig. 1c).

Our findings, showing that 'forcing' B cells to utilize galactose expands $B_{reg}$ cells, suggest a reliance on OXPHOS for $B_{reg}$ cells to meet their energy demands. Given the importance of the mitochondrial respiratory chain for OXPHOS, we questioned if different complexes of the electron transport chain (ETC) pivot $B_{reg}$ cell differentiation and function. Inhibiting OXPHOS with rotenone (Complex I inhibitor) ablated B cell IL-10 expression and secretion, and partial inhibition of OXPHOS following glutamine metabolism inhibition, by BPTES or culture in glutamine-free media, partially reduced IL-10$^+$ $B_{reg}$ cell frequencies (Fig. 1b and Extended Data Fig. 1d). Stimulation of B cells with the ETC Complex III inhibitor, antimycin A, impaired IL-10 intracellular expression and secretion, confirming the dependency of $B_{reg}$ cells on OXPHOS metabolism (Fig. 1b). There were no differences in B cell viability under these culture conditions (Extended Data Fig. 1e).

ETC blockade with rotenone or antimycin A inhibited B cell IL-10 expression after stimulation with TLR7 ligand R848 or CD40L, stimuli previously found to be important in the differentiation of $B_{reg}$ cells[3,19].

Stimulation with LPS, αBCR or IL-21 alone failed to induce IL-10$^+$ $B_{reg}$ cell differentiation (Extended Data Fig. 1f).

The reduction in IL-10 expression and production, observed following inhibition of Complex I and Complex III, was accompanied by a significantly reduced ability to suppress IFNγ production by autologous anti-CD3-activated CD4$^+$ T cells compared with vehicle-treated B cells (Fig. 1c). Additionally to IL-10, rotenone and antimycin A suppressed the expression of IL-35 and the inactive form of TGFβ (latency-associated peptide) (Extended Data Fig. 2a,b), immunoregulatory cytokines previously associated with $B_{reg}$ cell function[1]. The ability of B cells to produce pro-inflammatory cytokines including TNF, IL-6 and GM-CSF remained unaltered following OXPHOS inhibition (Extended Data Fig. 2c–e).

IL-10$^+$CD24$^{hi}$CD38$^{hi}$ transitional, IL-10$^+$CD24$^{int}$CD38$^{int}$ mature and IL-10$^+$CD24$^+$CD38$^{lo}$ memory B cell frequencies were equally reduced after rotenone and antimycin stimulation (Fig. 1d; gating strategy for B cell subsets after stimulation is shown in Supplementary Fig. 1c). We confirmed that the inhibition of IL-10$^+$ $B_{reg}$ cell differentiation by rotenone was independent of their stage of maturation as sorted transitional, mature and memory B cells cultured with both CpGC and rotenone failed to differentiate into IL-10$^+$ $B_{reg}$ cells (Extended Data Fig. 2f; Supplementary Fig. 1e shows the gating strategy for cell sorting). Our results confirm previously published data in mice showing that glycolysis, glutamine metabolism and OXPHOS are required for CD24$^{lo/-}$CD38$^{hi}$Blimp1$^+$ plasmablast differentiation, as inhibition of glutamine metabolism by BPTES, ETC inhibition by rotenone or antimycin A, and glucose or glutamine starvation all reduced CD24$^{lo/-}$CD38$^{hi}$ Blimp1$^+$ plasmablast frequencies (Extended Data Fig. 3a–f)[10,11].

### High concentrations of ROS inhibit IL-10$^+$ $B_{reg}$ cell differentiation

A major by-product of the ETC and OXPHOS is ROS; in addition, Complex I inhibition has been associated with a burst of mitochondrial ROS (Extended Data Fig. 4a), and ROS levels that exceed the capacity of the cellular antioxidant defense system induce oxidative stress[20]. Therefore, a tight control of ROS production is essential for the maintenance of cell functions. TLR signaling has also been shown to promote higher ROS production, for example, in macrophages[21].

To verify whether higher levels of ROS are detrimental for $B_{reg}$ cell differentiation, we stimulated B cells with increasing concentrations of CpGC and measured, respectively, the levels of cytoplasmic (CELLROX Orange) and mitochondrial (MitoSOX) ROS, and IL-10$^+$ $B_{reg}$ cell frequencies. Increased stimulation with CpGC led to incremental increases in cytoplasmic and mitochondrial ROS in B cells (Fig. 2a,b). IL-10$^+$ $B_{reg}$ cell frequencies increased up to 1 μM CpGC; however, at higher concentrations, IL-10 expression was significantly reduced compared with 1 μM CpGC (Fig. 2a,b and Extended Data Fig. 4b). Although higher concentrations of CpGC, leading to increased levels of ROS and decreased IL-10$^+$ $B_{reg}$ cell frequencies, did not alter the overall CD24$^{hi}$CD38$^{hi}$ transitional, CD24$^{int}$CD38$^{int}$ mature and CD24$^+$CD38$^{lo}$ memory B cell or CD24$^{lo/-}$CD38$^{hi}$Blimp1$^+$ plasmablast frequencies (Supplementary Fig. 2a,b), the IL-10$^+$ $B_{reg}$ cell frequencies within each of these subsets were significantly reduced after exposure to 5 μM CpGC compared with 1 μM CpGC (Extended Data Fig. 4c), with a net effect of a reduced ratio of IL-10$^+$ $B_{reg}$ cells to IL-10$^-$ plasmablasts (Extended Data Fig. 4d).

Next, we assessed whether ROS levels act as a rheostat, permitting (at low ROS levels) or inhibiting (at higher ROS levels) the differentiation of IL-10$^+$ $B_{reg}$ cells. B cells were cultured with a scalar range (0.1–5 μM) of CpGC alone or combined with *N*-acetylcysteine (NAC) (1 mM), which here we report to inhibit cytosolic and not mitochondrial ROS in B cells (Extended Data Fig. 4e,f), or with MitoTempo (10 μM), a selective inhibitor of mitochondrial ROS[22–24]. NAC and MitoTempo significantly reduced, respectively, cytoplasmic and mitochondrial ROS, at all CpGC doses tested (Extended Data Fig. 4e,g). Low levels of cytoplasmic and mitochondrial ROS, produced upon 0.1–0.5 μM CpGC challenge, play a crucial role in supporting the differentiation of $B_{reg}$

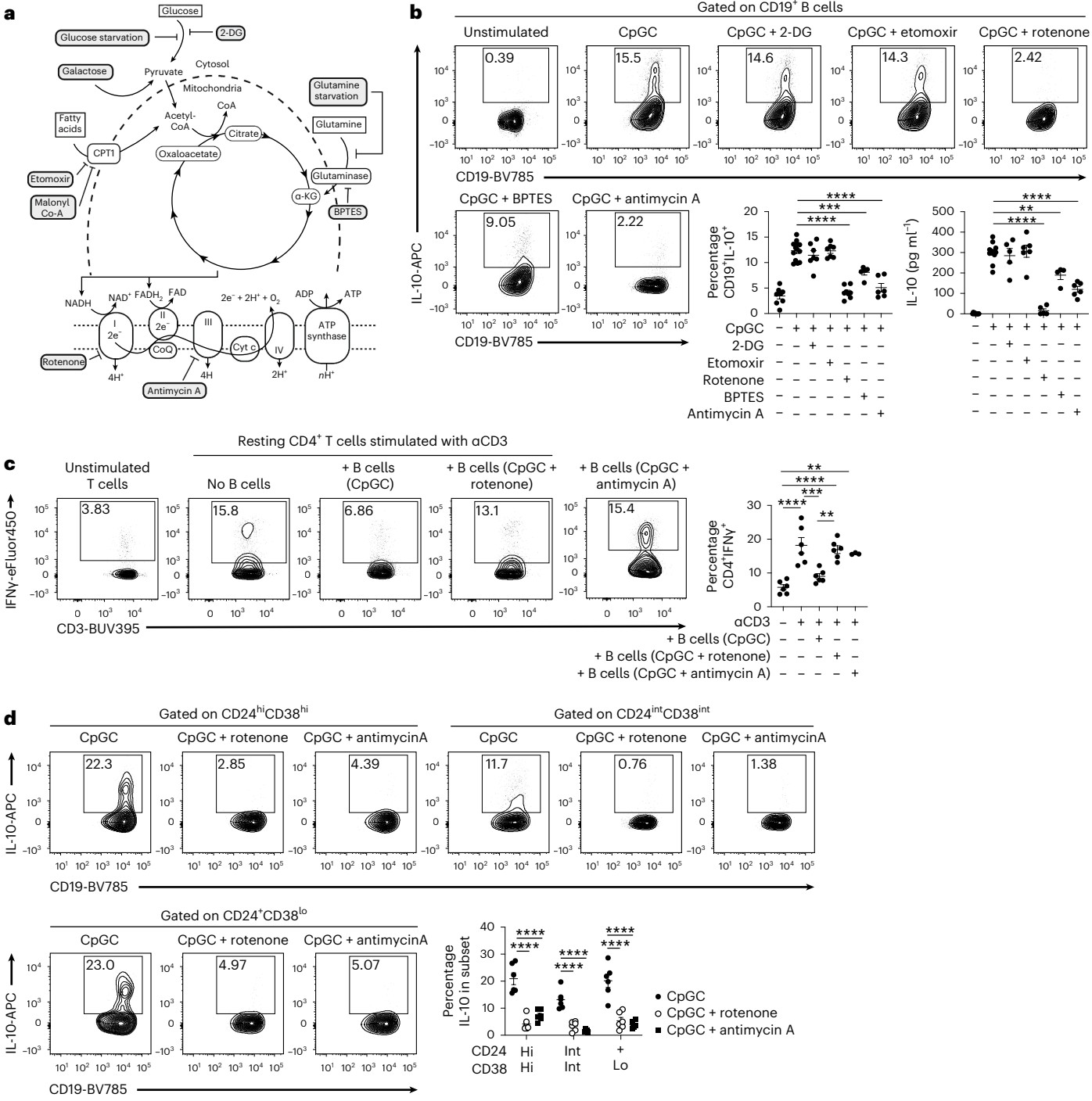

**Fig. 1 | ETC activity is critical for differentiation of IL-10⁺ B cells. a**, Schematic overview of cellular metabolism and the inhibition of glycolysis (2-DG, glucose starvation), FAO (etomoxir, malonyl co-A), glutamine metabolism (BPTES, glutamine starvation) and Complexes I and III of the ETC (rotenone and antimycin A, respectively). **b**, Representative contour plots and cumulative data show frequencies of CD19⁺IL-10⁺ B cells and IL-10 secretion following 72-h CpGC stimulation of isolated B cells with and without 1 mM 2-DG, 10 μM etomoxir, 1 μM rotenone, 500 nM BPTES or 10 nM antimycin A. **P = 0.001, ***P = 0.0002, ****P < 0.0001; data are representative of four independent experiments. **c**, Representative contour plots and cumulative data show the frequencies of CD3⁺CD4⁺IFNγ⁺ T cells after co-culture of CD4⁺ T cells stimulated with anti-CD3, with B cells preconditioned with CpGC, CpGC and rotenone, or CpGC and

antimycin A. n = 6 (n = 3 for antimycin A) biologically independent samples examined over two independent experiments. Top to bottom, **P = 0.0024, ****P < 0.0001, ***P = 0.0006, ****P < 0.0001, **P = 0.0031. **d**, Representative contour plots and cumulative data show frequencies of CD19⁺IL-10⁺ B cells within transitional (CD19⁺CD24ʰⁱCD38ʰⁱ), mature-naive (CD19⁺CD24ⁱⁿᵗCD38ⁱⁿᵗ) and memory (CD19⁺CD24⁺CD38ˡᵒ) gates following stimulation of isolated B cells for 72 h with CpGC, with and without 1 μM rotenone. n = 6 biologically independent samples examined over two independent experiments. ****P < 0.0001. Data were analyzed by one-way ANOVA followed by Tukey's multiple comparisons test (**b**,**c**) or two-way ANOVA followed by Sidak's test for multiple comparisons (**d**). Error bars are shown as mean ± s.e.m. Hi, high; Int, intermediate; Lo, low. Panel **a** created with Biorender.com.

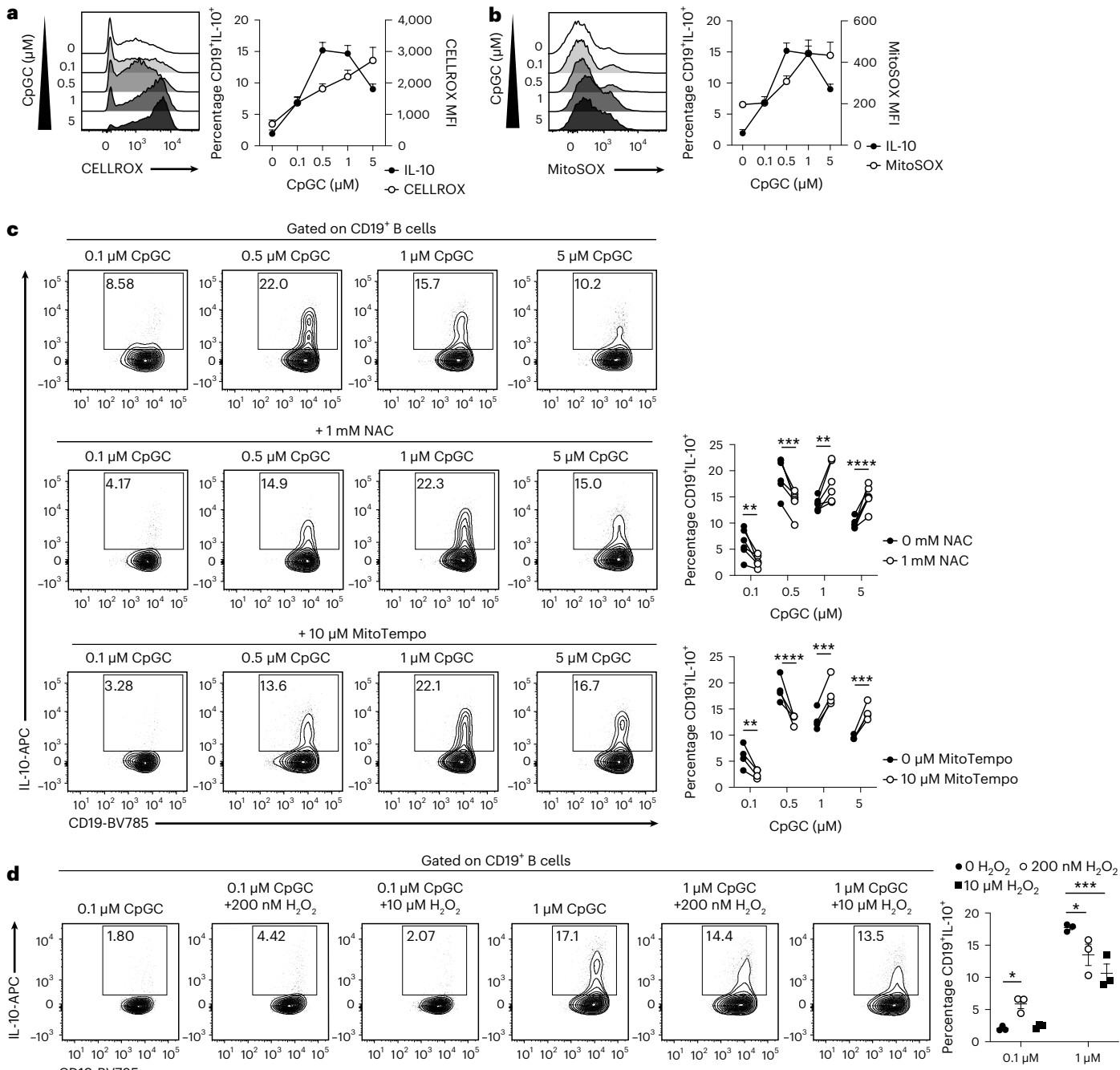

**Fig. 2 | High ROS levels impair B_reg cell differentiation. a,b**, Histograms and cumulative data show cytoplasmic ROS levels (CELLROX) (**a**) and mitochondrial ROS levels (MitoSOX) (**b**) following stimulation of isolated B cells with increasing concentrations of CpGC for 72 h, with paired CD19$^+$IL-10$^+$ B cell frequencies. $n = 5$ biologically independent samples, examined over two independent experiments. **c**, Representative contour plots and cumulative data show frequencies of CD19$^+$IL-10$^+$ B cells after stimulation of isolated B cells with 0.1–5 μM CpGC, with and without 1 mM NAC or 10 μM MitoTempo. $n = 6$ (NAC), $n = 4$ (MitoTempo) biologically independent samples examined over

three independent experiments. In graph order, **$P = 0.0053$, ***$P = 0.0001$, **$P = 0.0012$, ****$P < 0.0001$, **$P = 0.0067$, ****$P < 0.0001$, ***$P = 0.0002$, ***$P = 0.0003$. **d**, Representative contour plots and cumulative data show frequencies of CD19$^+$IL-10$^+$ B cells after stimulation with 0.1 μM or 1 μM CpGC, with and without low (0.2 μM) or high (10 μM) levels of H$_2$O$_2$. $n = 3$ biologically independent samples, examined over two independent experiments. *$P = 0.0384$, ***$P = 0.0006$, *$P = 0.0225$. Data were analyzed by two-way ANOVA followed by Sidak's test for multiple comparisons. Error bars are shown as mean ± s.e.m. MFI, median fluorescence intensity.

cells, as indicated by a marked reduction in B_reg cell frequencies during culture in the presence of NAC or MitoTempo (Fig. 2c). In contrast, inhibition of cytoplasmic or mitochondrial ROS in B cells stimulated with 1 μM or 5 μM CpGC significantly increased IL-10 expression by B cells compared with the control group (Fig. 2c), confirming that ROS levels need to be tightly controlled to achieve an optimal B_reg cell output.

Our data indicate that stimulation with low-dose CpGC (0.1 μM) fails to induce robust IL-10$^+$ B_reg cell differentiation. To investigate whether this deficiency is associated with insufficient ROS production, under these experimental conditions, we stimulated B cells with 0.1 μM CpGC and supplemented them with exogenous H$_2$O$_2$ (10 μM represents the highest dose that could be used without impairing cell

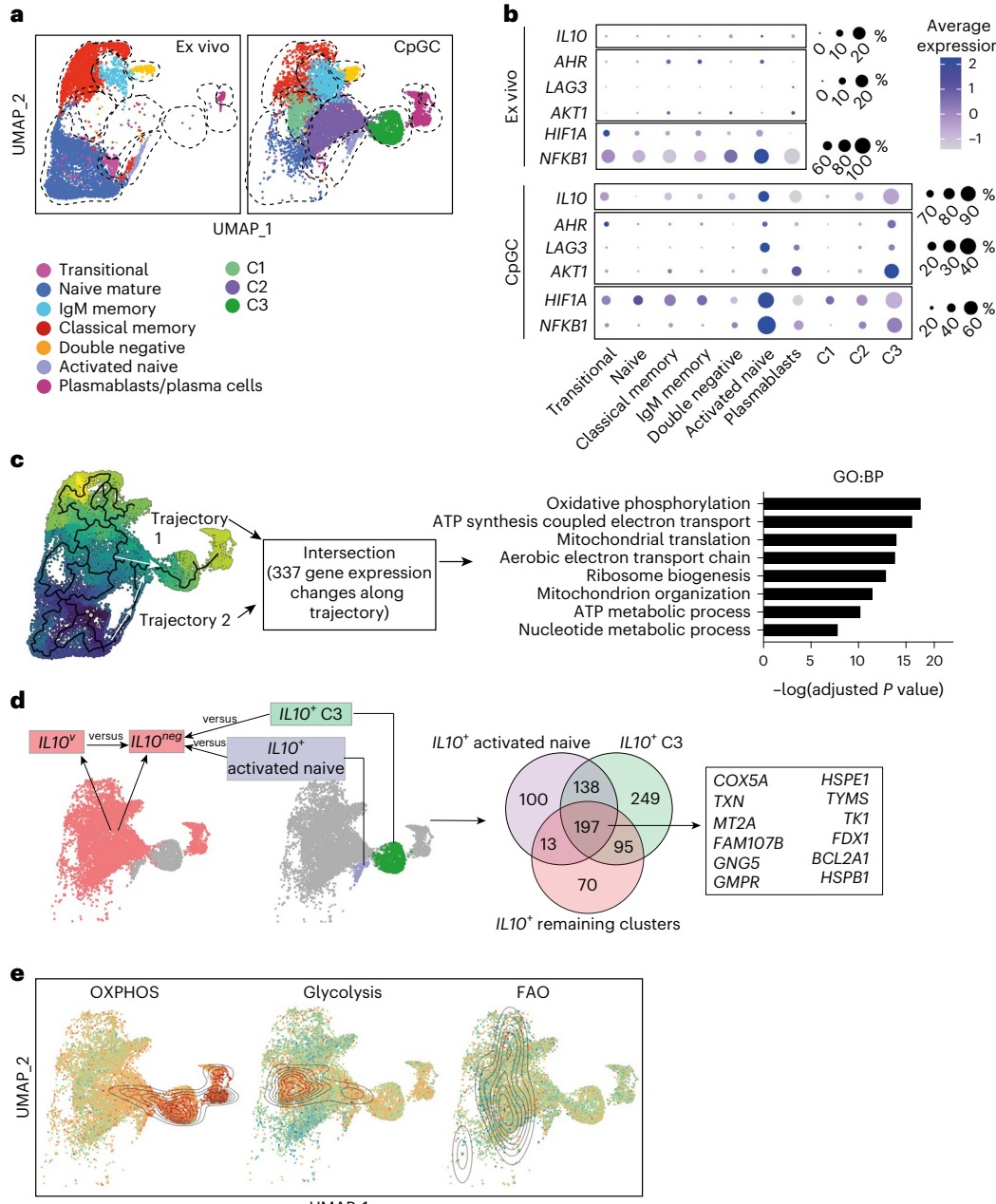

**Fig. 3 | scRNA-seq reveals an enrichment of OXPHOS-associated genes in *IL10*-expressing B cells. a**, UMAPs showing B cell clusters and phenotypic identities ex vivo (left) and following stimulation for 72 h with CpGC (right). **b**, Dot plots showing the expression of indicated genes in each subset ex vivo and following CpGC stimulation. **c**, UMAP and graph showing analysis of genes changing expression along the inferred trajectories through cluster C2 and activated naive into C3 and enriched biological processes. **d**, UMAP and Venn diagram show common genes upregulated in *IL10*⁺ C3 and *IL10*⁺ activated naive

B cells compared with *IL10*⁻ B cells in remaining clusters (including transitional, naive, IgM memory, classical memory, double-negative and plasma cells), and genes upregulated in *IL10*⁺ B cells within these remaining clusters compared with the *IL10*⁻ counterpart. **e**, UMAPs showing B cells scored according to the expression of panels of genes associated with OXPHOS, glycolysis and FAO. Red and high contour density indicate cells with high scores. GO:BP, Gene Ontology:Biological Process.

survival). Only the addition of low levels of $H_2O_2$ (200 nM) increased IL-10⁺ $B_{reg}$ cell frequencies compared with 0.1 µM CpGC alone, whereas the combination of 1 µM CpGC with both low and high $H_2O_2$ levels (10 µM) significantly reduced IL-10⁺ $B_{reg}$ cell frequencies (Fig. 2d).

## Single-cell RNA sequencing confirms IL-10⁺ $B_{reg}$ cell metabolism

To confirm that all IL-10⁺ $B_{reg}$ cells favor OXPHOS over other metabolic pathways, we performed single-cell RNA sequencing (scRNA-seq) on ex vivo and in vitro CpGC-stimulated isolated B cells and looked for

the expression of metabolism-related genes by *IL10*-expressing B cells. Dimensional reduction using the Seurat package revealed seven distinct clusters ex vivo, mainly based on the expression of genes marking B cell maturation stages as previously shown[25] (Fig. 3a and Supplementary Fig. 3a). Clusters identified ex vivo and after CpGC stimulation included transitional and naive ($CD24^+CD38^+CD27^-IGHD^+IGHM^+$), memory ($CD27^+CD24^+CD38^{low}IGHM^+/IGHG1/IGHA2$) and double-negative B cells ($IGHD^-CD27^-ITGAX^+FCRL5^+$). Plasmablasts/plasma cells were readily identifiable by the expression of *PRDM1*, which expanded after CpGC stimulation (Supplementary Fig. 3a).

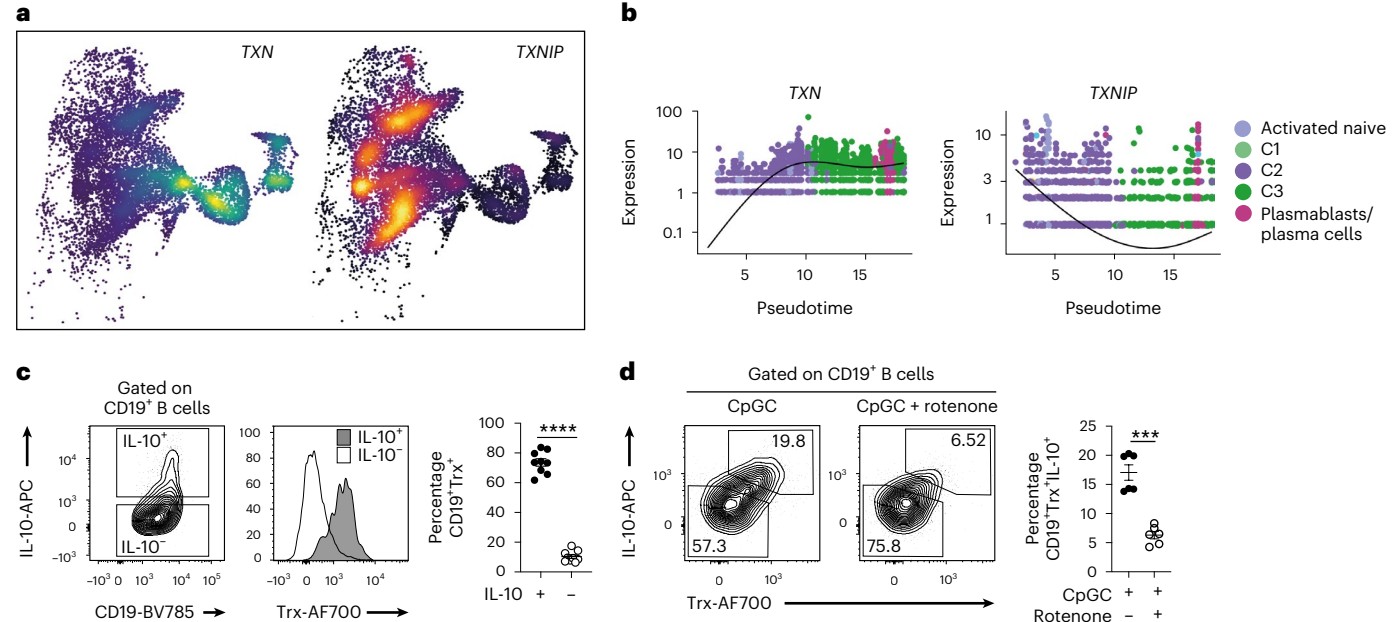

**Fig. 4 | Thioredoxin expression is enriched in IL-10⁺ B_reg cells and requires OXPHOS. a**, UMAP plots show the distribution of *TXN* and *TXNIP*⁺ B cells. **b**, Pseudotemporal gene expression profiles showing the expression levels of *TXN* and *TXNIP* with pseudotime within the trajectories seeding C3. **c**, Representative contour plot, histogram and cumulative data show IL-10 expression following 72-h stimulation with CpGC, and Trx expression and frequencies of CD19⁺Trx⁺ B cells within IL-10⁺ and IL-10⁻ B cell subsets.

*n* = 9 biologically independent samples examined over three independent experiments. ****$P$ < 0.0001. **d**, Representative contour plots and cumulative data show frequencies of CD19⁺Trx⁺IL-10⁺ B cells following stimulation with CpGC for 72 h with and without 1 μM rotenone. *n* = 6 biologically independent samples examined over two independent experiments. ***$P$ = 0.001. Data were analyzed by two-tailed paired *t*-test. Error bars are shown as mean ± s.e.m.

We found that 1.9% of total unstimulated B cells expressed *IL10*, and *IL10*⁺ B_reg cell distribution was across different clusters and independent of maturation stage, thus confirming the heterogeneity of B_reg cells previously described by flow cytometry (Fig. 3b). We identified ten clusters of B cells following CpGC activation which expressed differing levels of *IL10* (Fig. 3a,b). In response to CpGC, we reported the differentiation of three expanded clusters which were scarcely present ex vivo (C1, C2, C3), containing, respectively, 61%, 75% and 97% *IL10*-expressing B cells (Fig. 3a,b). C1, C2 and C3 expressed *CD24*, *CD38*, *IGHM* and *IGHD*, *CD27*⁺ and *TNFRSF13B* (encoding TACI), suggesting that these clusters have derived from several B cell subsets (Supplementary Fig. 3a). We observed the expansion of a population of naive B cells expressing markers of activation (activated naive B cells) which was 82% *IL10*⁺ (Extended Data Fig. 5a). B cells in C3 and in the activated naive cluster expressed higher levels of *AHR*, *LAG3*, *TGFB1*, *AKT1*, *HIF1A* and *NFKB1*, compared with the other clusters, further confirming their B_reg cell identity (Fig. 3b). In addition to *IL10*, B cells in the activated naive cluster following CpGC stimulation expressed higher levels of *TNF* and *IL6*, and *CD274* (encoding PD-L1), compared with other clusters (Extended Data Fig. 5a). CpGC stimulation induced the expansion of five plasmablast clusters, four of which expressed *IL10*⁺ (Extended Data Fig. 5b).

Trajectory analysis inferred multiple pathways seeding into C2, and subsequently into a single path leading to C3, and into plasmablasts/plasma cells. The activated naive B cell cluster formed a single trajectory leading to C3 (Fig. 3c). We compared the top 1,000 genes changing expression along each of these inferred trajectories which highlighted a common signature of 377 genes. Pathway analysis revealed an enrichment of processes primarily associated with mitochondrial respiration, supporting the high mitochondrial metabolic requirement of B_reg cells (Fig. 3c).

To identify a core gene signature expressed in all of the B_reg cells populating the different clusters, we first compiled a list of genes upregulated in C3 and activated naive clusters versus *IL10*⁻ B cells, genes that were also upregulated in *IL10*⁺ B cells in the other clusters

compared with their *IL10*⁻ counterparts. This revealed a signature of 197 genes commonly expressed in C3, in activated naive clusters and in *IL10*⁺ B_reg cells populating the other clusters. These common genes included those previously associated with OXPHOS and the antioxidant response (including *COX5A*, *TXN*, *FDX1*, *TYMS*) (Fig. 3d; gene signature is reported in Supplementary List 1).

Next, we scored each cell based on the expression of panels of genes associated with OXPHOS, glycolysis and FAO. High OXPHOS gene scores were enriched within the C2 and activated naive, C3, and plasmablasts. High glycolysis scores were enriched in the C1 cluster (Fig. 3e). These results confirm distinct metabolic programming in B_reg cells compared with non-B_reg cells, with *IL10*⁺ B_reg cells enriched for gene signatures associated with mitochondrial respiration and the antioxidant response.

**Trx favors IL-10⁺ B_reg and restrains effector B cell differentiation**

To identify potential gene pathways involved in the regulation of OXPHOS and the maintenance of homeostatic levels of ROS needed for B_reg cell differentiation and function, we searched the B_reg cell signature of 217 genes for those that have been previously shown to regulate the generation of ROS from the ETC. Among the most differentially expressed genes, we identified an upregulation of the antioxidant oxidoreductase thioredoxin (*TXN*) encoding Trx, a molecule present in the nuclei, mitochondrial membrane and cytoplasm, in the *IL10*⁺ B_reg cell populations, mirrored by an upregulation of its inhibitor, thioredoxin-interacting protein (*TXNIP*), in the *IL10*⁻ population (Fig. 4a).

*TXN* expression progressively increased, while *TXNIP* decreased, along the pseudotime axis underlying the inferred trajectory, and *TXN* expression was concentrated within C3 and plasmablasts while *TXNIP* was reciprocally expressed in all remaining clusters (Fig. 4b). This progressive upregulation of *TXN* was not observed in the remaining inferred trajectories seeding classical memory, IgM memory (which includes cells in C1) and double-negative B cell clusters

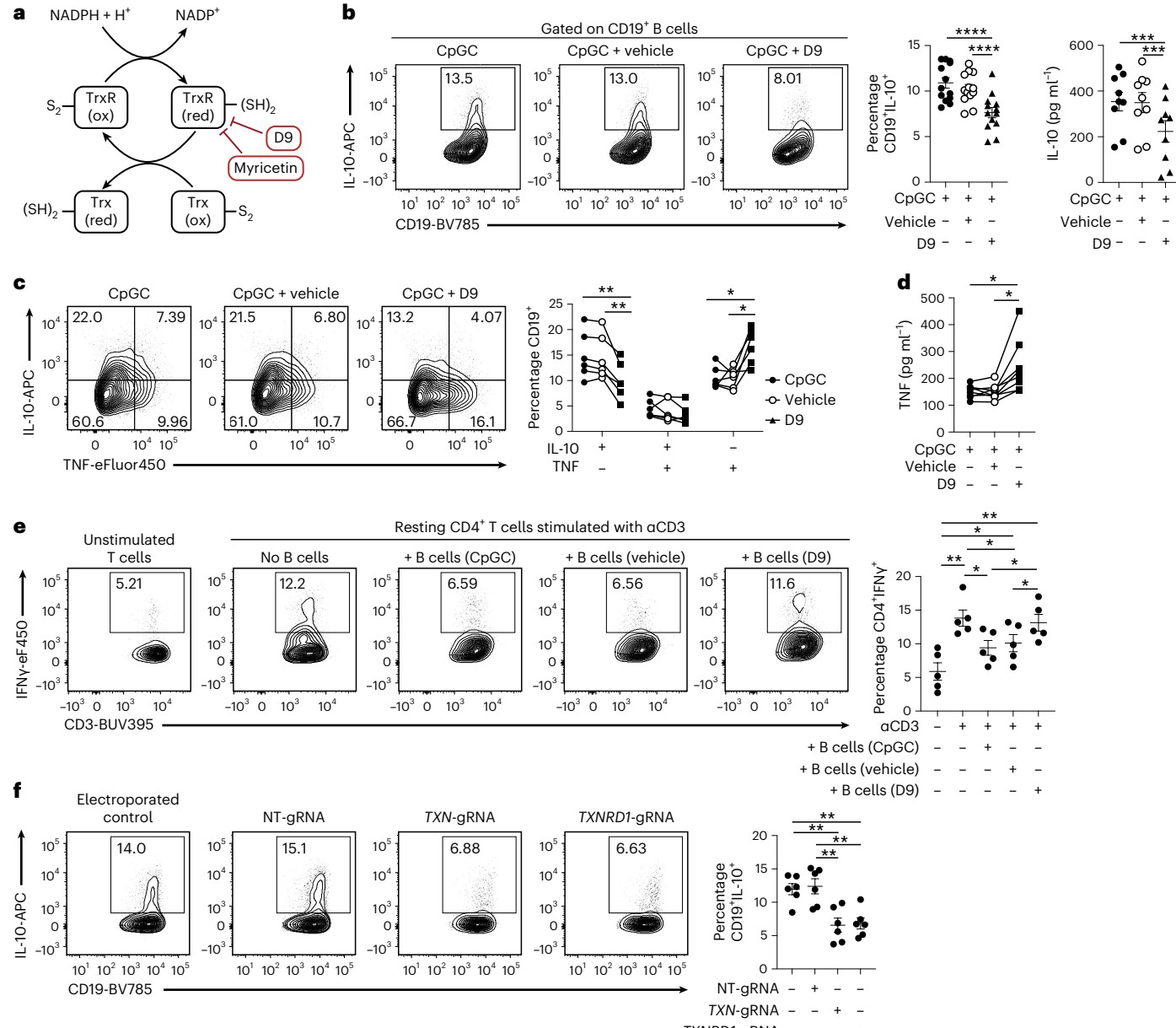

**Fig. 5 | Trx is required for B_reg cell expansion and suppressive ability.**
**a**, Schematic of the Trx system with the target of D9 and myricetin indicated.
**b**, Representative contour plots and cumulative data show frequencies of
CD19⁺IL-10⁺ B cells (*n* = 13 biologically independent samples examined over four
independent experiments) and IL-10 secretion (*n* = 9 biologically independent
samples examined over three independent experiments) following culture
of B cells with CpGC for 72 h, with and without vehicle control or 100 nM D9.
****$P$ < 0.0001, ****$P$ < 0.0001, ***$P$ = 0.0005, ***$P$ = 0.0007. **c**, Representative
contour plots and graph show frequencies of CD19⁺IL-10⁺TNF⁻, CD19⁺IL-10⁺TNF⁺
and CD19⁺IL-10⁻TNF⁺ B cells after 72-h culture of isolated B cells with CpGC,
with and without D9 or vehicle control. *n* = 6 biologically independent samples
examined over three independent experiments. Left to right, **$P$ = 0.0062,
**$P$ = 0.0083, *$P$ = 0.0237, *$P$ = 0.0172. **d**, Graph showing the levels of secreted
TNF by isolated B cells stimulated for 72 h with CpGC, with and without D9 or
vehicle control. *n* = 8 biologically independent samples examined over three

experiments. *$P$ = 0.0330. **e**, Representative contour plots and cumulative data
show the frequencies of CD3⁺CD4⁺IFNγ⁺ T cells after co-culture of resting CD4⁺
T cells stimulated with anti-CD3, with B cells preconditioned with CpGC, CpGC
and D9, or CpGC and vehicle control. *n* = 5 biologically independent samples
examined over three independent experiments. Top to bottom, **$P$ = 0.001,
*$P$ = 0.0359, *$P$ = 0.0447, **$P$ = 0.0071, *$P$ = 0.0191, *$P$ = 0.0264, *$P$ = 0.0325.
**f**, Representative contour plots and cumulative data showing frequencies of
CD19⁺IL-10⁺ B cells after CRISPR–Cas9 silencing of *TXN* and *TXNRD1* and culture
for 72 h with CpGC. *n* = 5 biologically independent samples examined over
three independent experiments. Top to bottom, **$P$ = 0.0068, **$P$ = 0.0044,
**$P$ = 0.0034, **$P$ = 0.0022. Data were analyzed by one-way ANOVA followed
by Tukey's multiple comparisons test (**b**,**d**,**e**,**f**) or two-way ANOVA followed by
Sidak's test for multiple comparisons (**c**). Error bars are shown as mean ± s.e.m.
Panel **a** created with Biorender.com.

(Extended Data Fig. 5c). By flow cytometry we confirm that IL-10⁺ B_reg
cells contain higher frequencies of Trx⁺ B cells compared with the
IL-10⁻ B cell population (Fig. 4c). Stimulation with CpGC, CD40L or
TLR7 ligand R848 alone, or CpGC in combination with IFNα, BAFF
or anti-BCR, signals pivotal in B_reg cell differentiation[26], increased

Trx⁺IL-10⁺ B_reg cell frequencies (Extended Data Fig. 5d). Although the
expression of mitochondrial thioredoxin *TXN2*, among different clus-
ters, to a certain extent mirrored that of *TXN* (Extended Data Fig. 5e), at
the protein level there were no differences in the frequencies of Trx2⁺
cells between IL-10⁺ and IL-10⁻ B cell subsets (Extended Data Fig. 5f,g).

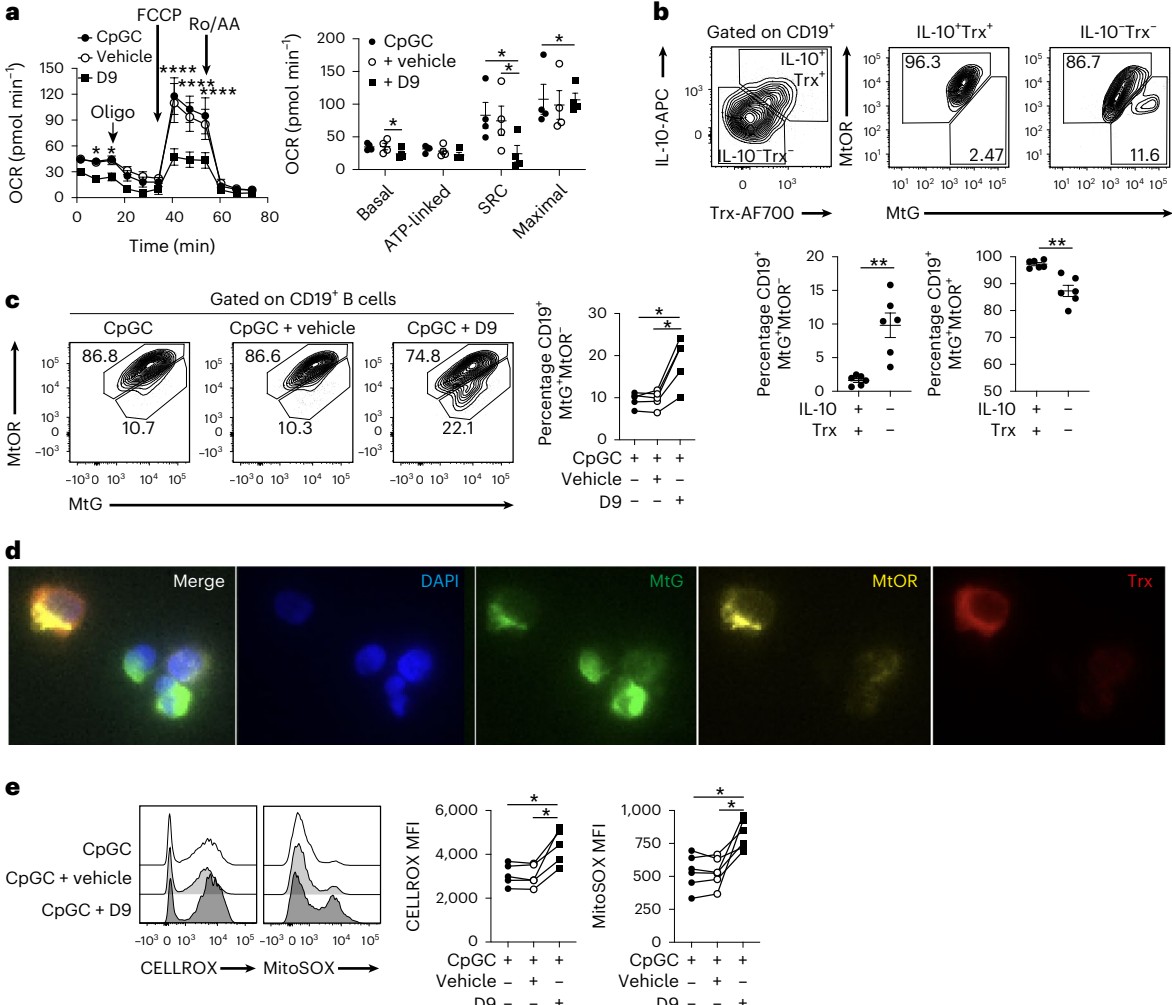

**Fig. 6 | Trx maintains mitochondrial health and low ROS levels. a**, Graphs show real-time Seahorse analysis and basal, ATP-linked, maximal respiration and spare respiratory capacity (SRC) of B cells following 72-h stimulation with CpGC, with and without vehicle control or 100 nM D9. $n$ = 4 biologically independent samples examined over two independent experiments. Left to right, *$P$ = 0.0115, *$P$ = 0.0213, ****$P$ < 0.0001, ****$P$ < 0.0001, ****$P$ < 0.0001, *$P$ = 0.0328, *$P$ = 0.0123, *$P$ = 0.0382, *$P$ = 0.0356. **b**, Representative contour plots stained with MitoTracker Green (MtG; to assess mitochondrial mass) and MitoTracker Orange (MtOR; its accumulation is dependent upon membrane potential) and graphs show frequencies of B cells with polarized (MtG$^+$MtOR$^+$) and depolarized (MtG$^+$MtOR$^-$) mitochondrial membranes within IL-10$^+$Trx$^+$ and IL-10$^-$Trx$^-$ B cell subsets. $n$ = 6 biologically independent samples examined over two independent experiments. Left to right, **$P$ = 0.0035, **$P$ = 0.0021. **c**, Representative contour plots and graph show frequencies of B cells with polarized (CD19$^+$MtG$^+$MtOR$^+$)

and depolarized (CD19$^+$MtG$^+$MtOR$^-$) mitochondrial membranes following 24-h stimulation of B cells with CpGC, with and without vehicle control or 100 nM D9. $n$ = 5 biologically independent samples examined over two independent experiments. Top to bottom, *$P$ = 0.0164, *$P$ = 0.0117. **d**, Confocal images show expression of Trx (red) alongside MtG (green) and MtOR (yellow) staining in B cells stimulated for 72 h with CpGC. **e**, Representative histograms and cumulative data show levels of cytoplasmic (CELLROX) and mitochondrial (MITOSOX) ROS after 72-h stimulation of isolated B cells with CpGC, with and without D9 or vehicle control. $n$ = 6 biologically independent samples examined over two independent experiments. Left to right, *$P$ = 0.0148, *$P$ = 0.0138, *$P$ = 0.04, *$P$ = 0.0250. Data were analyzed by two-way ANOVA followed by Sidak's test for multiple comparisons (**a**), two-sided paired $t$-test (**b**) or one-way ANOVA followed by Tukey's multiple comparisons test (**c**,**e**). Error bars are shown as mean ± s.e.m. Ro/AA, rotenone/antimycin A.

To assess whether the ETC regulates Trx$^+$IL-10$^+$ B$_{reg}$ cell frequencies, we cultured B cells with CpGC alone or with rotenone. Trx$^+$IL-10$^+$ B$_{reg}$ cell frequencies were significantly decreased following stimulation of B cells with rotenone (Fig. 4d), but not when co-cultured with inhibitors of glycolysis (2-DG), FAO (etomoxir) or glutamine metabolism (BPTES) (Extended Data Fig. 5h), demonstrating that Trx$^+$IL-10$^+$ B$_{reg}$ cells depend on OXPHOS.

Next, we tested whether Trx activity was necessary for the differentiation of IL-10$^+$ B$_{reg}$ cells. B cells were cultured with CpGC and TrxR inhibitors D9 and myricetin (schematic in Fig. 5a). Addition of D9 or myricetin to CpGC-stimulated B cells significantly reduced IL-10$^+$ B$_{reg}$ cell frequencies and IL-10 secretion compared with B cells stimulated with CpGC alone or CpGC with vehicle control (Fig. 5b and

Extended Data Fig. 6a). Neither D9 nor myricetin altered B cell viability, and D9 did not alter B cell proliferation (Extended Data Fig. 6b,c).

In addition to IL-10, inhibition of Trx by D9 reduced, respectively, the TGFβ$^+$, PD-L1$^+$ and IL-35$^+$ B cell frequencies, cytokines and immune-checkpoint inhibitors implicated in the regulatory function of B cells[2,27], compared with B cells stimulated with CpGC and vehicle control (Extended Data Fig. 6d,f). The decrease in immune-regulatory cytokines following Trx inhibition was mirrored by increased CD19$^+$IL-10$^-$TNF$^+$ B cell frequencies and an increase in the overall TNF and IL-6 secretion (Fig. 5c,d and Extended Data Fig. 6g,h). Trx-regulated IL-10 expression was also reduced by D9 following a range of B cell stimuli, including CpGC + IFNα, CD40L and anti-BCR (Extended Data Fig. 6i).

## Table 1 | Demographic and clinical characteristics

| | Healthy (n=31) | SLE (n=58) |
|---|---|---|
| Age (average) | 36.1 | 43.8 |
| Age (range) | 21–60 | 21–84 |
| Sex (F:M) | 23:8 | 56:2 |
| Sex % (F:M) | 74:26 | 96.5:3.5 |
| Ethnicity (C/AC/SA/EA) | 19/5/5/2 | 30/14/8/6 |
| Ethnicity % (C/AC/SA/EA) | 61.2/16.1/16.1/6.5 | 51.7/24.1/13.8/10.3 |
| Treatment (%) HCQ/Pred/MTX/MMF/Aza | | 55.1/63.8/8.6/18.9/17.2 |

Patients fulfilling the revised classification criteria for SLE were assessed for disease activity with the BILAG index. The BILAG index is a clinical measure of disease that distinguishes activity in nine different organ systems. Patients with a GS equal to or greater than 5 were considered active. The following abbreviations are used: female:male (F:M), African-Caribbean (AC), Caucasian (C), South Asian (SA), East Asian (EA), hydroxychloroquine (HCQ), prednisolone (Pred), methotrexate (MTX), mycophenolate mofetil (MMF), azathioprine (Aza).

We assessed the effect that Trx inhibition has on different IL-10[+] and IL-10[−] B cell subsets. Unlike rotenone, which blocked OXPHOS and inhibited both B$_{reg}$ cell and plasmablast differentiation, Trx inhibition reduced CD24$^{hi}$CD38$^{hi}$ B cells and increased CD24$^{lo/−}$CD38$^{hi}$Blimp1$^{+}$ plasmablast frequencies (Extended Data Fig. 7a,b). Inhibition of Trx function induced a significant decrease in IL-10[+] B$_{reg}$ cell frequencies within the CD24$^{hi}$CD38$^{hi}$ immature, CD24$^{int}$CD38$^{int}$ mature-naive and CD24$^{+}$CD38$^{lo}$ memory B cell and CD24$^{lo/−}$CD38$^{hi}$Blimp1$^{+}$ plasmablast populations (Extended Data Fig. 7c).

To measure if B$_{reg}$ cells, in addition to producing IL-10, depend on active Trx to achieve their suppressive function, we co-cultured anti-CD3-stimulated CD4$^{+}$ T cells with B cells cultured for 48 h with CpGC, with and without D9 or vehicle control. CD4$^{+}$ T cells cultured in the presence of D9-conditioned B cells displayed increased IFNγ expression compared with T cells cultured in the presence of B cells stimulated with CpGC alone, or B cells stimulated with CpGC and vehicle control (Fig. 5e).

Although there are no known off-site effects of D9 due to the specificity of its target (TrxR), to further validate the unique role of Trx in B$_{reg}$ cell differentiation, we selectively silenced *TXN* and *TXNRD1*, as well as *TXN2* and *TXNRD2* as negative control. Comparable to the results obtained following the pharmacological inhibition of Trx with D9, silencing of *TXN* and *TXNRD1* inhibits B$_{reg}$ cell differentiation whereas B cells continue to express IL-10 despite *TXN2* or *TXNRD2* silencing (Fig. 5f and Extended Data Fig. 7d,f).

### Trx maintains low ROS and high mitochondrial polarization in B cells

We next assessed B cell cellular bioenergetic profiles and found that in response to Trx inhibition, the O$_2$ consumption rate (OCR), an indicator of OXPHOS rate, was significantly lower when compared with B cells stimulated with CpGC alone or with CpGC and vehicle control, whereas the basal extracellular acidification rate, a consequence of lactic acid production and an indicator of glycolysis rates, was unaffected (Fig. 6a and Extended Data Fig. 8a). Inhibition of Trx reduced mitochondrial spare respiratory capacity when compared with CpGC- or CpGC and vehicle-treated B cells, as indicated by the difference between the maximal OCR (after stimulation with the uncoupler FCCP (carbonyl cyanide-p-trifluoromethoxyphenylhydrazone)) and basal OCR (Fig. 6a).

Because Trx has been shown to maintain mitochondrial structure, biogenesis and bioenergetics[28] we posited that B$_{reg}$ cells, in virtue of expressing elevated Trx and relying on ETC activity, present with highly polarized mitochondria. We found that Trx$^{+}$IL-10[+] B$_{reg}$ cells have highly polarized mitochondrial membranes (Fig. 6b) and that Trx inhibition with D9 or by gene silencing led to an accumulation of B cells with depolarized mitochondria (Fig. 6c and Extended Data Fig. 8b) and a loss of IL-10 expression (Fig. 5b). Under these culture conditions we showed that Trx is co-localized with polarized mitochondria (Fig. 6d and Extended Data Fig. 8c). Inhibition of Trx by D9 or *TXN* by gene silencing led to an increase of mitochondrial and, to a lesser extent, cytoplasmic ROS (Fig. 6e and Extended Data Fig. 8d,e). Collectively, these results show that Trx inhibition of ROS is accompanied with optimal mitochondrial polarization and respiration, necessary to meet the metabolic demands of IL-10[+] B$_{reg}$ cells.

### Trx deficiencies underpin IL-10[+] B$_{reg}$ cell dysfunction in SLE

We and others have previously shown that B cells fail to differentiate into B$_{reg}$ cells in patients with SLE[3]. Moreover, mitochondrial dysfunction and increased oxidative stress have been shown to play a role in SLE pathophysiology, but whether these defects contribute to the B$_{reg}$ cell aberrations present in these patients remains unknown[29]. We report that in addition to a decrease in IL-10[+] B$_{reg}$ cells in a new SLE patient cohort (demographics are reported in Table 1) (Fig. 7a), B cells from patients with SLE present with significantly greater mitochondrial membrane depolarization and elevated mitochondrial ROS levels compared with healthy donor B cells ex vivo (Extended Data Fig. 9a) or following CpGC stimulation (Fig. 7b–d). Patients with SLE also displayed significantly lower frequencies of Trx$^{+}$ B cells compared with healthy donors (Fig. 7e). Cumulative analysis of healthy and SLE B cells shows a positive correlation between the frequencies of Trx$^{+}$ and IL-10[+] B$_{reg}$ cells (Fig. 7f), further supporting the role of Trx in IL-10[+] B$_{reg}$ cell differentiation. Mitochondrial membrane depolarization and reduced IL-10[+] B$_{reg}$ cell frequencies were observed in SLE B cells after stimulation with the TLR7 ligand R848, or when cultured with 0.1 µM CpGC (Extended Data Fig. 9b,e). Of note, Trx expression was higher in B cells than in other immune cell subsets in healthy individuals, and in patients with SLE the defect was restricted to B cells, as other cells exhibited the same expression of Trx as healthy donors (Extended Data Fig. 9f).

We observed significantly higher mitochondrial membrane depolarization and significantly reduced Trx expression in both disease-active (British Isles Lupus Assessment Group (BILAG) index global score (GS) equal to or more than 5) and disease-inactive (BILAG GS less than 5) patients with SLE compared with healthy donors. Notably, disease-active patients exhibited significantly lower Trx$^{+}$ B cell frequencies compared with disease-inactive patients (Extended Data Fig. 9g). Of interest, we did not observe any associations between Trx$^{+}$ B cell frequencies or mitochondrial depolarization with treatment regime (Extended Data Fig. 9h,i).

Next, we assessed the impact that restoration of Trx levels has on both mitochondrial fitness and IL-10[+] B$_{reg}$ cell differentiation in SLE. B cells from patients with SLE and healthy donors were stimulated with CpGC alone or with exogenous recombinant human Trx (rhTrx). rhTrx was taken up by B cells as evidenced by detectable intracellular fluorescence-labeled exogenous rhTrx, measured by flow cytometry (Extended Data Fig. 9j). Provision of rhTrx restored mitochondrial membrane polarization and inhibited mitochondrial ROS levels in B cells from patients with SLE to similar levels as in healthy B cells (Fig. 7g,h). Restoration of mitochondrial fitness was accompanied by a significant increase in IL-10[+] B$_{reg}$ cell frequencies and IL-10 secretion compared with controls, confirming that Trx deficiency in these patients is partially responsible for mitochondrial depolarization and impaired IL-10 production (Fig. 7i). We confirmed that in B cells from patients with SLE, rhTrx restored the production of IL-10 by all B cell subsets, including CD24$^{hi}$CD38$^{hi}$ immature, CD24$^{int}$CD38$^{int}$ mature-naive and CD24$^{+}$CD38$^{lo}$ memory B cells and CD24$^{lo/−}$CD38$^{hi}$ plasmablasts (Extended Data Fig. 10a). Finally, we observed a selective expansion of IL-10[+] plasmablasts in both healthy donors and patients with SLE following exposure to rhTrx, in line with the contraction of this subset following Trx inhibition in healthy B cells (Extended Data Fig. 10b). We substantiated the validity of our results in B cells from

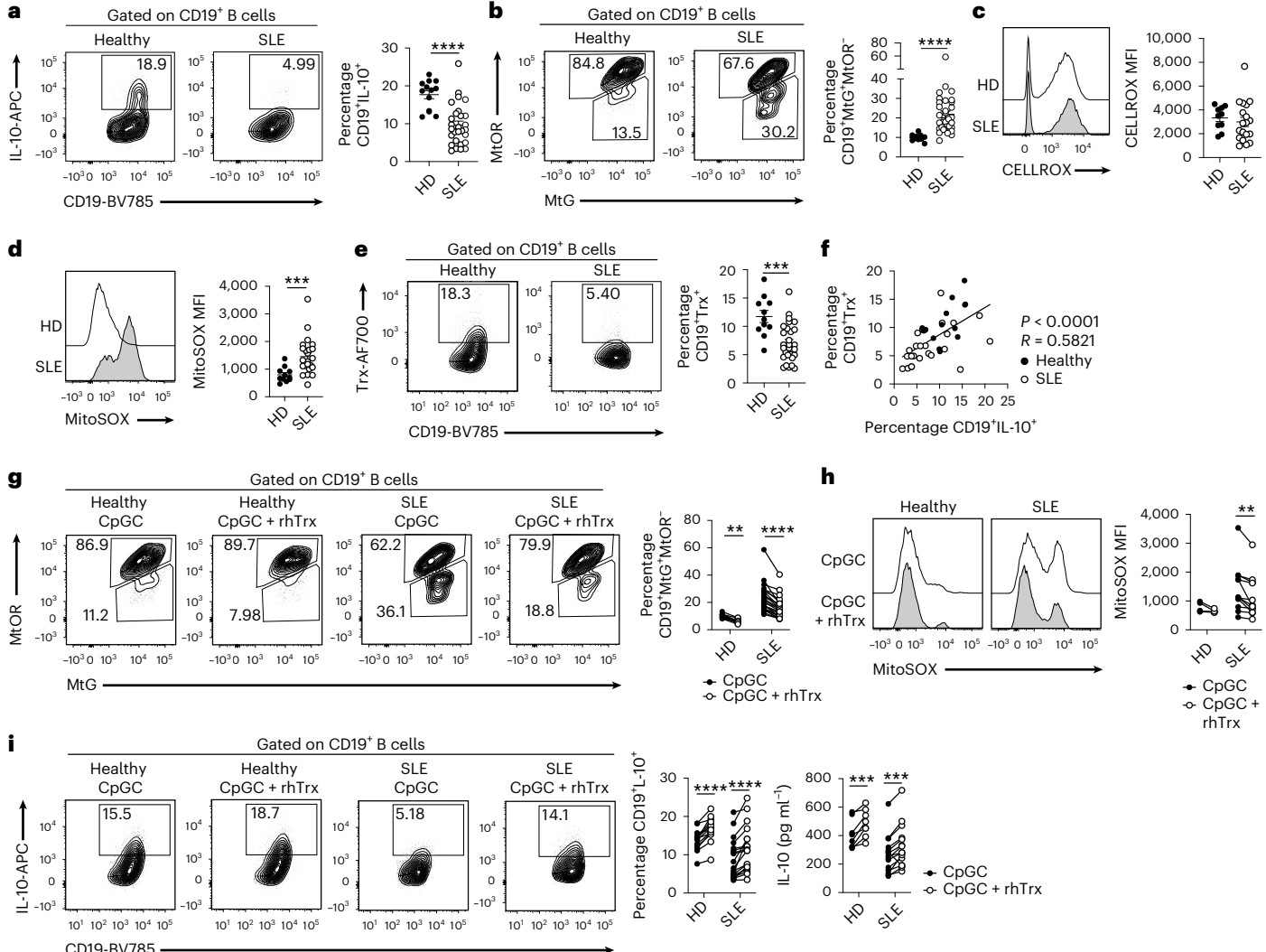

**Fig. 7 | Exogenous Trx rescues B$_{reg}$ cell deficiencies in SLE. a**, Representative contour plots and cumulative data show frequencies of CD19$^+$IL-10$^+$ B cells in healthy donors ($n = 13$) and patients with SLE ($n = 28$) following 72-h stimulation with CpGC. ****$P < 0.0001$. **b**, Representative contour plots and cumulative data show frequencies of B cells with polarized (CD19$^+$MtOR$^+$MtG$^+$) and depolarized mitochondria (CD19$^+$MtOR$^-$MtG$^+$) in healthy controls ($n = 10$) and patients with SLE ($n = 32$) following 24-h stimulation with CpGC. ****$P < 0.0001$. **c**, Representative histograms and cumulative data show the levels of cytoplasmic ROS (CELLROX) in B cells of healthy donors ($n = 10$) and patients with SLE ($n = 19$) after 24-h CpGC stimulation. **d**, Representative histograms and cumulative data show the levels of mitochondrial ROS (MitoSOX) in B cells of healthy donors ($n = 11$) and patients with SLE ($n = 24$) after 24-h CpGC stimulation. ***$P = 0.0003$. **e**, Representative contour plots and cumulative data show frequencies of Trx$^+$ B cells in healthy donors ($n = 11$) and patients with SLE ($n = 28$) following 72-h stimulation with CpGC. ***$P = 0.0004$. **f**, Cumulative data showing correlation between CD19$^+$IL-10$^+$ and CD19$^+$Trx$^+$ B cell frequencies in healthy donors

($n = 15$) and patients with SLE ($n = 26$) following 72-h stimulation with CpGC. ****$P < 0.0001$, $R = 0.5821$. **g**, Representative contour plots and cumulative data show frequencies of B cells with polarized (CD19$^+$MtOR$^+$MtG$^+$) and depolarized mitochondria (CD19$^+$MtOR$^-$MtG$^+$) in healthy donors ($n = 10$) and patients with SLE ($n = 25$) following 24-h stimulation with CpGC with and without 100 μM rhTrx. **$P = 0.0016$, ****$P < 0.0001$. **h**, Representative histograms and cumulative data show levels of mitochondrial ROS in B cells of healthy donors ($n = 5$) and patients with SLE ($n = 12$) after 24-h CpGC stimulation, with and without 100 μM rhTrx. **$P = 0.0019$. **i**, Representative contour plots and cumulative data show frequencies of CD19$^+$IL-10$^+$ B cells (HD $n = 14$, SLE $n = 19$) and IL-10 secretion (HD $n = 11$, SLE $n = 16$) in healthy donors and patients with SLE following 72-h stimulation with CpGC, with and without 100 μM rhTrx. Left to right, ****$P < 0.0001$, ****$P < 0.0001$, ***$P = 0.0007$, ***$P = 0.0006$. Data were analyzed by two-tailed unpaired $t$-test (**a**–**e**), two-tailed Pearson correlation (**f**) or two-way ANOVA followed by Sidak's test for multiple comparisons (**g**–**i**). Error bars are shown as mean ± s.e.m. HD, healthy donor.

lymph nodes (LNs) of patients with SLE and nonautoimmune patients, observing a similar reduction in mitochondrial depolarization and increase in IL-10 levels following exposure to exogenous Trx (Extended Data Fig. 10c,d). There was a trend towards increased B cell mitochondrial depolarization in LNs from patients with SLE compared with those from nonautoimmune individuals. Therefore, the importance of Trx in B cell mitochondrial health and IL-10 production is not restricted to peripheral blood and is a core B cell mechanism independent of tissue localization.

## Discussion

Whereas more is known about the metabolic drivers of regulatory versus effector T cell differentiation, there is still a scarcity of information with respect to the intracellular bioenergetic pathways required for B cell differentiation into B$_{reg}$ cells. B cells are functionally versatile, and their phenotypes and functions change in response to the surrounding environment. Here we show that B$_{reg}$ cells meet their metabolic demands primarily via OXPHOS. These findings match the metabolic requirements of other immunoregulatory cell subsets including regulatory

T cells and M2 macrophages[30,31], suggesting a common role for OXPHOS in driving the differentiation of suppressive immune cell subsets.

Increasing amounts of data suggest that B_{reg} cells differentiate in response to their specific local environments[1]. In this context, we and others have shown that a variety of pro-inflammatory signals, including IL-1β, IL-6 and IFNα, foster the differentiation of B_{reg} cells and plasma cells-producing antibodies[4]. However, it is established that when these inflammatory signals are chronically present, as seen, for example, in autoimmune patients, they lead to a significant reduction in B_{reg} cells and an increase in effector B cells[4]. Here we propose that Trx down-regulates ROS to 'physiological' levels required for the induction of B_{reg} cells with concurrent suppression of effector B cells. Supporting this are our findings showing that Trx inactivation in B cells, cultured under any of the B_{reg} cell polarizing conditions tested, impairs their expression of IL-10 and suppressive activity, resulting in an increased number of B cells producing pro-inflammatory cytokines.

Although Trx has been identified and studied for its function in the cytosol, it has also been found in the nuclei and in the mitochondrial intermembrane space where it acts as a functional barrier regulating the movement of small molecules between the cytosol and the matrix[32]. In our study, we have established a strong link between Trx activity in maintaining mitochondrial polarization and in inhibiting both mitochondrial and, to a lesser extent, cytoplasmic ROS in B cells. We have demonstrated that Trx, but not the mitochondrial-exclusive isoform Trx2, plays a pivotal role in regulating B_{reg} cell differentiation. These results were somewhat unexpected, considering the conserved structure between the two Trx isoforms. However, our findings align with previous data indicating that Trx in a human smooth muscle cell line prevents mitochondria-related apoptotic changes induced by As2O3 (ref. 33).

While our focus has been on the antioxidant properties of Trx in maintaining B_{reg} cell metabolism, Trx plays a pleiotropic role in the immune system, including promoting cell growth, modulating pro-inflammatory cytokines and regulating apoptosis[12]. In addition, it is well established that Trx controls protein structure and function, promoting the binding of transcription factors to target DNA by reducing cysteine residues present on these proteins[12]. In this context, cysteine residues targeted by Trx are present in Cyp1a1 (a proxy for Ahr), NF-κB, AP-1 and HIF-1α−transcription factors previously shown to regulate IL-10 expression[34]. Several of these factors, including HIF-1α, have been identified in the mitochondria[35,36]. Taken together, it is possible that, in addition to regulating the amounts of both cytoplasmic and mitochondrial ROS, Trx controls protein functions in the mitochondria, which consequently exert their effects in the nucleus[37].

B cell differentiation into plasmablasts requires mitochondrial membrane polarization to maximize the outputs of OXPHOS[38]. Our results also show that the same pathway is required for the differentiation of IL-10+ B_{reg} cells. To add a degree of complexity we have now shown that although inhibition of Trx in B cells impairs ETC function, the IL-10− plasmablast frequencies were unaffected by these metabolic changes. We propose that the decrease in mitochondrial membrane polarization observed in B cells following Trx inhibition triggers supplementary metabolic pathways in IL-10− plasmablasts. One potential pathway is glutamine metabolism, which we have shown to be essential for both B_{reg} cell and plasmablast differentiation (Extended Data Fig. 3a,e,f). Glutamine metabolism drives mTORC1 activity, thereby promoting plasma cell differentiation and antibody secretion[39,40].

Alternatively, it has been previously shown that mice deficient in the antioxidant Tumor Protein 53-Induced Nuclear Protein 1 (TP53INP1) in the presence of chronic oxidative stress display elevated plasma cell frequencies with enhanced antibody secretion[41], suggesting that increased oxidative stress may expedite plasma cell differentiation. Future investigations will aim to address this hypothesis.

Our findings show that patients with SLE present with Trx deficiencies restricted to B cells but not to other cells, and that addition of exogenous Trx restores IL-10+ B_{reg} cells and their mitochondrial membrane polarization to healthy levels. The cause of Trx impaired expression in SLE B cells, which we anticipate being multi-factorial, remains to be addressed. TXN single-nucleotide polymorphisms (SNPs) were not found in three separate genome-wide association studies and, given the low allele frequency of these SNPs, it is unlikely the reduction in Trx we observe is due to any given TXN SNP[42–44]. Due to the scarcity of information regarding factors inducing Trx, it is challenging to find if there is any link to known risk alleles for SLE. The Keap1-Nrf2 pathway is the principal protective response to oxidative stress, and Nrf2 (nuclear factor (erythroid-derived 2)-like 2) is one of the few transcription factors known to induce TXN expression[45]. In our scRNA-seq dataset, we show that KEAP1 expression is restricted to B_{reg}-enriched clusters. Similarly, NFE2L2 (encoding NRF2) expression is concentrated within B_{reg}-enriched clusters C2 and C3 (Supplementary Fig. 4). Both mouse models and genome-wide scans have identified Nrf2 as a candidate gene for susceptibility to SLE. Interestingly, aged female Nrf2-deficient mice are prone to developing an autoimmune condition closely resembling human SLE[46]. As Nrf2 is also linked to mitochondrial health[47], one could argue that the defect in Nrf2 expression contributes to the observed mitochondrial defects in SLE. Future studies will address if the defect in Nrf2 expression contributes to the observed mitochondrial defects in SLE.

It is known that patients with SLE have defects in Complex I and Complex III subunits and increased levels of mitochondrial ROS[48]. One might speculate about an environmental event that, together with some genetic predisposition, initially increases ROS in SLE B cells, leading to a vicious cascade that increases DNA/histone methylation, interferes with TXN expression and consequently results in reduced IL-10 expression.

The finding that Trx is pivotal in the maintenance of OXPHOS in B_{reg} cells and is an important regulator of their differentiation opens several new therapeutic opportunities. Promisingly, treatment of mice with rhTrx can suppress autoimmune myocarditis and diabetes, and in diabetes suppression of disease is achieved by transfer of lymphocytes isolated from Trx-treated mice[49,50]. Conversely, inhibition of Trx may prove to be a useful therapy for cancer as intratumoral infiltration of B_{reg} cells in solid cancer is associated with poor prognosis[51]. Taken together, Trx-dependent pathways may represent a novel targeted therapeutic approach to restore or inhibit B_{reg} activity in autoimmune diseases versus cancer.

## Online content

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

## Methods

### Study population

Peripheral blood samples were collected from healthy donors and from patients with SLE attending the University College London Hospital (UCLH) rheumatology outpatient clinic. Ethical approval was obtained from the UCLH Health Service Trust ethics committee, under Research Ethics Committee reference no. 14/SC/1200. Patients and healthy donors were recruited following informed consent. Participants did not receive compensation. Sample storage complied with requirements of the Data Protection Act 1998. LNs were collected from patients with SLE and nonautoimmune individuals undergoing kidney transplantation at the Royal Free Hospital, Hampstead, UK.

All patients recruited met at least four of the 11 American College of Rheumatology classification criteria, with a disease duration of more than 6 months, and positivity for antinuclear antibody or anti-double-stranded DNA autoantibodies. Patients under the age of 18, treated with B cell-depleting therapies (including rituximab, belimumab), participating in any interventional trial or who were pregnant were excluded from the study. Also excluded were patients with severe central nervous system lupus, glomerulonephritis or congestive heart failure, and patients with a history of infections including HIV, hepatitis B/C and tuberculosis. Patients also meeting criteria for other autoimmune diseases (such as multiple sclerosis, rheumatoid arthritis) were excluded. SLE disease activity was assessed by the BILAG index, a standardized disease activity assessment. Activity in each organ system was given a score of A–E, where A is most active and E is never active. Each score was numerically converted (A = 9, B = 3, C = 1, D = 0, E = 0). The sum of these values is represented as the global BILAG score. Low lupus disease activity was defined as a BILAG GS ≤ 5. Demographics, clinical characteristics, routine laboratory testing and therapeutic regimen (reported in Table 1) were collected from electronical medical files of the visit to the clinic recorded on the day blood was drawn.

### Cell isolation and culture

Peripheral blood (50 ml) from individual patients and healthy donors was collected in heparinized tubes. PBMCs were isolated from whole blood by Ficoll-based density centrifugation. LNs were chopped and mashed through a 70 µm cell strainer to obtain single-cell suspensions. Leukocytes from LNs were obtained by Ficoll-based density centrifugation.

B cells were isolated from PBMCs and LNs using the EasySep immunomagnetic negative selection kit (STEMCELL, cat. no. 19054). Isolated B cells were cultured for 72 h with CpGC ODN 2395 (1 µM; Invivogen, cat. no. tlrl-2395-1) in RPMI 1640 containing L-glutamine (Sigma-Aldrich). Media were supplemented with 10% FCS (LabTech) and 1% penicillin/streptomycin (100 U ml$^{-1}$ penicillin + 100 µg ml$^{-1}$ streptomycin; Sigma-Aldrich). Where applicable, cells were stimulated with recombinant human IFNα (1,000 U ml$^{-1}$; PBL, 11200-1), mega-CD40L (1 µg ml$^{-1}$; Enzo, cat. no. ALZ-522-110-C010), anti-IgM/G/A (10 µg ml$^{-1}$; Sigma-Aldrich, cat. no. AQ503), R848 TLR7L (1 µM; Invivogen, cat. no. tlrl-r848), recombinant human BAFF (10 µM; Peprotech, 310-13), LPS (1 µM; Invivogen, cat. no. tlrl-b5lps) or recombinant human IL-21 (20 nM; Peprotech, cat. no. 200-21). Cells were cultured at 37 °C and 5% CO$_2$.

Supernatants from cell cultures were collected and analyzed for IL-10 secretion using a standard sandwich IL-10 ELISA kit (R&D Systems, cat. no. DY217B-05) performed according to the manufacturer's instructions. TNF and IL-6 secretion were measured using a human inflammatory cytokine cytometric bead assay (BD, cat. no. 551811).

### Metabolism and Trx assays

For metabolism inhibition assays, B cells were incubated with 1 mM 2-DG, 10 µM etomoxir, 1 µM rotenone, 10 nM antimycin A (Abcam; ab254445, ab142242, ab143145, ab141904), 500 nM BPTES (Tocris, 5301) or 100 µM Malonyl coenzyme A (Sigma-Aldrich, M4263).

B cells were alternatively cultured in glucose- or glutamine-free RPMI 1640 (ThermoFisher Scientific, cat. nos. 11879020, 21870076), with replenished glucose or L-glutamine (Sigma-Aldrich, cat. nos. G8270, G7513), or with galactose (Sigma-Aldrich, cat. no. G0750). For measurement of mitochondrial polarization, MitoTrackers Green FM and Orange CMTMRos (ThermoFisher Scientific, M7510, M7514) were used at 100 nM and 200 nM, respectively. Trx assays were performed using the TrxR inhibitors D9 (100 nM; Diphenyl-2-thienylphosphine-kP [2-(4-methoxyphenyl) ethynyl]gold, Tocris, 5921) and myricetin (10 µM; Tocris, 6189) or an equivalently diluted dimethylsulfoxide vehicle. Glutathione synthetase activity was inhibited using 500 µM L-buthione-sulfoximine (Sigma-Aldrich, cat no. B2515). rhTrx was purchased from R&D Systems (190-TX-500) and used at 100 µM. ROS were scavenged using NAC (Abcam, ab143032) or MitoTempo (Sigma-Aldrich, cat. no. SML0737). Cytoplasmic ROS levels were measured using CELLROX Orange Reagent at 750 nM (ThermoFisher Scientific, cat. no. C10443), and mitochondrial superoxide levels were measured using MitoSOX Red Reagent at 500 nM (ThermoFisher Scientific, cat. no. M36008). H$_2$O$_2$ (Sigma-Aldrich, cat. no. H1009-5ML) was used to induce ROS.

### Real-time measurement of cellular respiration using a Seahorse bioanalyzer

The Seahorse XF HS Mini FluxPak cartridge (Agilent, 103724-100) was hydrated with 200 µl of H$_2$O and incubated at 37 °C and 0% CO$_2$, at least 12 h before running Seahorse assays. Before the assay, H$_2$O was removed from the FluxPak and replaced with prewarmed Seahorse calibrant (Agilent, 103059-000), then incubated for a further 1 h. Cells were washed in prewarmed Seahorse XF RPMI medium supplemented with 10 mM XF glucose, 1 mM XF sodium pyruvate and 2 mM XF L-Glutamine (Agilent, 103681-100). Cells were counted and washed in XF medium before plating at equal densities in XF HS Mini FluxPak PDL plates (Agilent, 103724-100). Cells were incubated for a further 20 min at 37 °C and 0% CO$_2$. The Seahorse XF Mito Stress Test Kit (Agilent, 103010-100) was used to assess real-time cellular respiration; oligomycin (final concentration 2 µM), FCCP (final concentration 2.5 µM) and rotenone/antimycin A (final concentration 0.5 µM) were loaded into injection ports A, B and C, respectively, in the FluxPak cartridge.

### Suppression assays

B cells were isolated as described and cultured at a density of $1 \times 10^6$ cells per ml for 48 h with 1 µM CpGC, with and without inhibitors. After 48 h, B cells were washed and incubated 1:1 with autologous resting CD4$^+$ T cells (T cells isolated using the EasySep Human CD4$^+$ T cell Isolation Kit, STEMCELL, cat. no. 17962) for a further 72 h in culture plates coated with plate-bound anti-CD3 (clone OKT3, ThermoFisher Scientific, cat. no. 16-0037-81).

### CRISPR–Cas9 silencing

The Lonza P3 Primary Cell 4D-Nucleofector X Kit S (Lonza, cat. no. V4XP-3032) was used for the nucleofection of B cells. *TXN*, *TXN2*, *TXNRD1* and *TXNRD2* guide RNAs (gRNAs) were purchased from IDT. TrueGuide sgRNA Negative Control (ThermoFisher, cat. no. A35526) was used as a nontargeting gRNA control. A cocktail of nucleofection reagents was prepared: per reaction, 11 µl of P3 buffer (Lonza, cat. no. V4XP-3032), 0.33 µl of 62 µM Alt-R Sp Cas9 nuclease V3 (IDT, 1081058) and 2.67 µl of 30 µM gRNA, and incubated for 30 min at room temperature. After incubation, 1 µl of 120 µM Alt-R Cas9 electroporation enhancer (IDT, 1075915) was added. Electroporated controls contained 14 µl of P3 buffer and 1 µl of electroporation enhancer. B cells were isolated as previously described, counted and washed in PBS (-MgCl$_2$, -CaCl$_2$), then resuspended in P3 buffer at $1 \times 10^6$ B cells per 15 µl. For each reaction, 15 µl of B cells in P3 buffer was added to nucleofection reagents, and the combined volume transferred to nucleocuvette strips. B cells were electroporated using a Lonza Amaxa Nucleofector

4D. Electroporated B cells were transferred to culture wells containing 220 µl of warm RPMI (10% FCS, 1% pen/strep) and 1 µM CpGC.

## Confocal microscopy

Isolated B cells were stimulated for 72 h with CpGC. Cultured B cells were incubated for 30 min with MitoTrackers Green and Orange, washed, fixed and permeabilized (ThermoFisher Scientific, 88-8824-00). Cells were incubated in PBS 1% BSA to block nonspecific staining. To stain intracellular Trx, cells were incubated with a 1/25 dilution of anti-Trx antibody (clone 2B1, Bio-Rad, cat. no. MCA1538) for 40 min, washed, then incubated for 30 min with APC-conjugated rat anti-mouse IgG1 (ThermoFisher, cat no. 17-4105-82). Cells were washed and counterstained with DAPI. Cells were imaged on a Nikon Ti2 widefield microscope. Images were analyzed using Fiji.

## Flow cytometry

Flow cytometry was performed with the following directly conjugated antibodies from Biolegend: CD19 BV785 (HIB19, cat. no. 302239), CD24 BV711 (ML5, cat. no. 311135), CD38 BV421 (HB7, cat. no. 356617), IL-6 PE (MQ2-13A5, cat. no. 501106), TGFβ PE/Cy7 (S20006A, cat. no. 300007), IL-27/IL-35 PE (B032F6, cat. no. 360903), CD4 PE (SK3, cat. no. 344605).

IL-10-APC (JES3-19F1, cat. no. 17-7101-82), GM-CSF PE-CF594 (BVD2-21C11, cat. no. 562857), Ki67 BUV395 (B56, cat. no. 564071) and CD3 BUV395 (UCHT-1, cat. no. 563546) were purchased from BD Biosciences. Blimp1/PRDM1 AlexaFluor488 (646702, cat. no. IC36081G) and IL-12/IL-35 p35 FITC (27537, cat. no. IC2191F) were purchased from R&D Systems. TNF eFluor450 (Mab11, cat. no. 48-7349-42) and IFNγ eFluor450 (4S.B3, cat. no. 48-7139-42) were purchased from ThermoFisher Scientific. For Trx and Trx2 staining, anti-Trx (clone 3A1, Bio-Rad, cat. no. VMA00585) and anti-Trx2 (EPR15225, Abcam, cat. no. ab185533) were conjugated to AlexaFluor700 and PE, respectively, using Lightning Link Fast Conjugation Kits (Abcam, cat. nos. ab269824, ab102918). To facilitate intracellular cytokine staining, cells were cultured in complete medium with PMA (50 ng ml$^{-1}$; Sigma-Aldrich, P1585), ionomycin (250 ng ml$^{-1}$; Sigma-Aldrich, I9657) and Brefeldin A (5 µg ml$^{-1}$; Sigma-Aldrich, B7651) for 4.5 h. For multi-color flow cytometric cell-surface staining, cells were stained at 4 °C for 30 min. LIVE/DEAD Fixable Blue Dead Cell Stain (ThermoFisher, L23105) was used to exclude dead cells from analysis. Cells were intracellularly fixed and permeabilized (ThermoFisher Scientific, 88-8824-00), or for additional detection of Blimp1 and Ki67 cells were fixed for 30 min using the FoxP3 Fixation buffer kit (ThermoFisher Scientific, 00-5523-00) before permeabilization. Cells were then incubated with intracellular and intranuclear antibodies for 30 min at 4 °C. Flow cytometric data were collected on an LSRII (BD PharMingen) using FACS Diva software. Data were analyzed using FlowJo (TreeStar).

## scRNA-seq samples, library preparation and sequencing

PBMCs were isolated from three female healthy donors (ages 24–28) and B cells isolated using the EasySep immunomagnetic B cell negative selection kit (STEMCELL). B cells were stimulated with 1 µM CpGC and cultured for 72 h as previously described. Cells were counted and run on a Chromium 10X controller using 5′ chemistry v1 (10X Genomics) in individual lanes according to the manufacturer's instructions. Libraries were generated according to 10X Genomics instructions and run on a High Output HiSeq2500 at one library per lane in 30-10-100 format. To increase the sensitivity of *IL10* detection, human *IL10* primers were used to amplify the signal in a nested PCR. Primer sequences were derived from a publication by Staples et al.[52] and are as follows: forward: 5′-GCC TAA CAT GCT TCG AGA TC-3′; reverse: 5′-TGA TGT CTG GGT CTT GGT TC-3′.

## Data preprocessing, clustering and differential expression

Data were processed through CellRanger (10X Genomics, v.3.1.0) and aligned to the GRCh38 genome. The raw transcript count matrix was loaded into R (v.4.2.1) using RStudio and the Seurat (v.4.0) package[53]. Output matrices were filtered to exclude cells with low gene frequencies (<300), high frequencies (>6,000) or high percentages of mitochondrial DNA (>10%). Feature expression measurements for each cell were normalized by the total expression, multiplied by a scaling factor of 10,000 and log normalized. Cell cycle regression was performed on the normalized dataset to minimize the impact of cell cycle phase on dataset variance. Regressed data were then scaled so each gene had a mean expression of 0 and variance of 1 across cells. To remove non-B cell contaminants, only cells expressing zero *CD4*, *CD8A* and *CD7* and with >0 *CD19* expression were included. In total, 10,018 cells passed quality control processing and contamination removal, with an average of 2,574 genes and 11,154 unique molecular identifiers per cell. The first 20 principal components were selected as these contained the majority of variance. Uniform Manifold Approximation and Projection (UMAP) was performed based on the first 20 principal components using the Seurat::RunUMAP function. Pseudotime trajectory analysis was performed using Monocle3 (ref. 54). Functional enrichment analysis of gene sets was performed using g:Profiler with g:SCS multiple testing correction method, applying significance threshold of 0.05 (ref. 55).

## Quantification and statistical analysis

All values are expressed as the mean ± s.e.m. Graphing of data and analysis of statistical significance were performed in Prism (GraphPad). Data distribution was tested using Shapiro–Wilk normality tests. Statistical significance for normal data was determined using parametric two-tailed paired *t*-tests and one- and two-way analyses of variance (ANOVAs), and for non-normal data by nonparametric two-tailed Mann–Whitney *U* tests. No statistical methods were used to determine sample sizes, but our sample sizes are similar to those reported in previous publications[4,56,57]. Data collection and analysis were not performed blind to the conditions of the experiments.

Results were considered significant at $P < 0.05$; *$P < 0.05$, **$P < 0.01$, ***$P < 0.001$, ****$P < 0.0001$.

## Reporting summary

Further information on research design is available in the Nature Portfolio Reporting Summary linked to this article.

## Data availability

Human scRNA-seq data have been deposited in ArrayExpress under accession code E-MTAB-13872. For scRNA-seq transcriptomic analysis, the GRCh38 reference genome was downloaded using the Bioconductor annotation package BSgenome.Hsapiens.NCBI.GRCh38. All other data are available in the article and Supplementary Information or from the corresponding authors upon reasonable request.

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

## Acknowledgements

This work is funded by a Versus Arthritis UK program grant (no. 21140), awarded to C. Mauri; a UCB BIOPHARMA SPRL/BBSRC PhD Studentship (grant no. BB/P504725/1) awarded to C. Mauri to fund H.F.B.; an MRC grant (no. MR/L01257X/2) awarded to D.D.-W.; and a BBSRC grant (no. BB/T002212/1) awarded to C. Mauri, F.F. and D.D.-W. T.C.R.M. is funded by a Medical Research Foundation Fellowship (grant no. MRF-057-0004-RG-MCDO-C0800) and a Versus Arthritis Senior Fellowship (grant no. ShS/SRF/22977). C. Mauro is funded by a British Heart Foundation Senior Basic Science Research Fellowship (grant no. FS/SBSRF/22/31031).

## Author contributions

H.F.B. designed and performed experiments, analyzed data and wrote the paper. T.C.R.M. designed experiments, provided resources and critically reviewed the paper. A. Stewart and D.D.-W. provided scRNA-seq datasets. Z.B. assisted with confocal imaging. A. Skelton, J.N. and F.F. assisted with bioinformatics analysis. A.R.K. provided resources and critically reviewed the paper. D.A.I. provided patient samples and clinical expertise. C. Mauro helped design experiments and critically reviewed the paper. C. Mauri designed experiments, analyzed data and wrote the paper.

## Competing interests

The authors declare no competing interests.

## Additional information

**Extended data** is available for this paper at https://doi.org/10.1038/s41590-024-01798-w.

**Correspondence and requests for materials** should be addressed to Hannah F. Bradford or Claudia Mauri.

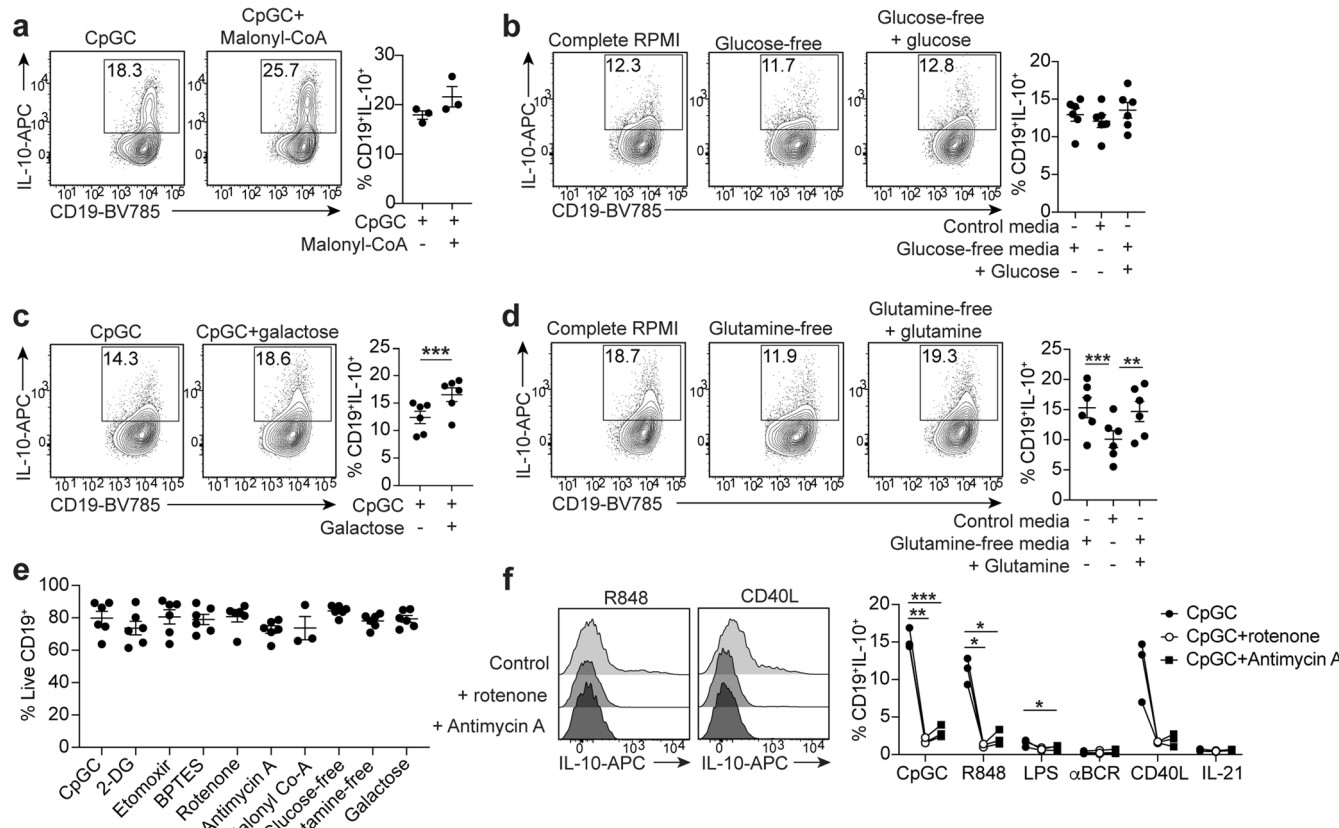

**Extended Data Fig. 1 | Glutamine and ETC activity are required for the differentiation of IL-10+ Breg cells. a-d**, Representative contour plots and cumulative data show frequencies of CD19+IL-10+ B cells after 72h CpGC stimulation when cultured (a) in the presence of 10mM Malonyl co-A, (b) in glucose-free media or glucose-free media replenished with glucose, (c) in the presence of 5mM galactose (***P = 0.0005), (d) in the absence of glutamine or glutamine-free media replenished with glutamine (***P = 0.0001, **P = 0.0022). (b,c,d) n = 6 biologically independent samples examined over 2 independent experiments, (a) n = 3 biologically independent samples examined over 2 independent experiments. **e**, Cumulative data showing percentage of live cells within CD19+B cells after 72h CpGC stimulation with and without metabolism inhibitors. n = 6 biologically independent samples examined over 2 independent experiments. **f**, Representative histograms and cumulative data show frequencies of CD19+IL-10+ B cells after 72h stimulation with CpGC, TLR7 ligand R848, TLR4 ligand LPS, aBCR, CD40L, or IL-21, with and without rotenone or antimycin A. n = 3 biologically independent samples examined over 2 independent experiments (left to right) **P = 00025, ***P = 0.0002, *P = 0.0179, *P = 0.0139, *P = 0.0372. Data analysed by (c) two-sided paired t-test, (d) one-way ANOVA followed by Tukey's multiple comparison test, (f) two-way ANOVA followed by Sidak's test for multiple comparisons. Error bars are shown as mean±SEM.

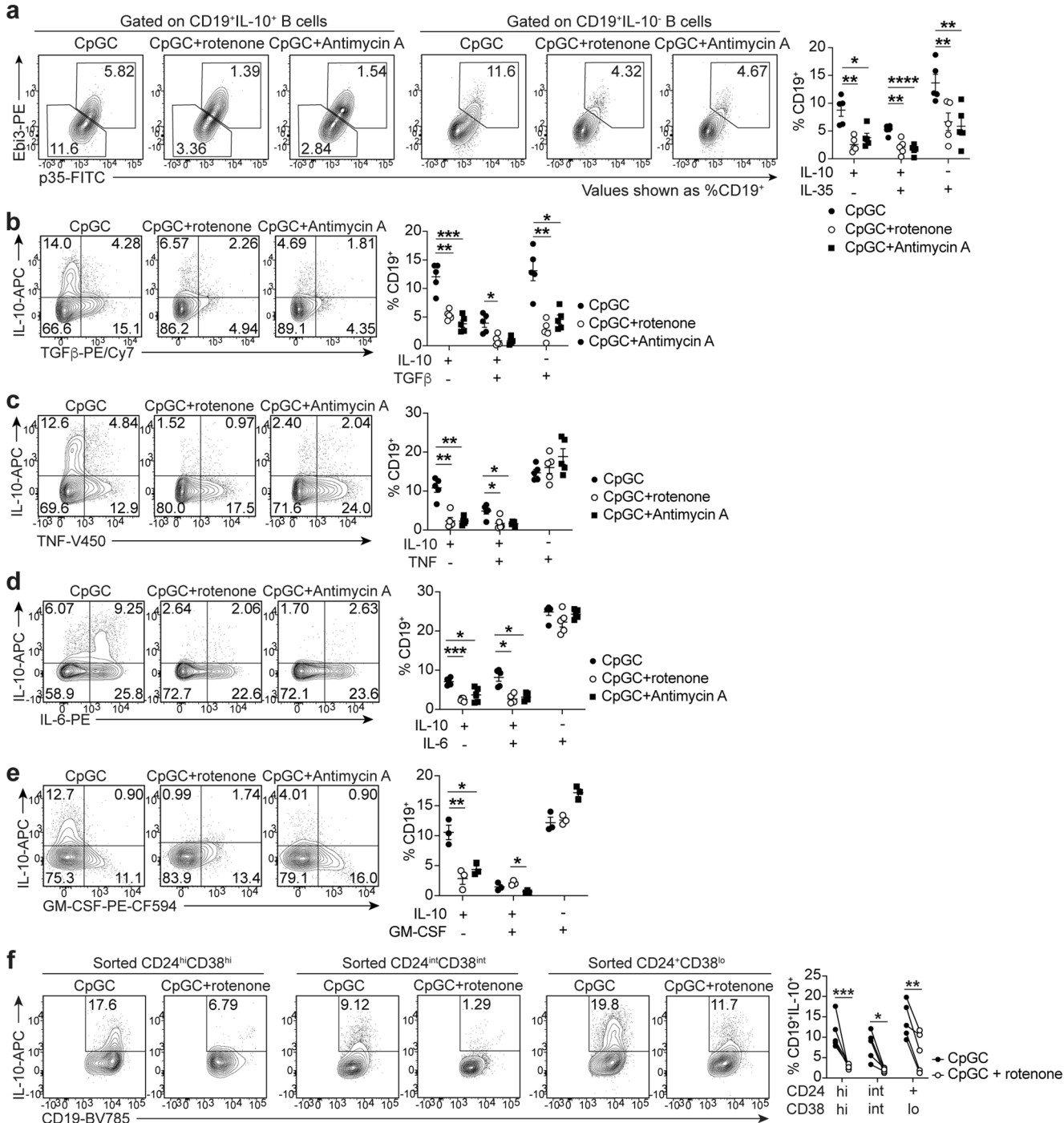

**Extended Data Fig. 2 | Electron transport chain regulate the induction of immunoregulatory cytokines expressed by Breg cells. a-e**, Representative contour plots and cumulative data show frequencies of (a) CD19⁺IL-10⁺IL-35⁻, CD19⁺IL-10⁺IL-35⁺, CD19⁺IL-10⁻IL-35⁺ B cells (left to right; **P = 0.0018, *P = 0.0255, **P = 0.0025, ****P < 0.0001, **P = 0.0042, **P = 0.0014) (b) CD19⁺IL-10⁺TGFb⁻, CD19⁺IL-10⁺TGFb⁺ and CD19⁺IL-10⁻TGFb⁺ B cells, (left to right; **P = 0.006, ***P = 0.0007, *P = 0.042, **P = 0.0044, *P = 0.043 (c) CD19⁺IL-10⁺TNF⁻, CD19⁺IL-10⁺TNF⁺ and CD19⁺IL-10⁻TNF⁺ B cells (left to right; **P = 0.0052, **P = 0.0053, *P = 0.0134, *P = 0.0419), (d) CD19⁺IL-10⁺IL-6⁻, CD19⁺IL-10⁺IL-6⁺, CD19⁺IL-10⁻IL-6⁺ B cells (left to right; *P = 0.0278, ***P = 0.0005, *P = 0.0174, *P = 0.0411), and (e) CD19⁺IL-10⁺GM-CSF⁻, CD19⁺IL-10⁺GM-CSF⁺ and CD19⁺IL-10⁻GM-CSF⁺ B cells

(left to right; *P = 0.0204, **P = 0.0063, *P = 0.0124) after 72h CpGC stimulation of isolated B cells with and without rotenone or antimycin A. CD19⁺IL-35⁺ B cells were identified as Ebi3⁺p35⁺. (a,b,c,d) n = 5 biologically independent samples examined over 2 independent experiments, (e) n = 3 biologically independent samples examined over 2 independent experiments. **f**, Representative contour plots and cumulative data show frequencies of CD19⁺IL-10⁺ B cells following 72h stimulation of sorted *ex vivo* transitional, naïve-mature and memory subsets with CpGC, with and without 1mM rotenone. n = 6 biologically independent samples examined over 2 independent experiments, ***P = 0.0010, *P = 0.0108, **P = 0.0018. Data analyzed by two-way ANOVA followed by Sidak's test for multiple comparisons. Error bars are shown as mean±SEM.

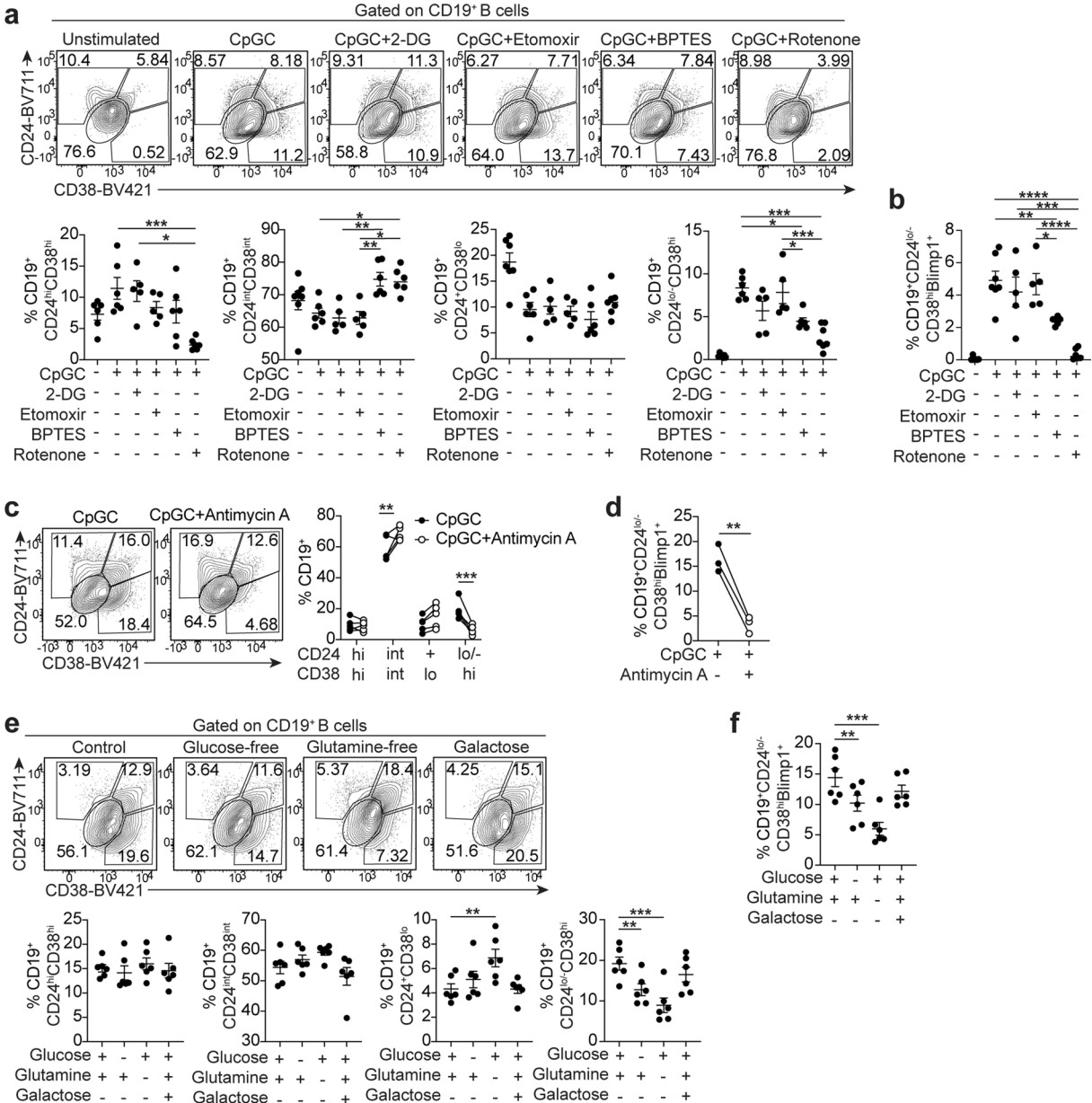

**Extended Data Fig. 3 | Inhibition of OXPHOS impairs plasma cell differentiation. a-b**, Representative contour plots and cumulative data show frequencies of (a) transitional (CD19⁺CD24ʰⁱCD38ʰⁱ), mature-naïve (CD19⁺CD24ⁱⁿᵗCD38ⁱⁿᵗ), memory (CD19⁺CD24⁺CD38ˡᵒ) B cells and (b) plasmablasts (CD19⁺CD24ˡᵒ/⁻CD38ʰⁱBlimp1⁺) following stimulation of isolated B cells for 72h with CpGC, with and without 1mM 2-DG, 10mM etomoxir, 500nM BPTES or 1mM rotenone. n = 6 biologically independent samples examined over 2 independent experiments. (a - left to right) ***P = 0.0005, *P = 0.0251, *P = 0.0242, **P = 0.0077, *P = 0.0108, **P = 0.0076, ***P = 0.0001, *P = 0.0120, ***P = 0.0007, *P = 0.0488, (b – left to right) ****P < 0.0001, ***P = 0.0001, **P = 0.0081, ****P < 0.0001, *P = 0.0378. **c-d**, Representative contour plots and cumulative data show frequencies of (c) transitional (CD19⁺CD24ʰⁱCD38ʰⁱ), mature-naïve (CD19⁺CD24ⁱⁿᵗCD38ⁱⁿᵗ), memory (CD19⁺CD24⁺CD38ˡᵒ) B cells (**P = 0.0037, ***P = 0.0001) and (d) plasmablasts (CD19⁺CD24ˡᵒ/⁻CD38ʰⁱBlimp1⁺) (**P = 0.005) following stimulation of isolated B cells for 72h with CpGC, with and without

10nM Antimycin A. (c) n = 5 biologically independent samples examined across 2 independent experiments. (d) n = 3 biologically independent samples examined over 2 independent experiments. **e-f**, Representative contour plots and cumulative data show frequencies of (e) transitional (CD19⁺CD24ʰⁱCD38ʰⁱ), mature-naïve (CD19⁺CD24ⁱⁿᵗCD38ⁱⁿᵗ), memory (CD19⁺CD24⁺CD38ˡᵒ) B cells (left to right; **P = 0.0027, **P = 0.0044, ***P = 0.0003) and (f) plasmablasts (CD19⁺CD24ˡᵒ/⁻CD38ʰⁱBlimp1⁺) (**P = 0.0097, ***P = 0.0008) following stimulation of isolated B cells for 72h with CpGC, in glucose-free RPMI or glucose-free RPMI supplemented with glucose, glutamine-free RPMI supplemented with glutamine, or galactose. n = 6 biologically independent samples examined over 2 independent experiments. Data analysed by (a,b,e,f) one-way ANOVA followed by Tukey's multiple comparisons test, (c) two-way ANOVA followed by Sidak's test for multiple comparisons, or (d) two-sided paired *t*-test. Error bars are shown as mean±SEM.

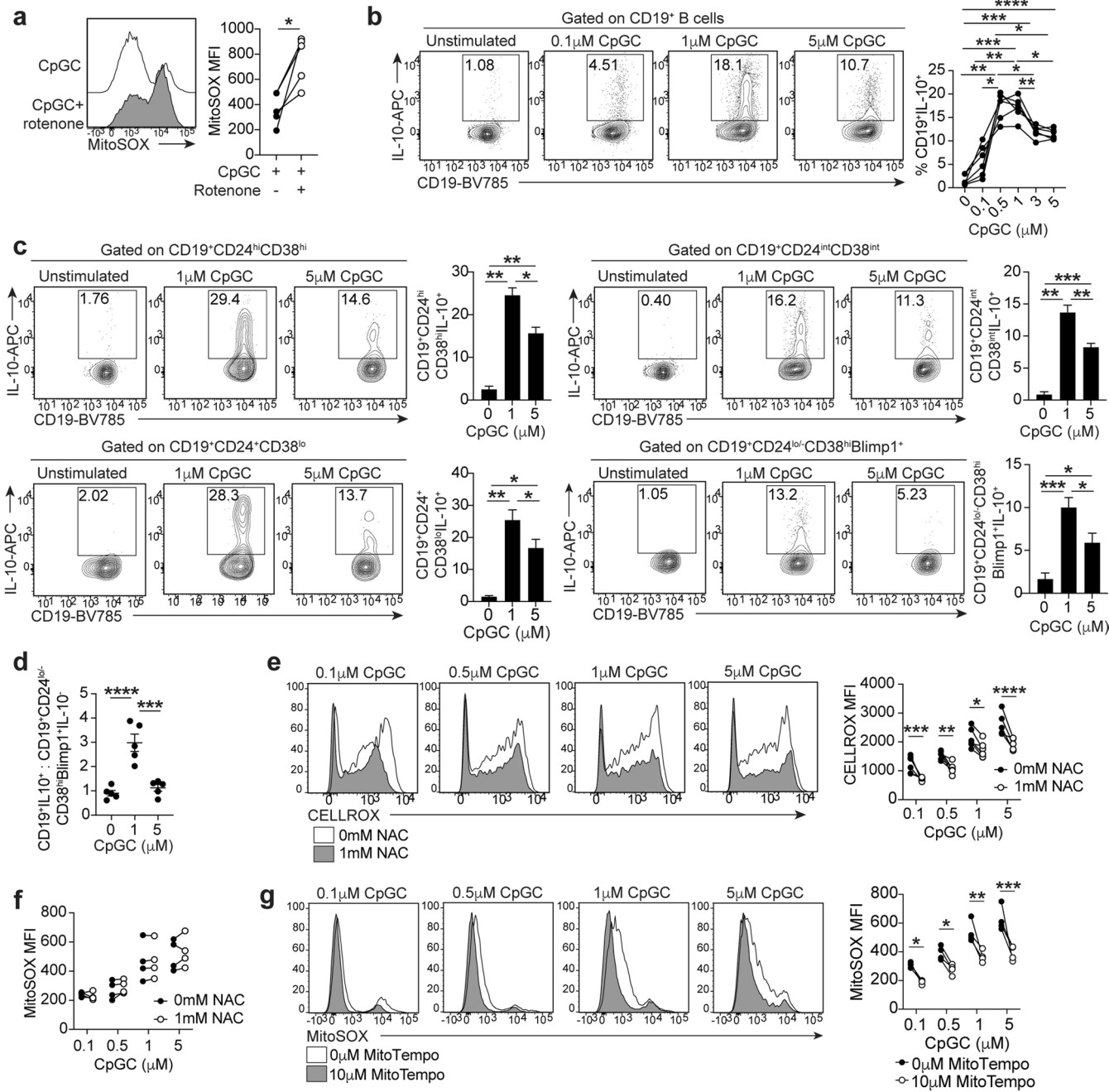

**Extended Data Fig. 4 | Elevated cellular ROS results in a reduced IL-10⁺Breg cell to IL-10ⁿᵉᵍᵃᵗⁱᵛᵉ plasmablast ratio. a**, Cumulative data showing the levels of mitochondrial ROS (MitoROS) in B cells after 24h CpGC stimulation with and without rotenone. n = 5 biologically independent samples examined over 2 independent experiments, *P = 0.0136. **b**, Representative contour plots and cumulative data showing frequencies of CD19⁺IL-10⁺B cells following stimulation of isolated B cells with increasing concentrations of CpGC for 72h. n = 5 biologically independent samples examined over 2 independent experiments. (top to bottom) ****P < 0.0001, ***P = 0.0005, *P = 0.0294, ***P = 0.0005, **P = 0.0064, *P = 0.0122, **P = 0.0012, *P = 0.0117, *P = 0.0218, **P = 0.0087. **c**, Representative contour plots and cumulative data show frequencies of CD19⁺IL-10⁺B cells within transitional (CD19⁺CD24ʰⁱCD38ʰⁱ), mature-naïve (CD19⁺CD24ⁱⁿᵗCD38ⁱⁿᵗ), memory (CD19⁺CD24⁺CD38ˡᵒ) B cells and plasmablasts (CD19⁺CD24ˡᵒ/⁻CD38ʰⁱBlimp1⁺) following stimulation of isolated B cells for 72h with 1mM or 5mM CpGC. n = 6 biologically independent samples examined over 2 independent experiments. (left to right, top to bottom) **P = 0.0025, **P = 0.0024, *P = 0.0209, ***P = 0.0006, **P = 0.0011, **P = 0.0059, *P = 0.0112, **P = 0.0035, *P = 0.0291, ***P = 0.0002, *P = 0.0284, *P = 0.0326. **d**, Cumulative

data showing the ratio of CD19⁺IL-10⁺Bregs to CD19⁺CD24ˡᵒ/⁻CD38ʰⁱBlimp1⁺ plasmablasts after stimulation of B cells with 1mM or 5mM CpGC for 72h. n = 5 biologically independent samples examined over 2 independent experiments. ****P < 0.0001, ***P = 0.0003. **e**, Histograms and cumulative data show the levels of cytoplasmic ROS (CELLROX) after stimulation of B cells with 0.1–5mM CpGC, with and without 1mM NAC. n = 6 biologically independent samples examined over 2 independent experiments. ***P = 0.0009, **P = 0.0017, *P = 0.01, ****P < 0.0001. **f**, Cumulative data shows the levels of mitochondrial ROS after stimulation of B cells with 0.1–5mM CpGC, with and without 1mM NAC. n = 4 biologically independent samples examined over 2 independent experiments. **g**, Histograms and cumulative data show the levels of mitochondrial ROS after stimulation of B cells with 0.1–5mM CpGC, with and without 10mM MitoTempo. n = 3 biologically independent samples examined over 2 independent experiments. *P = 0.0346, *P = 0.0477, **P = 0.0035, ***P = 0.0002. Data analysed by (a) two-sided paired t-test, (b-d) one-way ANOVA followed by Tukey's multiple comparisons test, or (e,g) two-way ANOVA followed by Sidak's test for multiple comparisons. Error bars are shown as mean±SEM.

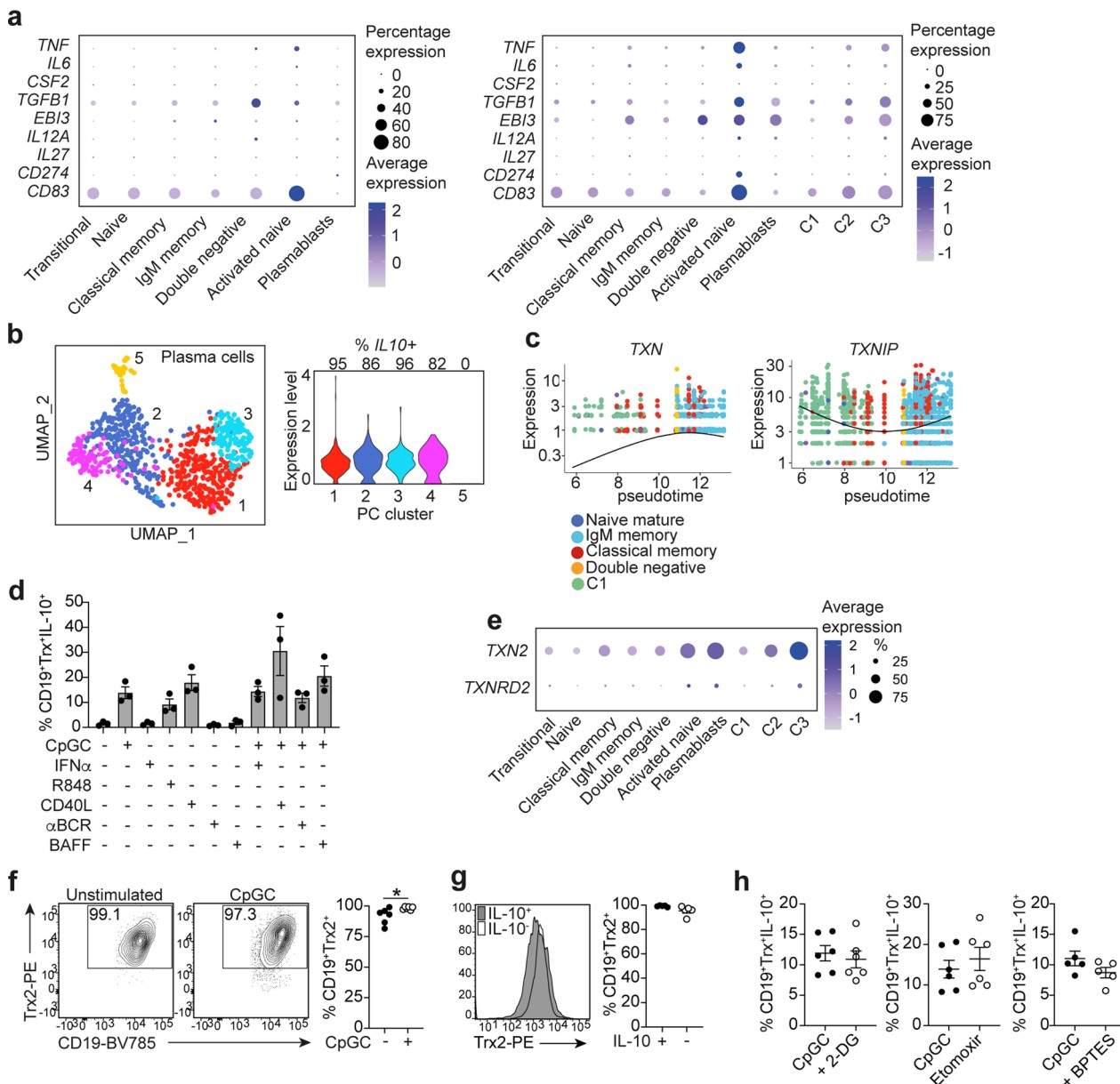

**Extended Data Fig. 5 | B cells expressing high levels of IL-10 express high levels of Trx.** Trx expression does not rely on glycolysis, fatty acid oxidation or glutamine metabolism and its expression mirrors that of IL-10 following different stimuli. **a**, Dotplots showing expression of other cytokines and immunoregulatory markers in (left) *ex vivo* B cells and (right) B cell clusters after CpGC stimulation. **b**, UMAP and violin plot show sub-clustering of plasmablasts/ plasma cells and associated expression of *IL10*. **c**, Graph shows the expression levels of *TXN* and *TXNIP* with pseudotime within trajectories seeding classical memory, IgM memory and double negative B cell subsets. **d**, Graph showing frequencies of CD19⁺IL-10⁺Trx⁺B cells following 72h culture with different stimuli. n = 3 biologically independent samples examined over 2 independent experiments. **e**, Dotplot showing average and percentage expression of *TXN2*,

and *TXNRD2* in CpGC-stimulated B cell clusters. **f**, Representative contour plots and cumulative data show frequencies of CD19⁺Trx2⁺B cells in B cells stimulated with and without CpGC for 72h. n = 6 biologically independent samples examined over 2 independent experiments. *P = 0.0448, by two-sided paired *t*-test. **g**, Representative histogram and cumulative data show frequencies of CD19⁺Trx2⁺B cells within respectively IL-10⁺ and IL-10⁻B cell subsets after 72h CpGC stimulation. n = 5 biologically independent samples examined over 2 independent experiments. **h**, Cumulative data show frequencies of CD19⁺Trx⁺IL-10⁺B cells following stimulation of isolated B cells with CpGC with and without 1mM 2-DG, 10mM etomoxir or 500nM BPTES for 72h. n = 6 biologically independent samples examined over 2 independent experiments. Error bars are shown as mean±SEM.

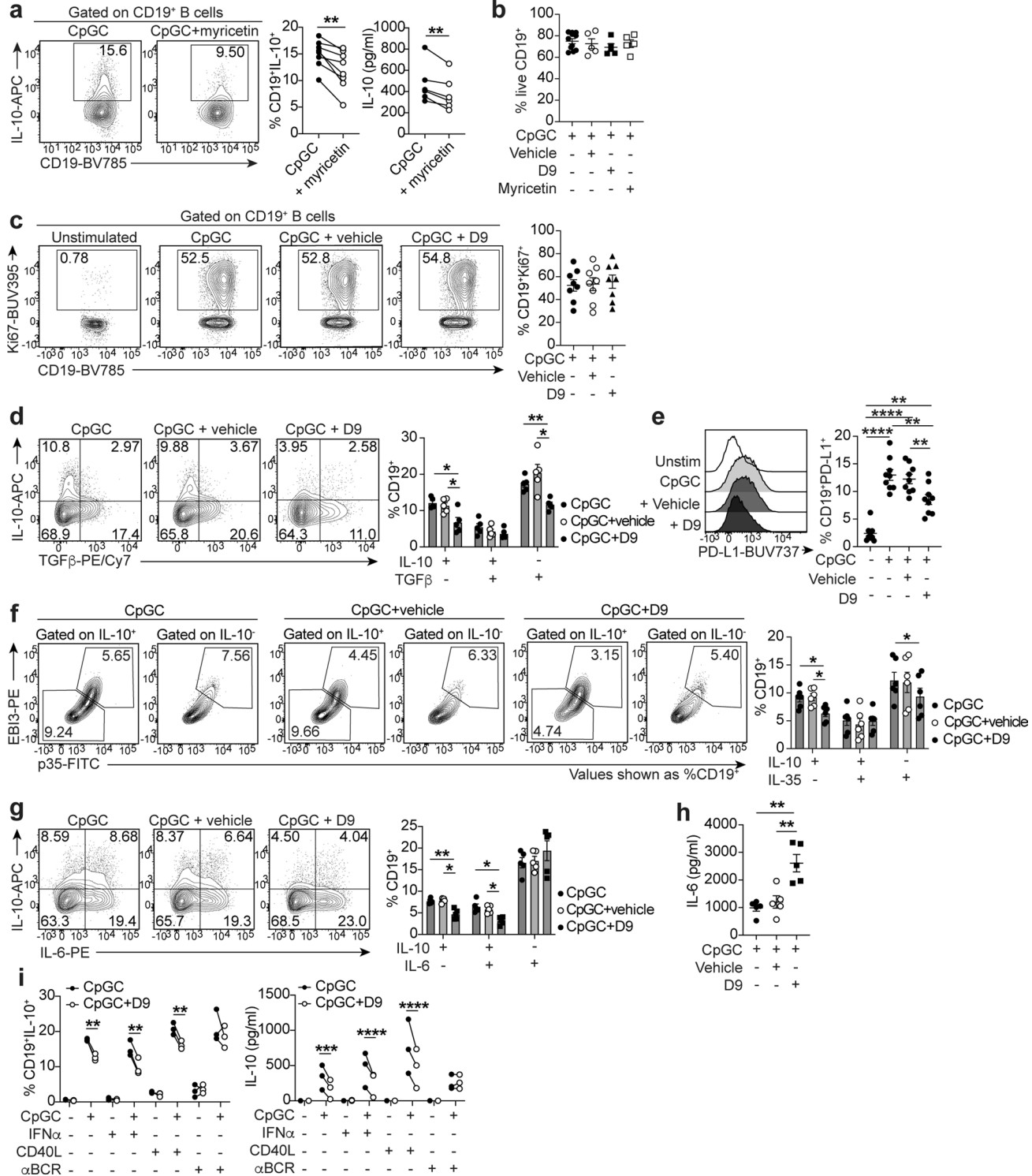

**Extended Data Fig. 6 | See next page for caption.**

**Extended Data Fig. 6 | Trx inhibition suppresses TGFb, IL-35 and PD-L1 expression by B cells, increases IL-6 secretion. a**, Representative contour plots and cumulative data show frequencies of CD19⁺IL-10⁺B cells and IL-10 secretion following stimulation of isolated B cells with CpGC with and without 10mM of the TrxR inhibitor myricetin. n = 8 biologically independent samples examined over 3 independent experiments. (left to right) **P = 0.0027, **P = 0.0056. **b**, Cumulative data show frequencies of live cells within CD19⁺ B cells after stimulation of B cells with CpGC, with and without D9, vehicle control, or myricetin. **c**, Representative contour plots and cumulative data show frequencies of CD19⁺Ki67⁺B cells following stimulation of isolated B cells with CpGC with and without 100nM D9 or vehicle control. n = 8 biologically independent samples examined over 3 independent experiments. **d-f**, Representative contour plots and cumulative data show frequencies of (d) CD19⁺IL-10⁺TGFb⁻, CD19⁺IL-10⁺TGFb⁺ and CD19⁺IL-10⁻TGFb⁻ B cells (left to right) *P = 0.0119, *P = 0.0208, **P = 0.0049, *P = 0.0264, (e) PD-L1⁺ B cells (top to bottom) **P = 0.003, ****P < 0.0001, **P = 0.0017, ****P < 0.0001, **P = 0.0011, and (f) CD19⁺IL-10⁺IL-35⁻, CD19⁺ IL-10⁺IL-35⁺ and CD19⁺IL-10⁻IL-35⁺ B cells (left to right) *P = 0.0184, *P = 0.0245, *P = 0.0296, following 72h CpGC stimulation of isolated B cells with and without

D9 or vehicle control. (d) n = 5 biologically independent samples examined over 2 independent experiments, (e) n = 9 biologically independent samples examined over 3 independent experiments, (f) n = 6 biologically independent samples examined over 2 independent experiments. **g,h**, Representative contour plots and graphs show (g) the frequencies of CD19⁺IL-10⁺IL-6⁻, CD19⁺IL-10⁺IL-6⁺, and CD19⁺IL-10⁻IL-6⁺ B cells (left to right) **P = 0.0096, *P = 0.0189, *P = 0.0126, *P = 0.0223, and (h) IL-6 secretion (top to bottom) **P = 0.001, **P = 0.0031, following 72h CpGC stimulation of isolated B cells with and without D9 or vehicle control. n = 5 biologically independent samples examined over 2 independent experiments. **i**, Cumulative data showing frequencies of CD19⁺IL-10⁺B cells and IL-10 secretion following 72h stimulation of isolated B cells with CpGC alone, or CD40L, IFNa, anti-BCR either alone or in combination with CpGC, with and without 100nM D9. n = 3 biologically independent samples examined over 2 independent experiments. (left to right) **P = 0.0042, **P = 0.0026, **P = 0.0078, ***P = 0.0007, ****P < 0.0001, ****P < 0.0001. Data analyzed by (a) two-sided paired t-test, (e,h) one-way ANOVA followed by Tukey's test for multiple comparisons, (d,f,g,i) or two-way ANOVA followed by Sidak's test for multiple comparisons. Error bars are shown as mean±SEM.

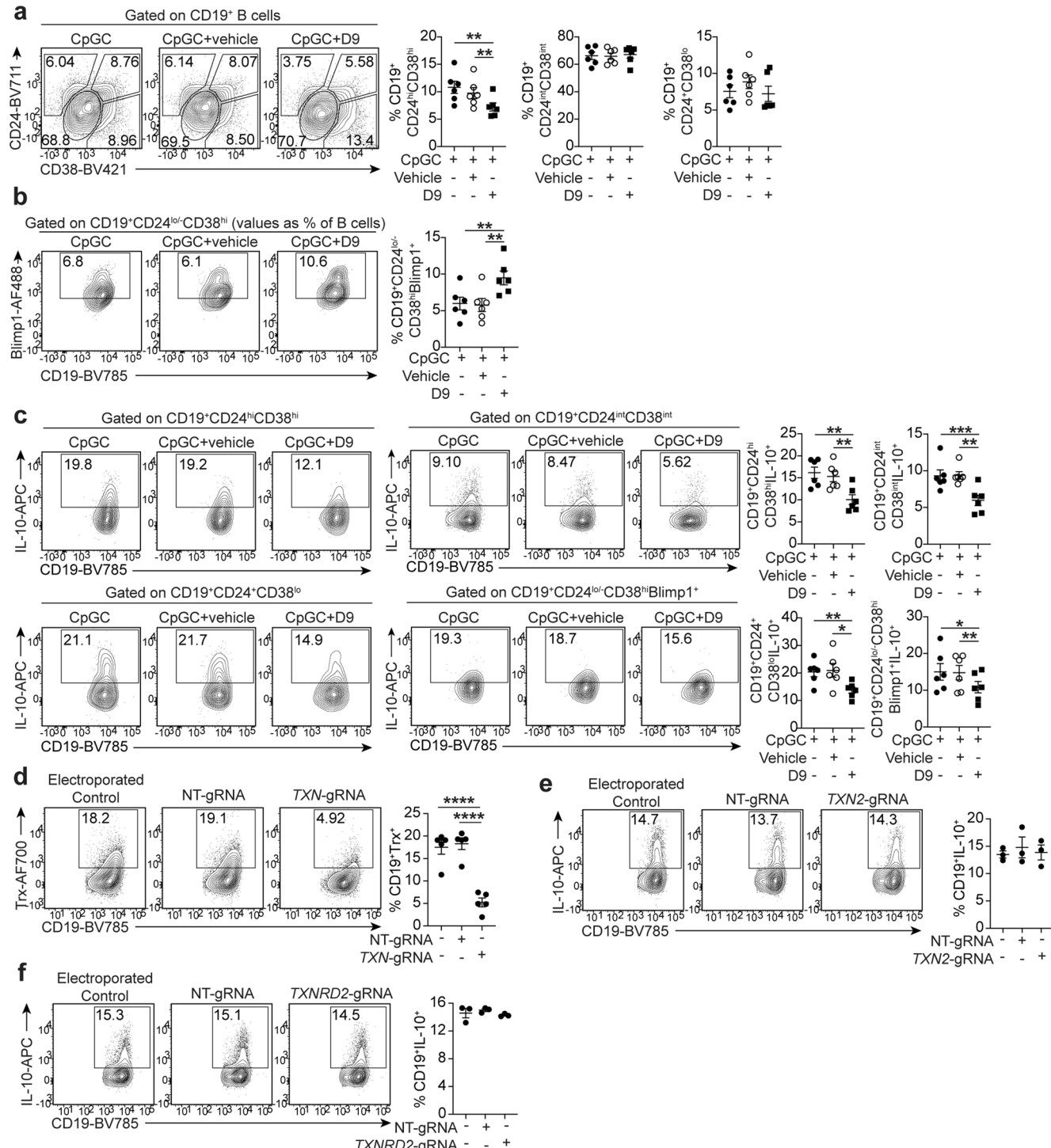

**Extended Data Fig. 7 | Trx system inhibition expands CD24$^{lo/-}$CD38$^{hi}$Blimp1$^{+}$ plasmablasts and inhibits IL-10$^{+}$Breg cell differentiation. a-b,** Representative contour plots and cumulative data show (a) frequencies of CD19$^{+}$CD24$^{hi}$CD38$^{hi}$ (immature), CD19$^{+}$CD24$^{int}$CD38$^{int}$ (mature naïve), CD19$^{+}$CD24$^{+}$CD38$^{lo}$ (memory) B cells (**$P$ = 0.0022, **$P$ = 0.0018) and (b) CD19$^{+}$CD24$^{lo/-}$CD38$^{hi}$Blimp1$^{+}$ plasmablasts (**$P$ = 0.0014) following 72h stimulation of isolated B cells with CpGC, with and without vehicle control or 100nM D9. n = 6 biologically independent samples examined over 2 independent experiments. **c,** Cumulative data show frequencies of IL-10$^{+}$B cells within CD19$^{+}$CD24$^{hi}$CD38$^{hi}$ (immature), CD19$^{+}$CD24$^{int}$CD38$^{int}$ (mature naïve), CD19$^{+}$CD24$^{+}$CD38$^{lo}$ (memory) B cells and CD19$^{+}$CD24$^{lo/-}$CD38$^{hi}$ plasmablasts following 72h stimulation of isolated B cells with CpGC, with and without vehicle control or 100nM D9. (left to right, top

to bottom) **$P$ = 0.0061, **$P$ = 0.0042, ***$P$ = 0.0006, **$P$ = 0.0031, **$P$ = 0.0018, *$P$ = 0.0208, *$P$ = 0.03, **$P$ = 0.0094. n = 6 biologically independent samples examined over 2 independent experiments. **d,** Representative contour plots and cumulative data show frequencies of CD19$^{+}$Trx$^{+}$ B cells after CRISPR/Cas9 silencing of *TXN* and culture for 72h with CpGC. ****$P$ < 0.0001. n = 5 biologically independent samples examined over 2 independent experiments. **e-f,** Representative contour plots and cumulative data show frequencies of CD19$^{+}$IL-10$^{+}$ B cells after CRISPR/Cas9 silencing of (d) *TXN2* and (e) *TXNRD2* and culture for 72h with CpGC. n = 3 biologically independent samples examined over 2 independent experiments. Data analyzed by one-way ANOVA followed by Tukey's test for multiple comparisons. Error bars are shown as mean±SEM.

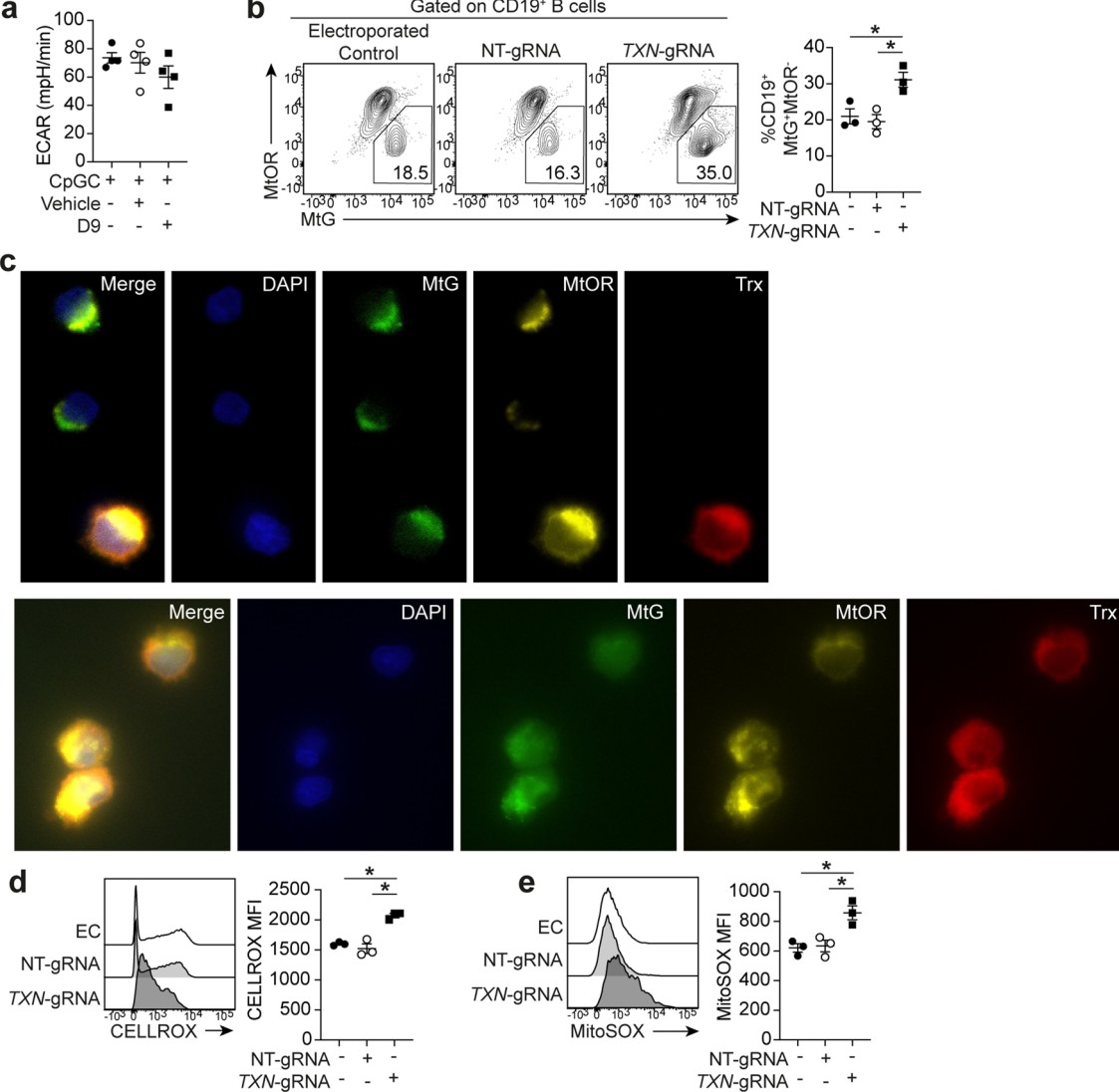

**Extended Data Fig. 8 | Silencing *TXN* results in mitochondrial membrane depolarisation and increased levels of cytoplasmic and mitochondrial ROS. a**, Graph shows extracellular acidification rate (ECAR) of B cells following stimulation with CpGC, with and without vehicle control or 100nM D9. n = 4 biologically independent samples examined over 2 independent experiments. **b**, Representative contour plots stained with MitoTracker Green (MtG: to assess mitochondrial mass) and MitoTracker Orange (MtOR: its accumulation is dependent upon membrane potential) and graph show frequencies of B cells with depolarized (CD19⁺MtG⁺MtOR⁻) mitochondrial membranes following CRISPR/Cas9 silencing of *TXN* and 24h stimulation of B cells with CpGC, with and without vehicle control or 100nM D9. (top to bottom) *P = 0.0154, *P = 0.0324.

n = 3 biologically independent samples examined over 2 independent experiments. **c**, Confocal images show expression of Trx (red) alongside MtG (green) and MtOR (yellow) staining in B cells stimulated for 72h with CpGC. **d,e**, Histograms and cumulative data show the levels of (d) cytoplasmic (CELLROX) ROS (*P = 0.0126, *P = 0.0231) and (e) mitochondrial (MitoSOX) ROS (top to bottom - *P = 0.0406, *P = 0.0437) in B cells after CRISPR/Cas9 silencing of *TXN* and CpGC stimulation for 24h. n = 3 biologically independent samples examined over 2 independent experiments. Data analyzed by one-way ANOVA followed by Tukey's test for multiple comparisons. Error bars are shown as mean±SEM.

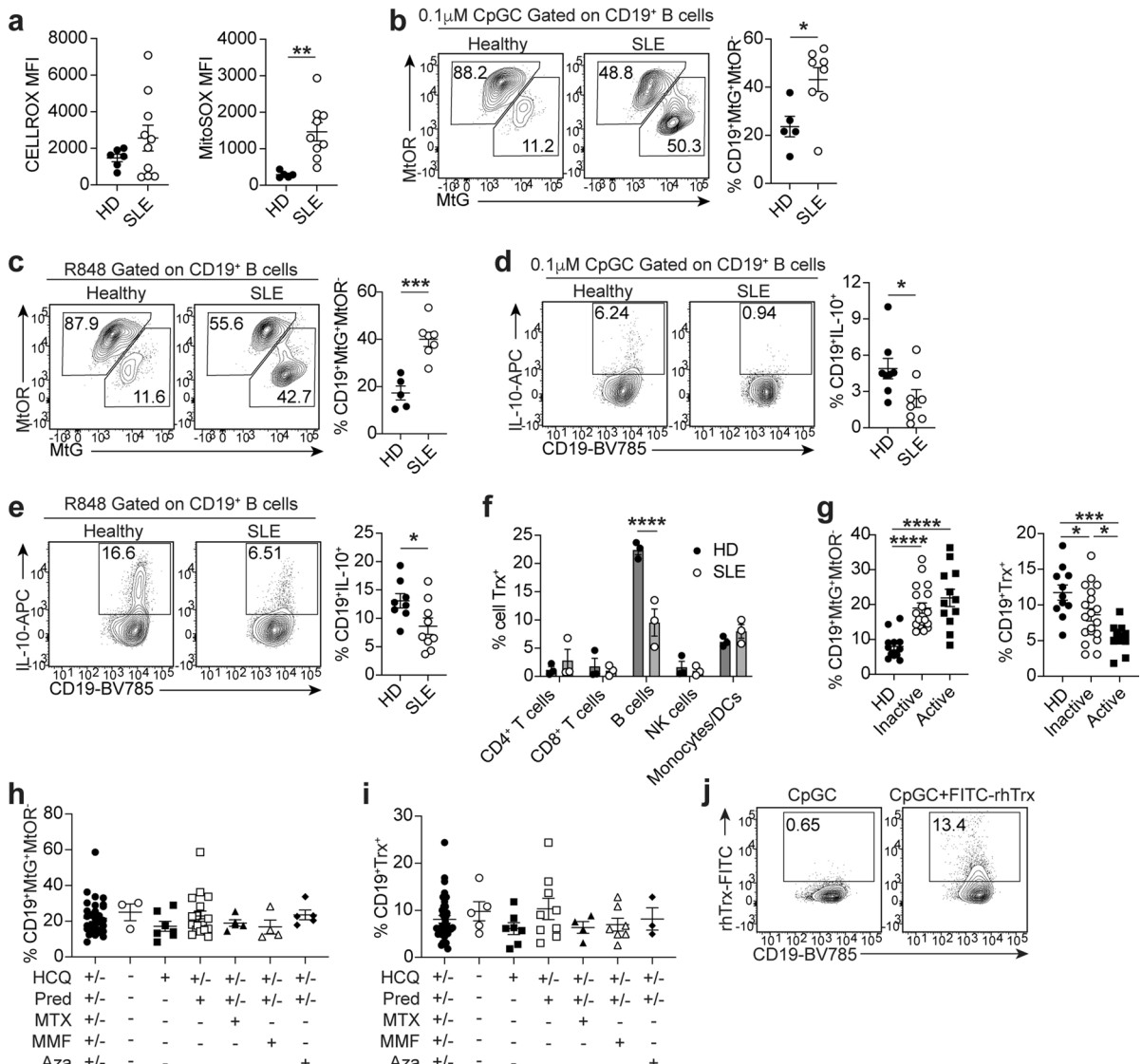

**Extended Data Fig. 9 | B cells, but not other immune-cells, from SLE patients display reduced IL-10⁺Breg cells and depolarised mitochondrial membranes when stimulated with R848 or low-dose CpGC. a**, Cumulative data showing the levels of cytoplasmic (CELLROX) and mitochondrial ROS (MitoSOX) in *ex vivo* B cells of SLE patients and healthy donors (HD). (CELLROX – SLE n = 10, HD n = 6, MITOSOX – SLE n = 9, HD n = 5) **P = 0.0051. **b-c**, Representative contour plots and cumulative data showing frequencies of B cells with depolarised mitochondrial membranes (CD19⁺MtG⁺MtOR⁻) from SLE patients and healthy donors after 72h stimulation with (b) 0.1mM CpGC (SLE n = 8, HD n = 5) (*P = 0.0197) or (c) TLR7 ligand R848 (SLE n = 7, HD n = 5) (***P = 0.0004). **d-e**, Representative contour plots and cumulative data showing frequencies of CD19⁺IL-10⁺ B cells from SLE patients and healthy donors after 72h stimulation with (d) 0.1mM CpGC (SLE n = 8, HD n = 8) (*P = 0.0439) or (e) the TLR7 ligand R848 (SLE n = 9, HD n = 8) (*P = 0.0357). **f**, Cumulative data show the frequencies of Trx⁺ cells within different immune cell subsets following 72h CpGC stimulation of SLE patient

(n = 3) or healthy donor (n = 3) PBMCs (****P < 0.0001). **g**, Cumulative data showing frequencies of CD19⁺MtG⁺MtOR⁻ (****P < 0.0001) and CD19⁺Trx⁺B cells (***P = 0.0001, *P = 0.0439, *P = 0.0345) in SLE patients with active disease (BILAG global score (GS) over or equal to 5, n = 13) and inactive (BILAG GS < 5) disease, compared to healthy donors (CD19⁺MtG⁺MtOR⁻; HD n = 13, active SLE n = 12, inactive SLE n = 18, CD19⁺Trx⁺; HD n = 11, active SLE n = 12, inactive SLE n = 20). **h-i**, Cumulative data showing frequencies of (h) CD19⁺MtG⁺MtOR⁻ (n = 22) and (i) CD19⁺Trx⁺B cell frequencies in SLE patients (n = 26) treated with hydroxychloroquine (HCQ), prednisolone (Pred), methotrexate (MTX), mycophenolate mofetil (MMF) or azathioprine (Aza). **j**, Contour plots show the intracellular presence of FITC-conjugated recombinant human Trx (rhTrx) in B cells after 72h culture with CpGC and exogenous FITC-rhTrx. Data analyzed by (a-e) two-sided unpaired *t*-test, (f) two-way ANOVA followed by Sidak's test for multiple comparisons, or (g) one-way ANOVA followed by Tukey's test for multiple comparisons. Error bars are shown as mean±SEM.

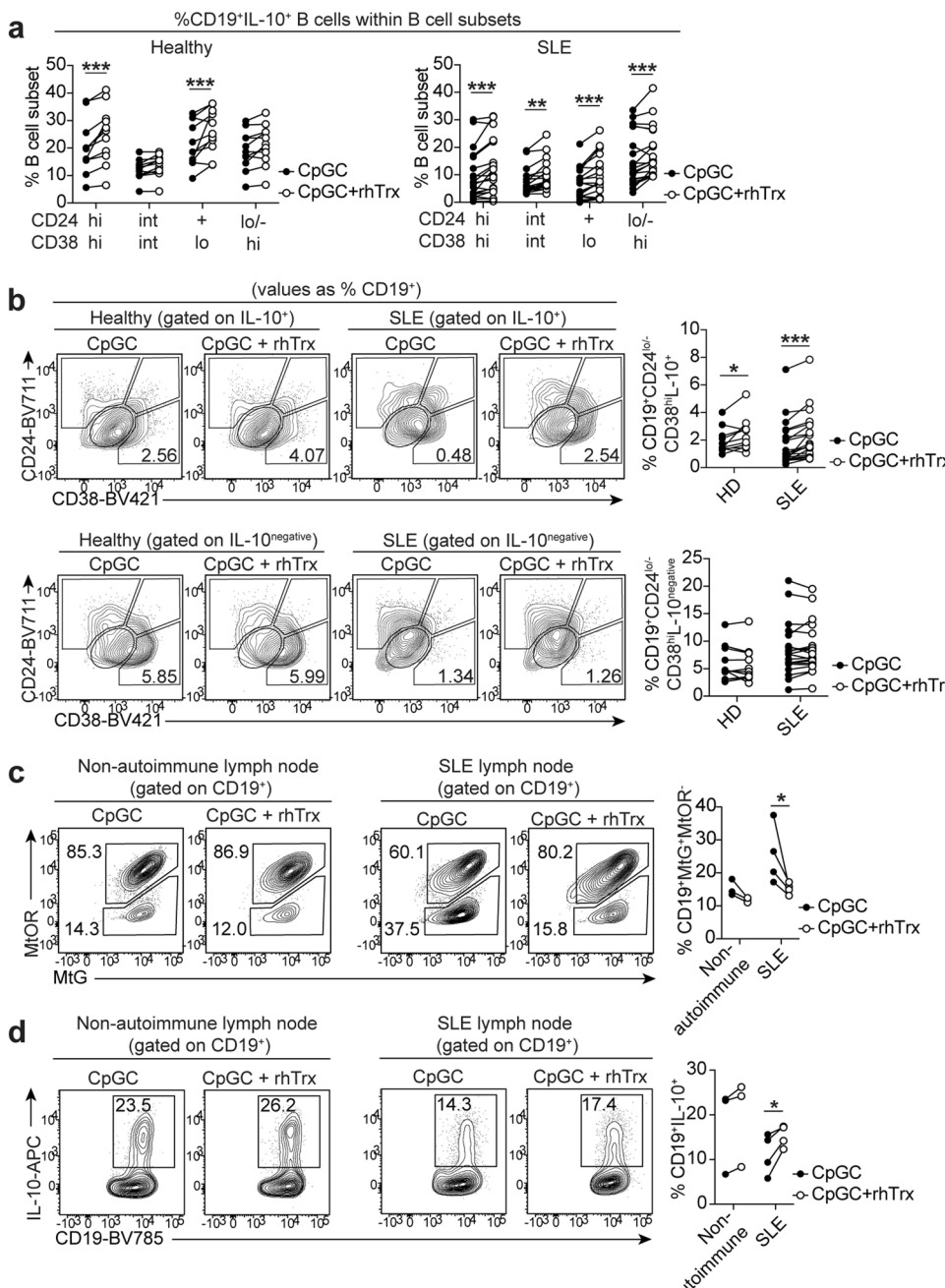

**Extended Data Fig. 10 | Exogenous Trx expands IL-10+Breg cells within all B cell subsets in SLE patients and rescues Mitochondrial Membrane Polarization and IL-10+Breg cells in SLE patient lymph node B cells.**
**a**, Cumulative data showing frequencies of IL-10+B cells within CD24hiCD38hi (immature), CD24intCD38int (mature naïve), CD24+CD38lo (memory) B cells and CD24lo/-CD38hi plasmablasts in healthy donors (n = 11) and SLE patients (n = 20) following 72h stimulation of isolated B cells with CpGC, with and without 100mM recombinant human thioredoxin (rhTrx). (left to right) ***P = 0.0002, ***P = 0.0003, ***P = 0.0004, **P = 0.00068, ***P = 0.0009, ***P = 0.0009.
**b**, Cumulative data showing frequencies of IL-10+ and IL-10− CD24lo/−CD38hi

plasmablasts (as a frequency of total CD19+B cells) in healthy donors (n = 11) and SLE patients (n = 20) following 72h stimulation of isolated B cells with CpGC with and without 100mM rhTrx. *P = 0.0152, ***P = 0.0002. **c-d**, Representative contour plots and graph show (c) frequencies of B cells with polarized (MtOR+MtG+) and depolarized mitochondria (MtOR−MtG+) (*P = 0.0476) and (d) frequencies of IL-10+B cells (*P = 0.0114) isolated from lymph nodes of SLE patients (n = 4) and non-autoimmune donors (Control) (n = 3), following 24h stimulation with CpGC, with and without 100mM rhTrx. Data analyzed by two-way ANOVA followed by Sidak's test for multiple comparisons.

| | |
|---|---|

# Reporting Summary

## Statistics

For all statistical analyses, confirm that the following items are present in the figure legend, table legend, main text, or Methods section.

| n/a | Confirmed | |
|---|---|---|
| ☐ | ☒ | The exact sample size (*n*) for each experimental group/condition, given as a discrete number and unit of measurement |
| ☐ | ☒ | A statement on whether measurements were taken from distinct samples or whether the same sample was measured repeatedly |
| ☐ | ☒ | The statistical test(s) used AND whether they are one- or two-sided<br>*Only common tests should be described solely by name; describe more complex techniques in the Methods section.* |
| ☒ | ☐ | A description of all covariates tested |
| ☐ | ☒ | A description of any assumptions or corrections, such as tests of normality and adjustment for multiple comparisons |
| ☐ | ☒ | A full description of the statistical parameters including central tendency (e.g. means) or other basic estimates (e.g. regression coefficient) AND variation (e.g. standard deviation) or associated estimates of uncertainty (e.g. confidence intervals) |
| ☐ | ☒ | For null hypothesis testing, the test statistic (e.g. *F*, *t*, *r*) with confidence intervals, effect sizes, degrees of freedom and *P* value noted<br>*Give P values as exact values whenever suitable.* |
| ☒ | ☐ | For Bayesian analysis, information on the choice of priors and Markov chain Monte Carlo settings |
| ☒ | ☐ | For hierarchical and complex designs, identification of the appropriate level for tests and full reporting of outcomes |
| ☐ | ☒ | Estimates of effect sizes (e.g. Cohen's *d*, Pearson's *r*), indicating how they were calculated |

*Our web collection on statistics for biologists contains articles on many of the points above.*

## Software and code

Policy information about availability of computer code

| Data collection | FACS Diva v9 |
|---|---|
| Data analysis | RStudio v4.2.1, Seurat v4.0, Monocle v3, Flowjo v10, Fiji v2.15.0 |

For manuscripts utilizing custom algorithms or software that are central to the research but not yet described in published literature, software must be made available to editors and reviewers. We strongly encourage code deposition in a community repository (e.g. GitHub). See the Nature Portfolio guidelines for submitting code & software for further information.

## Data

Policy information about availability of data

All manuscripts must include a data availability statement. This statement should provide the following information, where applicable:
- Accession codes, unique identifiers, or web links for publicly available datasets
- A description of any restrictions on data availability
- For clinical datasets or third party data, please ensure that the statement adheres to our policy

All data are available in the main text or the supplementary materials. For scRNA-seq transcriptomic analysis, the GRCh38 reference genome was downloaded using the Bioconductor annotation package BSgenome.Hsapiens.NCBI.GRCh38. The scRNA-seq dataset has been deposited in ArrayExpress, under accession code E-MTAB-13872. Link: https://www.ebi.ac.uk/biostudies/arrayexpress/studies/E-MTAB-13872.

# Research involving human participants, their data, or biological material

Policy information about studies with human participants or human data. See also policy information about sex, gender (identity/presentation), and sexual orientation and race, ethnicity and racism.

| | |
|---|---|
| Reporting on sex and gender | We have used gender based on patient characteristics available to us. |
| Reporting on race, ethnicity, or other socially relevant groupings | Population (SLE and Healthy donor) characteristics are reported in Table 1 |

Population characteristics

Population (SLE and Healthy donor) characteristics are reported in Table 1

SLE participant and healthy donor (HD) demographics used in each experiment are as follows;
Age is reported as F:M

7a - (Sex) HD 5:8, SLE 1:27; (Avg Age) HD 41.5 SLE 46.8
7b - (Sex) HD 4:6, SLE 1:29 (Avg Age) HD 39.4, SLE 46.6
7c - (Sex) HD 2:8, SLE 1:17; (Avg Age) HD 37.1, SLE 42.9
7d - (Sex) HD 2:9, SLE 1:22; (Avg Age) HD 33, SLE 38.5
7e - (Sex) HD 3:8, SLE 1:27; (Avg Age) HD 31.2, SLE 46.8
7f - (Sex) HD 5:8, SLE 1:25; (Avg Age) HD 31.2, SLE 45.8
7g - (Sex) HD 4:6, SLE 1:24; (Avg Age) HD 29.4, SLE 46.9
7h - (Sex) HD 1:4. SLE 1:10; (Avg Age) HD 29, SLE 34.5
7i - (Sex) HD 3:8, SLE 1:15; (Avg Age) HD 39.4, SLE 39.1

Extended Data Fig 9a - (CELLROX) (Sex) HD 2:4, SLE 1:9; (Avg Age) HD 32.5, SLE 35.1 (MITOSOX) (Sex) HD 1:4, SLE 0:9; Avg Age) HD 30.2, SLE 34.3
Extended Data Fig. 9b - (Sex) HD 2:3, SLE 0:8; (Avg Age) HD 27.2, SLE 30.8
Extended Data Fig. 9c - (Sex) HD 2:3, SLE 0:7; (Avg Age) HD 33, SLE 33.8
Extended Data Fig. 9d - (Sex) HD 2:6, SLE 1:7; (Avg Age) HD 29.75, SLE 36.8
Extended Data Fig. 9e - (Sex) HD 2:6, SLE 1:8; (Avg Age) HD 29.75, SLE 35.1
Extended Data Fig. 9f - (Sex) HD 0:3, SLE 0:3; (Avg Age) HD 30, SLE 36.6

Recruitment

SLE patients and healthy donors were recruited following informed consent. SLE patients were recruited from the University College London Hospital (UCLH) Rheumatology outpatient clinic. All patients recruited met at least 4 of the 11 American College of Rheumatology classification criteria, with a disease duration lasting more than 6 months, and positivity for antinuclear antibody (ANA) or anti-double-stranded DNA (dsDNA) autoantibodies. Patients under the age of 18, those treated with B cell depleting therapies (including rituximab, belimumab), those participating in any interventional trial, or pregnancy, were excluded from the study. Also excluded were patients with severe CNS lupus, glomerulonephritis, or congestive heart failure, and patients with a history of infections including HIV, hepatitis B/C and tuberculosis. Patients alto meeting criteria for other autoimmune diseases (such as multiple sclerosis, rheumatoid arthritis) were excluded. Participants did not receive compensation. Participants were recruited based on these criteria and attendance to the UCLH Rheumatology outpatients clinic. Healthy donors included research team co-workers. No self-selection bias that could impact results has been identified.

Ethics oversight

UCLH Health Service Trust ethics committee - REC reference no. 14/SC/1200

Note that full information on the approval of the study protocol must also be provided in the manuscript.

# Field-specific reporting

Please select the one below that is the best fit for your research. If you are not sure, read the appropriate sections before making your selection.

☒ Life sciences ☐ Behavioural & social sciences ☐ Ecological, evolutionary & environmental sciences

For a reference copy of the document with all sections, see nature.com/documents/nr-reporting-summary-flat.pdf

# Life sciences study design

All studies must disclose on these points even when the disclosure is negative.

| | |
|---|---|
| Sample size | No statistical methods were used to predetermine sample sizes, but our sample sizes are similar to those reported in previous publications (Shankar 2022; https://doi.org/10.1038/s41467-022-30613-z, Menon 2016; https://doi.org/10.1016/j.immuni.2016.02.012, Lighaam 2018; https://doi.org/10.3389/fimmu.2018.01913) |
| Data exclusions | All our data points have been included. |
| Replication | Experiments were repeated twice if sample size was >=6 or three times if sample size was between 3 and 6. All attempts at replication were successful |

| | |
|---|---|
| Randomization | No randomisation criteria could be applied. People were chosen according to disease (SLE or healthy donors), and allocated into these two groups. |
| Blinding | Data collection and analysis were not performed blind to the conditions of the experiments as all biological samples were labeled. |

# Reporting for specific materials, systems and methods

We require information from authors about some types of materials, experimental systems and methods used in many studies. Here, indicate whether each material, system or method listed is relevant to your study. If you are not sure if a list item applies to your research, read the appropriate section before selecting a response.

## Materials & experimental systems

| n/a | Involved in the study |
|---|---|
| ☐ | ☒ Antibodies |
| ☒ | ☐ Eukaryotic cell lines |
| ☒ | ☐ Palaeontology and archaeology |
| ☒ | ☐ Animals and other organisms |
| ☒ | ☐ Clinical data |
| ☒ | ☐ Dual use research of concern |
| ☒ | ☐ Plants |

## Methods

| n/a | Involved in the study |
|---|---|
| ☒ | ☐ ChIP-seq |
| ☐ | ☒ Flow cytometry |
| ☒ | ☐ MRI-based neuroimaging |

## Antibodies

| | |
|---|---|
| Antibodies used | BV785-conjugated mouse anti-human CD19 (Biolegend, Cat# 302239, clone HIB19, lot# B373234).<br>BV711-conjugated mouse anti-human CD24 (Biolegend, Cat# 311135, clone ML5, lot# B376031)<br>BV421-conjugated mouse anti-human CD38 (Biolegend, Cat# 356617, clone HB7, lot# B346969)<br>APC-conjugated rat anti-human IL-10 (Biolegend, Cat# 506807, clone JES3-19F1, lot# B370178)<br>PE-conjugated rat anti-human IL-6 (Biolegend, Cat# 501106, clone MQ2-13A5, lot# B365304)<br>PE/Cy7-conjugated mouse anti-human LAP(TGFb) (Biolegend, Cat# 300007, clone S20006A, lot# B371003)<br>PE-conjugated mouse anti-human IL-27/IL-35 (Biolegend, cat# 360903, clone B032F6, lot# B371876)<br>PE/Dazzle-conjugated rat anti-human GM-CSF (Biolegend, cat# 502317, clone BVD2-21C11, lot# B377133)<br>FITC-conjugated mouse anti-human IL-12/IL-35 p35 (R&D Systems, cat# IC2191F clone 27537 lot# LIR0721051)<br>AlexaFluor488-conjugated mouse anti-human Blimp1/PRDM1 (R&D Systems, cat# IC36081G, clone 646702, lot# ACTS0420031)<br>eFluor450-conjugated mouse anti-human TNF (ThermoFisher Scientific, cat# 48-7349-42, clone Mab11, lot# B370178)<br>eFluor450-conjugated mouse anti-human IFNg (ThermoFisher Scientific, cat# 48-7139-42, clone 4S.B3, lot# 273837<br>BUV395-conjugated mouse anti-human Ki67 (BD, cat# 564071, clone B56, lot# 1102683)<br>BUV395-conjugated mouse anti-human CD3 (BD, cat# 563546, clone UCHT-1, lot# 1214287)<br>PE-conjugated mouse anti-human CD4 (Biolegend, cat# 344605, clone SK3, lot# B303125)<br>Mouse anti-human thioredoxin (BioRad, cat# VMA00585, clone 3A1, lot# 161202)<br>Rabbit anti-human thioredoxin-2 (Abcam, cat# ab18554, clone EPR15225, lot# GR325111)<br><br>All surface marker antibodies (CD19, CD24, CD38, CD3, CD4) were used at 1:50 dilution.<br>All intracellular and intranuclear antibodies (IL-10, IL-6, TNF, IFNg, TGFb, IL-27/IL-35, IL-12/IL-35 p35, GM-CSF, Ki67, Blimp1, Trx, Trx2) were used at 1:100 dilution. |
| Validation | All antibodies have been validated by other publications and/or the manufacturer. |

## Plants

| | |
|---|---|
| Seed stocks | NA |
| Novel plant genotypes | NA |
| Authentication | NA |

# Flow Cytometry

## Plots

Confirm that:

☒ The axis labels state the marker and fluorochrome used (e.g. CD4-FITC).

☒ The axis scales are clearly visible. Include numbers along axes only for bottom left plot of group (a 'group' is an analysis of identical markers).

☒ All plots are contour plots with outliers or pseudocolor plots.

☒ A numerical value for number of cells or percentage (with statistics) is provided.

## Methodology

| | |
|---|---|
| Sample preparation | B cells were isolated from PBMCs and LNs using the EasySep™ immunomagnetic negative selection kit (STEMCELL, 19054). Isolated B cells were cultured for 72h with CpGC ODN 2395 (1µM, Invivogen tlrl-2395-1) in RPMI 1640 containing L-glutamine (Sigma-Aldrich). Media were supplemented with 10% fetal calf serum (FCS; LabTech) and 1% penicillin/streptomycin (100U/ml penicillin + 100µg/ml streptomycin; Sigma-Aldrich). To facilitate intracellular cytokine staining, cells were cultured in complete medium with PMA (50ng/ml; Sigma-Aldrich, P1585), ionomycin (250ng/ml, Sigma-Aldrich, I9657) and Brefeldin (5µg/ml; Sigma-Aldrich, B7651) for 4.5h. For multi-colour flow cytometric cell surface staining, cells were stained at 4oC for 30 minutes. LIVE/DEAD fixable blue Dead Cell Stain (ThermoFisher, L23105) was used to exclude dead cells from analysis. Cells were intracellularly fixed and permeabilised (ThermoFisher Scientific, 88-8824-00), or for additional detection of Blimp1 and Ki67 cells were fixed for 30 minutes using the FoxP3 Fixation buffer kit (ThermoFisher Scientific, 00-5523-00) before permeabilization. Cells were then incubated with intracellular and intranuclear antibodies for 30 minutes at 4oC. |
| Instrument | Flow cytometer: BD Pharmingen LSR II<br>Sorter: BD FACSAria Fusion |
| Software | BD FACS Diva v9.0 was used for data acquisition, Flowjo v10 was used for flow cytometry data analysis. |
| Cell population abundance | Reported values in the figures are percentages of either total CD19+B cells, or percentages of B cell phenotypic subsets |
| Gating strategy | Acquired samples were gated on FSC-A/SSC-A to select lymphocytes, FSC-H/FSC-A to exclude cell doublets, then cells negatively staining for Live/Dead Blue to exclude dead cells. Cells were then gated as FCS-A/CD19-BV785+ to identify B cells. All subsequent gating (e.g. CD24/CD38 to identify B cell subsets, IL-10+ B cells) were performed within the CD19+ gate, or for some figures IL-10+ or Blimp1+ cells were gated within CD24/CD38 B cell subsets. |

☒ Tick this box to confirm that a figure exemplifying the gating strategy is provided in the Supplementary Information.

