## [Peer Review File · Nature Immunology]

Peer Review Information

Journal: Nature Immunology

Manuscript Title: Thioredoxin: a metabolic rheostat controlling regulatory B cells

Corresponding author name(s): Professor Claudia Mauri; Dr Hannah Bradford

Reviewer Comments & Decisions:

Decision Letter, initial version:
--

11th Apr 2023

Dear Dr. Mauri,

We have now finished reviewing your manuscript entitled "Thioredoxin: a metabolic rheostat controlling regulatory B cells", reference number NI-A35350-T.

Although the editors thought that the manuscript was interesting enough to send out for in-depth review, the reviewers were not in favor of publishing the paper in Nature Immunology because although they are interested in the concept they both point to an over-reliance on inhibitors and some missing mechanistic insight.

We appreciate that the paper is a human study, but as you will see, reviewer 2 provides some suggestions that do not involve mouse work. Technically we think that the paper could be revised, but at this point we are not prepared to offer a standard route to revision as we think reviewer 1 has somewhat dismissed the paper because of this over-reliance on inhibitors, and so we think has not raised other possible issues that might be raised in any subsequent review. Having said all that, if you are prepared to perform some sort of genetic interference and address the mechanistic issues raised then we would be happy to see an appeal, but in that event and if we decided to accept the appeal then please be prepared that reviewer 1 might raise what you would consider new criticisms - although this is just my opinion, not anything the reviewer has told us explicitly.

We realize that this is disappointing. I hope that you continue to consider Nature Immunology for your results most significant for the immunology community and wish you well in your future investigations.

Sincerely,

Nick Bernard, PhD
Senior Editor

Nature Immunology

Reviewers' comments:

Reviewer #1 (Remarks to the Author):

This is an interesting paper that shows differential metabolism of Bregs compared to non-Bregs. The most interesting is TXN observation as well as which fuels could sustain mitochondrial metabolism, FAO or glutaminolysis or perhaps pyruvate metabolism. However, their sole reliance on pharmacologic interventions makes this not appropriate for Nature Immunology.

Major concerns:

- (1) Fuel utilization is an interesting aspect. However, there is GLS1 floxed mice (Rathmell paper in Cell) as well as FAO (CPT2 floxed mice- published by Toren Finkel in Nature metabolism). They could also use MPC floxed knockouts to test necessity of pyruvate dependent mitochondrial metabolism.
- (2) TXN is not clear data. Is it cytosolic TXN1 or mitochondrial TXN2? This is important as they argue the effects of pharmacological inhibition of Trx resulted in mitochondrial membrane depolarization and increased levels of ROS which selectively suppressed Bregs. The use of inhibitor is not appropriate. Also how does exogenous TRX rescue? How does the protein get into cells?
- (3) It is not clear whether ROS or some other aspect of mitochondrial respiratory function is to maintain Bregs.

Reviewer #2 (Remarks to the Author):

In their manuscript entitled "Thioredoxin: a metabolic rheostat controlling regulatory B cells", Braddford et al. investigated the key metabolic pathways regulating the differentiation of IL-10-producing regulatory B cells. The authors performed single-cell RNA sequencing of CpGC-induced IL-10-producing regulatory B cells (Bregs) and found that OXPHOS-related genes, particularly thioredoxin, were enriched in these cells. Using various inhibitors, they demonstrated that electron transport chain activity in mitochondria is crucial for expressing IL-10 from Bregs. The authors also found that SLE patients have a severe defect in mitochondrial function and IL-10 production from B cells, which can be rescued by adding thioredoxin.

Major Critique:

1. In this study, the authors show that CpGC-treated IL-10-producing B cells are regulatory B cells and represent a distinct subpopulation in their single-cell RNA seq data. However, the immune regulatory functionality of these cells is lacking. Do these B cells have an actual immune suppressive function? Have the authors ever tried the co-culture of this subset with other immune cells such as T cells? Can the treatment of rotenone or D9 block the suppressive function of these cells? Or does the blockade of Trx transdifferentiate the CpGC-treated B cells into more pro-inflammatory B cells able to potentiate effector T cells?
2. Although IL10 is one of the essential immuno-suppressive molecules from regulatory B cells, numerous studies have suggested other functional molecules derived from Bregs such as IL-27, IL-35 or immune checkpoint molecules. The authors only focused on the expression of IL-10 as a marker for

Breg differentiation. What happens to other immune suppressive molecules such as TGF-beta, IL-35, IL-27 or co-inhibitory molecules after CpGC-treatment with or without blockade of electron transport chain activity?

3. Activation induced-expression of IL-10 from B cells sometimes also comes together with pro-inflammatory cytokines. For example, LPS can induce IL-10 but it also triggers IL-6 from B cells at the same time. IL-6 or GM-CSF from B cells are known to be pathogenic in several autoimmune diseases. What was the expression of pro-inflammatory cytokines such as TNF- α , IL-6 or GM-CSF in CpGC-induced Bregs? For example, the authors showed Breg signature genes in Table 1., but some of pro-inflammatory molecules such as TNF and CD83 seems also upregulated which reflects on my previous comment to test the suppressive capacity of these cells. In addition, how pro-inflammatory gene expression is regulated with rotenone (electron transport chain activity inhibitor) or D9 (TrxR inhibitor) treatment?

4. The authors have shown the changes in IL-10 production by B cells by using several pharmaceutical inhibitors. However, many inhibitors can have possible off-target effects so that it is recommended to confirm the phenotype using gene-targeting (genetic) approaches. Have the authors ever tried the loss-of-function approaches using siRNA or Crispr-Cas9 targeting TXN or TXNIP genes in healthy or SLE patient samples? Or Have the authors tried lupus autoimmune disease mouse model using knockout mice having defect in these genes? One is available from Jax <https://www.jax.org/strain/030221>. Further did the authors check effect of Txn1 inhibitor on NADP/NADPH ratio?

5. The authors have demonstrated, in Extended Data Figure 5, that both CD40L and R848 are capable of inducing IL-10 production. Furthermore, several studies have indicated that IL-10 production can also be triggered by LPS, BCR signaling, and IL-21. Can we also call all of them as Bregs? If this is the case, what are the distinguishing features between these Bregs and those induced by CpGC? Are there any noteworthy differences in the involvement of the OXPHOS pathway in other forms of Breg differentiation?

6. The balance between TLR7 and TLR9 is known to play a critical role in SLE disease, both in vivo models of mice and in human patients. TLR9 signaling has a protective effect against SLE, so it is logical that CpGC-treatment would promote the differentiation of Bregs. However, there are still unresolved questions. The authors demonstrate that B cells from SLE patients with defective mitochondria produce less IL-10 in response to TLR9 ligand (CpGC). Is this due to a general defect in Breg differentiation in SLE patients, or does it only occur when TLR9 ligands are utilized? Do SLE patients have normal TLR9 components and signaling pathways? Have alternative methods, such as R848 (TLR7/8 agonist) or IL-21, been attempted to promote Breg differentiation or IL-10 production in SLE patient B cells? Do they still have defect in IL-10 production along with dysfunction of mitochondria?

7. In SLE patients, the authors showed that they produce less Trx in B cells. Please discuss the reason they have less expression of Trx in their B cells. Is there any mutation in their Trx locus? Do they express the normal level of Trx in other cell types but less only in B cells?

8. The authors demonstrated that high levels of cellular ROS inhibit Breg differentiation in (Figure 2). The correlation between ROS levels and IL-10 production was observed until CpGC concentration reached 1 μ M, where the ROS level continued to increase but IL-10 production began to decrease. In the 0-1 μ M range of CpGC, the ROS level appeared to be positively associated with IL-10 expression. It is unclear whether ROS aids IL-10 production in this 0-1 μ M range. When NAC is added to B cells treated with 0.1 or 0.5 μ M CpGC, it remains uncertain whether ROS continues to suppress or promote the expression of IL-10.

9. Authors showed higher ex vivo ROS in SLE patients than healthy control in B cells. Do they still have higher ROS after CpGC stimulation? Does SLE patient have a defect in the production of IL-10

from B cell in lower doses (0.1, 0.5uM) of CpGC?

10. The link between mitochondrial metabolism and Txn is weak. The authors do utilize a generalized ROS inhibitor. Did the authors try mitoQ or other mitochondrial ROS inhibitors and assess the effect on Bregs/ Il10 secretion?

11. Txn is activated under generalized oxidative stress. Txn is localized and functions in cytosol and upon activation can be translocated in the nucleus. Given the mitochondria-independent role of Txn, could exogenous ROS inducers increasing concentration of h202 mimic the effect of complex 1 inhibition in Bregs? Similarly, what is the effect of CpGC on mitoROS vs CytoROS?

Minor comments:

- The scRNAseq analysis would benefit more clarity regarding the marker gene expression patterns defining the author's annotations on the UMAP, further description of the CpGC de novo clusters differentially expressed genes and projection of known Breg signatures to further examine the changes caused by CpGC stimulation.

- Any Effect of CpGC and various inhibitors on B-cell proliferation/survival should be shown.

In summary, while the findings are interesting and provide insight into the regulation of IL-10-producing B cells, the manuscript could be improved by increasing mechanistic understanding of the observation of how the electron transport chain in mitochondria can regulate IL-10 production in B cells.

Author Rebuttal to Initial comments

See inserted PDF

We would like to take the opportunity to thank both reviewers for their insightful comments which we believe we have now addressed in full either by performing the experiments requested or by providing alternative experiments to address the question.

In particular, we have included the silencing of *TXN*, and *TXNRD1* and *TXN2* or *TXNRD2* and show the unicity of *TXN*, and *TXNRD1* silencing in suppressing IL-10⁺B cells differentiation. We have demonstrated that inhibition of Trx not only reduced IL-10 but also IL35, TGF β and PDL-1 whilst increasing the secretion of TNF and IL-6. We have also confirmed that TRX mediate their suppressive function, indeed inhibition of TRX dramatically reduces their capacity to suppress Th1 cells differentiation.

We also show that the levels of ROS determine the differentiation or suppression of Bregs. And that high concentration of ROS suppresses Bregs, whereas lower concentration is permissive for their differentiation and that Trx reduces cytoplasmic ROS but to an even greater extent mitochondrial ROS.

We show that exogenous FITC-tagged Trx can enter the cells and by confocal we show that Trx is co-localised with mitochondria.

We hope that the reviewers will find that the inclusion of these novel results has strengthened the manuscript which would be now suitable for publication.

The reviewer's comments are in **bold**.

Reviewer #1.

This is an interesting paper that shows differential metabolism of Bregs compared to non-Bregs. The most interesting is TXN observation as well as which fuels could sustain mitochondrial metabolism, FAO or glutaminolysis or perhaps pyruvate metabolism. However, their sole reliance on pharmacologic interventions makes this not appropriate for Nature Immunology.

Major concerns:

(1) Fuel utilization is an interesting aspect. However, there is GLS1 floxed mice (Rathmell paper in Cell) as well as FAO (CPT2 floxed mice- published by Toren Finkel in Nature metabolism). They could also use floxed knockouts to test necessity of pyruvate dependent mitochondrial metabolism.

We agree that it would be interesting to address these questions. However, to address the reviewers' concerns we will need to import 4 strains of mice (including the mb1Cre to generate conditional KO where the required gene is selectively deleted on B cells). The mice will then have to be rederived, as our mouse facilities don't accept any strains from non-approved facilities. Then cross them to Mb1-Cre mice to generate conditional KO mice. They will then need to be crossed into a susceptible lupus-prone strain. This will take a minimum of 2-3 years to get going with the

first experiment. Moreover, even KO mice such as CPT2 floxed mice (as suggested here) appears to have off-target effects as using a low concentration of etomoxir as tested in our manuscript¹.

To address this question, we cultured:

1.1.1. CpGC stimulated B cells under nutrient starvation or under repleting conditions. When B cells were cultured either in glucose-free media, or in glucose-free media replenished with high glucose concentrations, there were no alterations in the frequencies of IL-10⁺Bregs compared to the control group, thus supporting a redundant role for glycolysis on Bregs differentiation (Extended Data Fig. 1f).

1.1.2. The addition of malonyl-coA (an allosteric inhibitor limiting the uptake of fatty acids by mitochondria) to CpGC-stimulated B cells did not significantly alter Breg intracellular expression or secretion of IL-10 compared to the control group (Extended Data Fig. 1e).

To strengthen our findings of the reliance on OXPHOS for Bregs, we have added the following experiments:

1.1.3. We replaced glucose with galactose, a substrate known to promote oxidative phosphorylation², and showed that this replacement boosted IL-10⁺B cells differentiation compared to the control group (Extended Data Fig. 1g).

1.1.4. To strengthen our findings on Bregs dependency on glutamine metabolism we cultured B cells in glutamine-free media. Similarly, to our original data with the glutamine synthetase inhibitor BPTES (Figure 1b), B cells cultured in media deprived of glutamine presented with a reduced capacity to differentiate into Bregs. (Extended Data Fig. 1h).

Thus, the implementation of these alternative strategies has further confirmed that Bregs rely on OXPHOS and suggests that it is independent from not glycolysis or FAO metabolism.

(2) TXN is not clear data. Is it cytosolic TXN1 or mitochondrial TXN2? This is important as they argue the effects of pharmacological inhibition of Trx resulted in mitochondrial membrane depolarization and increased levels of ROS which selectively suppressed Bregs. The use of inhibitor is not appropriate.

1.2.1. Trx is not solely confined in the cytoplasm and has also been observed within the inner mitochondrial membrane, acting as a functional barrier regulating the movement of small molecules between the cytosol and the matrix. Additionally, Trx is recognized for its role in sustaining the proton gradient essential for driving oxidative phosphorylation³. To clarify the role of Trx and Trx2 in Breg differentiation we have silenced *TXN*, *TXNRD1*, as well as *TXN2* and *TXNRD2* (the latter encoding the mitochondrial isoform of TrxR). Comparable to the results obtained following the pharmacological inhibition of Trx with D9, silencing of *TXN* and *TXNRD1* inhibited Breg differentiation (Figure 5f, Extended Data Fig.9d), whereas B cells continued to express IL-10 following *TXN2* or *TXNRD2* silencing (Extended Data Figs. 9e-f). *TXN* silencing, similar to D9, resulted in a reduction in mitochondrial membrane polarization (Extended Data Figs 10b, Figure 6c) and increased levels of mitochondrial ROS production, with a comparatively lesser impact on cytoplasmic ROS production (Extended Data Figs 10d, e, Figure 6e).

1.2.2. We have also included additional flow cytometry analysis showing that there are no differences in the frequencies of Trx2⁺ cells between IL-10⁺ and IL-10⁻B cell subsets (Extended Data Fig. 7d-e). Thus, excluding the involvement of Trx-2 in Bregs differentiation.

We have added additional experiments strengthening the function of Trx in the maintenance of mitochondrial health and mitochondrial ROS (please refer to Rev #2 question#10, reply 2.10.1.).

Also how does exogenous TRX rescue? How does the protein get into cells?

1.2.4. TRX is an extremely small molecule (10-12 kDa) moving within different cellular compartments. To assess if exogenous Trx can enter B cells, we fluorescently labelled recombinant human Trx (rhTrx) and detected its intracellular presence by flow cytometry (Extended Data Fig. 11j).

(3) It is not clear whether ROS or some other aspect of mitochondrial respiratory function is to maintain Bregs.

1.3.1 We apologise but we are not sure that we have fully understood this question. We cannot exclude that in addition to ROS other aspects of mitochondrial respiratory function are involved in the maintenance of Bregs and this is part of our continued investigations. However, we have now showed that the amount of ROS present in the culture system determine whether B cells differentiate into Bregs or B-effector cells and that complex I and III of the ETC play an important role, and indirectly Trx which keeps ROS to “permissive” levels that control Bregs over B-effector cells differentiation. Please also refer to the detailed reply for Rev#2 questions #8, 9 and 10.

Reviewer #2 (Remarks to the Author):
In their manuscript entitled "Thioredoxin: a metabolic rheostat controlling regulatory B cells", Bradford et al. investigated the key metabolic pathways regulating the differentiation of IL-10-producing regulatory B cells. The authors performed single-cell RNA sequencing of CpGC-induced IL-10-producing regulatory B cells (Bregs) and found that OXPHOS-related genes, particularly thioredoxin, were enriched in these cells. Using various inhibitors, they demonstrated that electron transport chain activity in mitochondria is crucial for expressing IL-10 from Bregs. The authors also found that SLE patients have a severe defect in mitochondrial function and IL-10 production from B cells, which can be rescued by adding thioredoxin.

Major Critique:

2 In this study, the authors show that CpGC-treated IL-10-producing B cells are regulatory B cells and represent a distinct subpopulation in their single-cell RNA seq data. However, the immune regulatory functionality of these cells is lacking. Do these B cells have an actual immune suppressive function? Have the authors ever tried the co-culture of this subset with other immune cells such as T cells? Can the treatment of rotenone or D9 block the suppressive function of these cells?

2.1.1. We have now included new data showing that and that blocking either complex I or III in the ETC (by rotenone and Antimycin A respectively) or Trx activity by D9 inhibits the suppressive capacity of B regs. We cultured B cells for 48h with CpGC, with and without rotenone, Antimycin A or D9 before co-culturing with autologous T cells stimulated with anti-CD3. B cells conditioned with rotenone, Antimycin A, or with D9 in addition to reduced IL-10 expression, failed to suppress CD4⁺T cell IFN γ production, compared to control Bregs (Fig. 1c), (Fig. 5e).

Or does the blockade of Trx transdifferentiate the CpGC-treated B cells into more pro-inflammatory B cells able to potentiate effector T cells?

2.1.2. In addition to IL-10, inhibition of Trx by D9 also reduced the frequencies of TGF β ⁺, IL-35⁺ and PD-L1⁺B cells, which have been ascribed to the regulatory function of Bregs (Extended Data Figs. 8d,e,f). The decrease in immunoregulatory cytokines following Trx inhibition was accompanied by increased frequencies of CD19⁺IL-10⁺TNF⁺, in addition to increased TNF and IL-6 secretion (Figs. 5c,d, Extended Data Figs. 8g,h).

2. Although IL10 is one of the essential immuno-suppressive molecules from regulatory B cells, numerous studies have suggested other functional molecules derived from Bregs such as IL-27, IL-35, or immune checkpoint molecules. The authors only focused on the expression of IL-10 as a marker for Breg differentiation. What happens to other immune suppressive molecules such as TGF-beta, IL-35, IL-27 or co-inhibitory molecules after CpGC-treatment with or without blockade of electron transport chain activity?

2.2.1. We have included new scRNA-seq analysis showing that the Breg clusters also express genes encoding TGF β (*TGFB1*) and IL-35 (*EBI3*, *IL-12A*) (Extended Data Fig. 6b). More importantly, we have also included new flow cytometry data showing that CD19⁺IL-10⁺B cells co-express IL-35 and TGF β , and that blocking either complex I or III in the ETC (by rotenone and Antimycin A respectively) prevents the differentiation of IL-10⁺IL-35⁺ and IL-10⁺TGF β ⁺ B cells (Extended Data Figs 2b,c).

We were unable to detect IL-27 by B cells under any of the conditions tested (including after stimulation with CpGC, CD40L, α BCR or in combination), and have not included these results in the manuscript due to severe space constraints.

3. Activation induced-expression of IL-10 from B cells sometimes also comes together with pro-inflammatory cytokines. For example, LPS can induce IL-10 but it also triggers IL-6 from B cells at the same time. IL-6 or GM-CSF from B cells are known to be pathogenic in several autoimmune diseases.

For example, the authors showed Breg signature genes in Table 1, but some of pro-inflammatory molecules such as TNF and CD83 seems also upregulated which reflects on my previous comment to test the suppressive capacity of these cells.

2.3.1. Please refer to our previous answer and to Extended Data Fig. 6b. Our results show that the IL10⁺B cell clusters express higher levels of *TGFB1*, *EBI3* and *CD83*, TGF β and CD83 compared to *TNF*, *IL6*, *CFS2* (Extended Data Fig. 6b). We hope that the inclusion of the data showing that this subset express immunoregulatory cytokines and the suppression assays described above and shown in Figs 1c, 5e has strengthened our findings.

In response to the upregulation of CD83, several studies have shown that CD83 is not a typical co-stimulatory molecule, and indeed has been involved in dampening inflammation in chronic disorders⁴ and in the maintenance of tolerance. Thus, it is possible that in the context of Bregs, CD83 represents another component in the armoury of B cell regulatory mechanisms. As we are well over the word limit due to the extensive revisions, we have not included this in the discussion but will be happy to do so if this reviewer thinks it is needed.

In addition, how pro-inflammatory gene expression is regulated by rotenone (electron transport chain activity inhibitor) or D9 (TrxR inhibitor) treatment?

We show that rotenone and Antimycin A reduced the frequencies of IL-10⁺TNF⁺ and IL-10⁺IL-6⁺ B cells, however there were no changes to the frequencies of IL-10⁻TNF⁺, IL-10⁻IL-6⁺ B cells, and a trend to increased frequencies of IL-10⁻GM-CSF⁺ (Extended Data Figs. 2d,e). This therefore leads to a higher ratio of pro-inflammatory cells over Bregs.

Interestingly TrxR inhibition with D9 selectively reduced IL-10⁺TNF⁻ and IL-10⁺IL-6⁻B cell frequencies, increased the frequencies of IL-10⁻TNF⁺B cells, and increased TNF and IL-6 secretion (Fig. 5c,d, Extended Data Fig. 8g,h). While we did not perform scRNA-seq after rotenone, we hope that the inclusion of this new flow cytometry data showing the changes at the protein level is sufficient.

4. The authors have shown the changes in IL-10 production by B cells by using several pharmaceutical inhibitors. However, many inhibitors can have possible off-target effects so that it is recommended to confirm the phenotype using gene-targeting (genetic)x approaches. Have the authors ever tried the loss-of-function approaches using SiRNA or Crispr-Cas9 targeting TXN or TXNIP genes in healthy or SLE patient samples?

We have successfully silenced *TXN*, *TXNRD1*, as well as *TXN2* and *TXNRD2* and confirmed the function of Trx in the differentiation of Bregs. Please refer to reply to question raised by Rev1 answers: 1.2.1. 1.2.2.

The expression levels of Trx B cells from SLE patients are virtually undetectable, and this is the reason why we chose these patients as a surrogate of the “Trx B cell-KO mice”. It is also important to note that B cells expressed the highest levels of Trx in healthy individuals, and other immune cell subsets expressed marginal levels of Trx (Extended Data Fig. 11f).

Or have the authors tried lupus autoimmune disease mouse model using knockout mice having defect in these genes? One is available from Jax <https://www.jax.org/strain/030221>. Please refer to our first answer to reviewer 1 regarding performing experiments in mice.

Further did the authors check effect of Txn1 inhibitor on NADP/NADPH ratio?

Unfortunately, we have tried multiple time to perform this experiment several times, but it has not been possible due to the insufficient amount of B cells available. The kit requires 3 million B cells per sample, so for 3 experimental conditions we require upwards of 9 million B cells. Unfortunately, although we tried by using blood cones, we did not obtain the numbers of live B cells required to perform this experiment. We hope that the inclusion of all the other experiments is sufficient to strengthened further our data.

5. The authors have demonstrated, in Extended Data Figure 5, that both CD40L and R848 are capable of inducing IL-10 production. Furthermore, several studies have indicated that IL-10 production can also be triggered by LPS, BCR signaling, and IL-21. Can we also call all of them as Bregs? If this is the case, what are the distinguishing features between these Bregs and those induced by CpGC? Are there any noteworthy differences in the involvement of the OXPHOS pathway in other forms of Breg differentiation?
2.5.1. We confirm that rotenone and Antimycin A suppress Bregs differentiation under all the conditions tested where we were able to measure detectable levels of IL-10 (CpGC, R848, LPS, CD40L) (Extended Data Fig. 2a). We did not detect IL-10 expression when B cells were stimulated with either LPS, anti-BCR or IL-21 alone, thus confirming that regardless how Bregs are generated they share a reliance on OXPHOS. Of note, unlike mouse B cells the expression of TLR4 in human B cells is very low⁵.

6. The balance between TLR7 and TLR9 is known to play a critical role in SLE disease, both in vivo models of mice and in human patients. TLR9 signaling has a protective effect against SLE, so it is logical that CpGC-treatment would promote the differentiation of Bregs. However, there are still unresolved questions. The authors demonstrate that B cells from SLE patients with defective mitochondria produce less IL-10 in response to TLR9 ligand (CpGC). Is this due to a general defect in Breg differentiation in SLE patients, or does it only occur when TLR9 ligands are utilized?

2.6.1. Here we have extended the significance of these results and tested the production of IL-10 after stimulation with the TLR7 ligand R848 and confirmed that B cells from SLE patients express less IL-10 than B cells from the control group (Extended Data Fig. 11e).

In addition, we and others have previously shown that different types of stimulation, for example with agonistic anti-CD40, or with anti-CD40+IFN α , induces Bregs in healthy controls but not in SLE patients, suggesting a generalised rather than a TLR9 ligand linked defects.

Do SLE patients have normal TLR9 components and signaling pathways?

2.6.2. We have previously shown that B cells expressed the same mRNA levels of TLR9 in healthy and SLE⁶. However, it has been extensively reported that SLE patients present with abnormal TLR signalling compared to healthy individuals⁷, further supported by the findings showing the efficacy of hydroxychloroquine (a TLR7/9 antagonist) in SLE patients⁸. Recently it has been reported that there are polymorphisms in genes that are shared amongst TLR and IFN-I signaling pathways, (transcription factors interferon regulatory factor (IRF) 5, IRF7, and IRF8, interleukin-1 receptor-associated kinase 1 (IRAK1), tumor necrosis factor, alpha-induced protein 3 (TNFAIP3)⁹.

Have alternative methods, such as R848 (TLR7/8 agonist) or IL-21, been attempted to promote Breg differentiation or IL-10 production in SLE patient B cells? Do they still have defect in IL-10 production along with dysfunction of mitochondria?

2.6.3. We have shown that R848 fails to restore mitochondrial membrane depolarisation or to induce IL-10⁺Breg differentiation in SLE patient B cells (Extended Data Figs. 11c,e). As described above and shown in Extended Data Fig. 2a, IL-21 stimulation alone does not induce IL-10⁺Bregs in healthy individuals, which was also true for SLE patient B cells and due to space constraints, we have not included this in the manuscript.

7. In SLE patients, the authors showed that they produce less Trx in B cells. Please discuss the reason they have less expression of Trx in their B cells. Is there any mutation in their Trx locus?

2.7.1. Using gnomAD (https://gnomad.broadinstitute.org/gene/ENSG00000136810?dataset=gnomad_r2_1) we took the known SNPs (both exonic and intronic) in the *TXN* gene and cross-referenced their respective rsID's to published GWAS datasets in SLE and known SLE associated loci. (<https://academic.oup.com/hmg/article/29/10/1745/5741615> ; <https://www.nature.com/articles/s41467-021-21049-y> ; <https://ard.bmj.com/content/81/9/1273>). We found no *TXN* SNPs in three separate GWAS studies. Given the low allele frequency of these SNPs, we believe it is unlikely that the reduction in Trx in SLE B cells is due to any given SNP. We will add a comment to the discussion to speculate as to what could drive lower expression of Trx in SLE B cells. We have previously shown that the lack of IL-10 is only observed in patients with active disease and that for example repopulating B cell after rituximab, in patients that are responding to the therapy, have regained the capacity to produce IL-10⁵. Inferring that the defect is a consequence of inflammation and not genetic. We have included these findings in the discussion.

Do they express the normal level of Trx in other cell types but less only in B cells?

2.7.2. This is correct. B cells express the highest levels of Trx compared to other immune-cells in healthy individuals and much higher levels compared to SLE B cells; no differences in Trx expression were identified amongst other immune-cells in SLE (Extended Data Fig. 11f).

8. The authors demonstrated that high levels of cellular ROS inhibit Breg differentiation in (Figure 2). The correlation between ROS levels and IL-10 production was observed until CpGC concentration reached 1 μ M, where the ROS level continued to increase but IL-10 production began to decrease. In the 0-1 μ M range of CpGC, the ROS level appeared to be positively associated with IL-10 expression. It is unclear whether ROS aids IL-10 production in this 0-1 μ M range. When NAC is added to B cells treated with 0.1 or 0.5 μ M CpGC, it remains uncertain whether ROS continues to suppress or promote the expression of IL-10.

2.8.1. We cultured B cells with increasing doses of CpGC (0.1-5 μ M) with 1mM NAC, which reduced cytoplasmic ROS levels at all CpGC doses tested (Extended Data Fig. 5a). At higher CpGC concentrations (1 μ M, 5 μ M), NAC increased the frequencies of IL-10⁺Bregs, while at lower CpGC doses (0.1 μ M, 0.5 μ M) the frequencies of IL-10⁺Bregs were reduced (Fig 2c). Of note, we did not observe changes in mitochondrial ROS levels when B cells were cultured in the presence of NAC indicating that NAC only inhibits cytosolic ROS (Extended Data Fig. 5b). We also cultured B cells with MitoTempo, a selective inhibitor of mitochondrial ROS. MitoTempo reduced mitochondrial ROS levels at all CpGC concentrations tested and, like NAC, increased IL-10⁺B cell frequencies at high CpGC concentrations (1 μ M, 5 μ M), and reduced IL-10⁺Breg frequencies at low CpGC concentrations (0.1 μ M, 0.5 μ M) (Fig. 2c, Extended Data Fig. 5c).

Collectively this shows that a moderate rise in levels of ROS is required for IL-10⁺Breg differentiation, and high levels are detrimental for Bregs differentiation.

9. Authors showed higher ex vivo ROS in SLE patients than healthy control in B cells. Do they still have higher ROS after CpGC stimulation? Does SLE patient have a defect in the production of IL-10 from B cell in lower doses (0.1, 0.5 μ M) of CpGC?

2.9.1. We have included new data showing that SLE patient B cells have higher levels of mitochondrial ROS (whereas not changes in cytosol ROS were identified) both *ex vivo* and after stimulation with CpGC (Fig. 7d, Extended Data Fig. 11a). We have also included new data showing that SLE patient B cells present reduced frequencies of IL-10⁺Bregs and increased mitochondrial membrane depolarisation even after stimulation with 0.1 μ M CpGC (Extended Data Figs. 11b,d).

10. The link between mitochondrial metabolism and Txn is weak. The authors do utilize a generalized ROS inhibitor. Did the authors try mitoQ or other mitochondrial ROS inhibitors and assess the effect on Bregs/ IL10 secretion?

2.10.1. In addition to NAC, we have now included in Figure 2 the selective mitochondrial ROS inhibitor MitoTempo (we found that MitoQ had a toxic effect on B cells even at lower concentrations). We performed the CpGC titration experiment as described above with 10 μ M MitoTempo, which reduced mitochondrial ROS levels at all CpGC doses tested (Extended Data Fig. 5c). Similar to the results obtained following NAC stimulation, MitoTempo treatment with lower CpGC doses reduces the frequencies of IL-10⁺Bregs, while at high CpGC concentrations the frequencies of IL-10⁺Bregs were increased (Fig 2c). Therefore, moderate raise in mitochondrial ROS is required for IL-10⁺Breg differentiation but suppress IL-10⁺Bregs at high concentrations.

11. Txn is activated under generalized oxidative stress. Txn is localized and functions in cytosol and upon activation can be translocated in the nucleus. Given the mitochondria-independent role of Txn, could exogenous ROS inducers increasing concentration of H₂O₂ mimic the effect of complex 1 inhibition in Bregs? Similarly, what is the effect of CpGC on mitoROS vs CytoROS?

2.11.1. While Trx is classically regarded as cytoplasmic/nuclear, it is not solely confined to the cytoplasm and has been found within the inner mitochondrial membrane, acting as a functional barrier regulating the movement of small molecules between the cytoplasm and mitochondrial matrix^{3,10}. Trx, as for Trx2, has also been recognised for its role in sustaining the proton gradient required for driving OXPHOS³, thus supporting the participation of Trx in the regulation of mitoROS and mitochondria polarization. In support of these data, we observed that Trx inhibition with D9 or gene silencing increased the levels of mitochondrial ROS in B cells, and by confocal microscopy found Trx colocalised with polarised mitochondria (Figs. 6d,e, Extended Data Figs. 10c-e). These results combined with the other findings showing that Trx inhibition reduces mitochondrial respiration rate (Figs. 6a), strongly support a mitochondria-dependent role of Trx.

2.11.2. We have also cultured B cells with H₂O₂ and found that at higher concentrations (10μM the highest dose that could be used without impairing cell survival), H₂O₂ mimicked the effects of rotenone and reduced the frequencies of IL-10⁺Bregs (Fig. 2d). In contrast, B cells stimulated with low CpGC (0.1μM) and cultured with low concentrations of H₂O₂ (0.2μM) expressed more IL-10 compared to control B cells, supporting our findings detailed in response to questions 8 and 10 that moderate concentration of ROS are permissive for IL-10⁺Breg differentiation.

It is important to note the existence of a crosstalk between mitochondrial and cytosol derived ROS which has been previously reported¹¹. For example, it has been previously shown that the production of cytosol ROS led to activation of the NADPH oxidases, conversion of xanthine dehydrogenase to xanthine oxidase, which can stimulate the production of mitochondrial ROS¹¹.

Thus, addition of exogenous H₂O₂ mimics a similar effect of complex I inhibitor.

2.11.3. Regarding the effects of CpGC on mitochondrial versus cytoplasmic ROS, increasing concentrations of CpGC induced incremental increases in both cytoplasmic and mitochondrial ROS (Figs 2a,b).

Minor comments:

The scRNAseq analysis would benefit more clarity regarding the marker gene expression patterns defining the author's annotations on the UMAP, further description of the CpGC de novo clusters differentially expressed genes and projection of known Breg signatures to further examine the changes caused by CpGC stimulation.

We have now addressed this issue and included new analysis in Extended Data Fig. 6b and commentary in the main text.

Any Effect of CpGC and various inhibitors on B-cell proliferation/survival should be shown.

We have included the effects on proliferation and/or survival in Extended Data Figs. 1i and 8c,

References

1. Yao, C. H. *et al.* Identifying off-target effects of etomoxir reveals that carnitine palmitoyltransferase i is essential for cancer cell proliferation independent of β -oxidation. *PLoS Biol* **16**, (2018).
2. Aguer, C. *et al.* Galactose enhances oxidative metabolism and reveals mitochondrial dysfunction in human primary muscle cells. *PLoS One* **6**, (2011).
3. Subramani, J., Kundumani-Sridharan, V. & Das, K. C. Thioredoxin protects mitochondrial structure, function and biogenesis in myocardial ischemia-reperfusion via redox-dependent activation of AKT-CREB- PGC1 α pathway in aged mice. *Aging* **12**, (2020).
4. Feng, M. *et al.* CD83+ B cells alleviate uveitis through inhibiting DCs by sCD83. *Immunology* (2023) doi:10.1111/imm.13654.
5. Bekeredjian-Ding, I. & Jego, G. Toll-like receptors - Sentries in the B-cell response. *Immunology* vol. 128 Preprint at <https://doi.org/10.1111/j.1365-2567.2009.03173.x> (2009).
6. Menon, M., Blair, P. A., Isenberg, D. A. & Mauri, C. A Regulatory Feedback between Plasmacytoid Dendritic Cells and Regulatory B Cells Is Aberrant in Systemic Lupus Erythematosus. *Immunity* **44**, (2016).
7. Fillatreau, S., Manfroi, B. & Dörner, T. Toll-like receptor signalling in B cells during systemic lupus erythematosus. *Nature Reviews Rheumatology* vol. 17 Preprint at <https://doi.org/10.1038/s41584-020-00544-4> (2021).
8. Kužnik, A. *et al.* Mechanism of Endosomal TLR Inhibition by Antimalarial Drugs and Imidazoquinolines. *The Journal of Immunology* **186**, (2011).
9. Rullo, O. J. & Tsao, B. P. Recent insights into the genetic basis of systemic lupus erythematosus. *Ann Rheum Dis* **72**, (2013).
10. Vögtle, F. N. *et al.* Intermembrane space proteome of yeast mitochondria. *Molecular and Cellular Proteomics* **11**, (2012).
11. Dikalov, S. Cross talk between mitochondria and NADPH oxidases. *Free Radical Biology and Medicine* vol. 51 1289–1301 Preprint at <https://doi.org/10.1016/j.freeradbiomed.2011.06.033> (2011).

Decision Letter, first revision:

26th Jan 2024

Dear Dr. Mauri,

Thank you for submitting your revised manuscript "Thioredoxin: a metabolic rheostat controlling regulatory B cells" (NI-A35350B). As you know it has now been seen by the original referees and their comments are below. We also looked over your author response to the lingering concerns of reviewer 2.

Although we agree with the criticisms raised by reviewer 2, we do not think that it is necessary to perform mouse modelling as your paper is clearly a human immunology paper. With regards to their other remaining concern (point 2), we have looked over your response and concur that resolving these issues would be an extensive undertaking that is not necessary here. However, we would like you to account for both of these limitations noted by reviewer 2 with some further textual revision, ensuring that your conclusions are not overstated and any gaps are clearly conceded. Assuming you make these and other changes we will instruct you for editorial/formatting purposes, we'll be happy in principle to publish it in Nature Immunology.

We will now perform detailed checks on your paper and will send you a checklist detailing our editorial and formatting requirements in about a week. Please do not upload the final materials and make any revisions until you receive this additional information from us.

If you had not uploaded a Word file for the current version of the manuscript, we will need one before beginning the editing process; please email that to immunology@us.nature.com at your earliest convenience.

Thank you again for your interest in Nature Immunology. Please do not hesitate to contact me if you have any questions.

Sincerely,

Nick Bernard, PhD
Senior Editor
Nature Immunology

Reviewer #1 (Remarks to the Author):

I am satisfied.

Reviewer #2 (Remarks to the Author):

In the revised paper "Thioredoxin: a metabolic rheostat controlling regulatory B cells", the authors have added significant new data, in response to the comments raised by the reviewer. These include

functional validation showing the T cell suppression by CpGC-polarized IL-10 producing “Breg,” and the immunoregulatory effect is suppressed by treatment of the thioredoxin reductase inhibitors, D9. In addition to suppressing anti-inflammatory cytokines such as IL-10 and TGFb, they also showed that thioredoxin reductase inhibition can increase the pro-inflammatory cytokines TNFa or IL-6 in B cells, which is interesting. They have also applied the Crispr-Cas9 KO system to delete the TXN or TXNRD1 gene and showed that TXN expression is important for the expression of IL-10 in B cells.

This new work has added significantly to the understanding of the paper. However, there are still a few issues that remain unresolved.

1. The paper is still missing the effect of thioredoxin on the progression of SLE. Even though they showed the co-culture of T and B cells in vitro, it does not necessarily mean it affects the in vivo mouse disease model. Generation of the B cell-specific TXN KO in the Lpr strain will be the best. The authors can also try B cell transfer or pharmacological approaches targeting Txn-related pathways. In vitro cell culture systems have many limitations in a dish, primarily if it is related to metabolism or reactive oxygen species. It would be critical to show the effect of Txn or Txn-related pathways of regulatory B cells in vivo autoimmune disease mouse system.
2. The paper lacks an explanation of why SLE patients have less Trx, especially in B cells resulting in less IL-10. The authors answered in the rebuttal letter that they could not find Txn in the GWAS study in SLE patients and mentioned that it “infers the defect is a consequence of inflammation and not genetic”. In addition, according to the author’s response, less IL-10 and Txn expression in SLE patients are not limited to the CpG-TLR9 pathway but can also happen in other conditions like TLR7 agonist treatment. Then, what is the possible mechanism that SLE patients have fewer Txn+ B cells? Can they find the link between less Txn and other known risk genes in SLE patients? Can the authors describe the mechanism of this phenomenon?

While the new data and findings about Txn and the immune-regulatory function of B cells are quite interesting, the authors could not show the effect of thioredoxin on B cells in vivo mouse disease models either by using genetic or pharmacological approaches.

Author Rebuttal, first revision:

Dear Dr Bernard,

Firstly, we would like to express our gratitude to the Editor for providing us with the opportunity to address the reviewer’s request to establish a genetic link between SLE and Trx. In response to reviewer 1 we like to discuss plausible mechanisms underlying the Trx deficiencies in SLE B cells in response to reviewer 2 question 2 we have included the following points in the discussion and new supporting results in Supplementary Data 5.

1. We have shown that rotenone (known to inhibit complex I of the ETC) suppresses Trx expression and IL-10 completely, leading to a burst in mitochondrial ROS production. It is well known that SLE patients have defects in complex I and III subunits and increased

levels of mitochondrial ROS (which we have shown) (Leishangthem et al., Lupus 2016). Therefore, the most plausible explanation is that is this mitochondrial defects that leads to Trx deficiency in SLE.

2. Due to the scarcity of information regarding factors inducing Trx, it is challenging to find if there is any direct link to known risk alleles for SLE. The Keap1-Nrf2 pathway is the principal protective response to oxidative stress, and Nrf2 (nuclear factor (erythroid-derived 2)-like 2) is one of the few transcription factors known to induce *TXN* expression (Malhotra et al., Nucleic Acids Res 2010). In our scRNA-seq dataset, we show that *KEAP1* expression is restricted to Breg-enriched clusters. Similarly, *NFE2L2* (encoding NRF2) expression is concentrated within Breg-enriched clusters C2 and C3 (Figure 1). There is evidence suggesting that Nrf2 has a central role in the pathogenesis of SLE. Both mouse models and genome-wide scans have identified Nrf2 as a candidate gene for susceptibility to SLE. Interestingly, aged female Nrf2-deficient mice are prone to developing an autoimmune condition closely resembling human SLE (Yoh et al., Kidney Int 2001). As Nrf2 is also linked to mitochondrial health (Esteras et al., Free Radical Biology and Medicine 2022), one could argue that the defect in Nrf2 expression contributes to the observed mitochondrial defects in SLE, thus consolidating our previous point.
3. An alternative but not mutually exclusive possibility is that, as previously shown, ROS can induce promoter region hypermethylation (Niu et al., Free Radic Biol Med 2015). We propose that the *TXN* locus is hypermethylated, and consequently inactivated, due to high levels of ROS in SLE. This would also explain why diverse stimuli (TLR9/TLR7 stimulation) do not restore Bregs in SLE patients. One might speculate about an environmental event that, together with some genetic predisposition, initially increases ROS in B cells (e.g. causing mitochondrial dysfunction), leading to a vicious cascade that increases DNA/histone methylation, interferes with *TXN* expression, and consequently results in reduced IL-10 expression.

Final Decision Letter:

Dear Dr. Mauri,

I am delighted to accept your manuscript entitled "Thioredoxin is a metabolic rheostat controlling regulatory B cells" for publication in an upcoming issue of Nature Immunology.

Over the next few weeks, your paper will be copyedited to ensure that it conforms to Nature Immunology style. Once your paper is typeset, you will receive an email with a link to choose the appropriate publishing options for your paper and our Author Services team will be in touch regarding any additional information that may be required.

After the grant of rights is completed, you will receive a link to your electronic proof via email with a

request to make any corrections within 48 hours. If, when you receive your proof, you cannot meet this deadline, please inform us at rjsproduction@springernature.com immediately.

Please note that *Nature Immunology* is a Transformative Journal (TJ). Authors may publish their research with us through the traditional subscription access route or make their paper immediately open access through payment of an article-processing charge (APC). Authors will not be required to make a final decision about access to their article until it has been accepted. Find out more about Transformative Journals.

Your paper will be published online soon after we receive your corrections and will appear in print in the next available issue.

Also, if you have any spectacular or outstanding figures or graphics associated with your manuscript - though not necessarily included with your submission - we'd be delighted to consider them as candidates for our cover. Simply send an electronic version (accompanied by a hard copy) to us with a possible cover caption enclosed.

If you have not already done so, we strongly recommend that you upload the step-by-step protocols used in this manuscript to the Protocol Exchange. Protocol Exchange is an open online resource that allows researchers to share their detailed experimental know-how. All uploaded protocols are made freely available, assigned DOIs for ease of citation and fully searchable through nature.com. Protocols can be linked to any publications in which they are used and will be linked to from your article. You can also establish a dedicated page to collect all your lab Protocols. By uploading your Protocols to Protocol Exchange, you are enabling researchers to more readily reproduce or adapt the methodology you use, as well as increasing the visibility of your protocols and papers. Upload your Protocols at www.nature.com/protocolexchange/. Further information can be found at www.nature.com/protocolexchange/about .

Please note that we encourage the authors to self-archive their manuscript (the accepted version before copy editing) in their institutional repository, and in their funders' archives, six months after publication. Nature Portfolio recognizes the efforts of funding bodies to increase access of the research they fund, and strongly encourages authors to participate in such efforts. For information about our editorial policy, including license agreement and author copyright, please visit www.nature.com/ni/about/ed_policies/index.html

Sincerely,

Nick Bernard, PhD
Senior Editor
Nature Immunology